# The double-stranded DNA-binding proteins TEBP-1 and TEBP-2 form a telomeric complex with POT-1

Sabrina Dietz [1,6], Miguel Vasconcelos Almeida [1,4,5,6], Emily Nischwitz[1], Jan Schreier[1], Nikenza Viceconte [1], Albert Fradera-Sola [1], Christian Renz[1], Alejandro Ceron-Noriega[1], Helle D. Ulrich[1], Dennis Kappei [2,3], René F. Ketting [1] & Falk Butter [1✉]

Telomeres are bound by dedicated proteins, which protect them from DNA damage and regulate telomere length homeostasis. In the nematode *Caenorhabditis elegans*, a comprehensive understanding of the proteins interacting with the telomere sequence is lacking. Here, we harnessed a quantitative proteomics approach to identify TEBP-1 and TEBP-2, two paralogs expressed in the germline and embryogenesis that associate to telomeres in vitro and in vivo. *tebp-1* and *tebp-2* mutants display strikingly distinct phenotypes: *tebp-1* mutants have longer telomeres than wild-type animals, while *tebp-2* mutants display shorter telomeres and a Mortal Germline. Notably, *tebp-1;tebp-2* double mutant animals have synthetic sterility, with germlines showing signs of severe mitotic and meiotic arrest. Furthermore, we show that POT-1 forms a telomeric complex with TEBP-1 and TEBP-2, which bridges TEBP-1/-2 with POT-2/MRT-1. These results provide insights into the composition and organization of a telomeric protein complex in *C. elegans*.

[1] Institute of Molecular Biology (IMB), Mainz, Germany. [2] Cancer Science Institute of Singapore, National University of Singapore, Singapore, Singapore. [3] Department of Biochemistry, Yong Loo Lin School of Medicine, National University of Singapore, Singapore, Singapore. [4] Present address: Wellcome Trust/ Cancer Research UK Gurdon Institute, University of Cambridge, Cambridge, UK. [5] Present address: Department of Genetics, University of Cambridge, Cambridge, UK. [6] These authors contributed equally: Sabrina Dietz, Miguel Vasconcelos Almeida. ✉email: f.butter@imb-mainz.de

Most telomeres in linear eukaryotic chromosomes end in tandem repeat DNA sequences. Telomeres solve two major challenges of chromosome linearity: the end-protection problem and the end-replication problem[1,2]. The end-protection problem originates from the structural similarity between telomeres and DNA double-strand breaks, which can lead to recognition of the telomere by the DNA damage surveillance machinery[2]. When telomeres are falsely recognized as DNA damage, they are processed by the non-homologous end joining or homologous recombination pathways, leading to genome instability[3,4]. The end-replication problem arises from the difficulties encountered by the DNA replication machinery to extend the extremities of linear chromosomes, which results in telomere shortening with every cell division[5–7]. When a subset of telomeres shorten beyond a critical point, cellular senescence or apoptosis are triggered[8–10].

Specialized proteins have evolved to deal with the complications arising from telomeres, which in vertebrates are composed of double-stranded (ds) (TTAGGG)$_n$ repeats ending in a single-stranded (ss) 3' overhang[11]. In mammals, a telomere-interacting complex of six proteins termed shelterin constitutively binds to telomeres in mitotic cells[12]. This complex consists of the ds telomere binders TRF1 and TRF2, the TRF2-interacting protein RAP1, the ss binding protein POT1 and its direct interactor TPP1, as well as the bridging protein TIN2. Altogether, the proteins of this complex shield telomeres from a DNA damage response by inhibiting aberrant DNA damage signaling[3]. In addition, shelterin components are required for the recruitment of the telomerase enzyme, which adds de novo repeats to the telomeric ends, allowing maintenance of telomere length in dividing cells[6]. Telomerase is a ribonucleoprotein, comprised of a catalytic reverse-transcriptase protein component and an RNA moiety. Besides the core shelterin complex, additional proteins have been described to interact with telomeres and assist in the maintenance of telomere length, e.g., HMBOX1 (also known as HOT1), ZBTB48 (also known as TZAP), NR2C2, and ZNF827[13–17].

In *Schizosaccharomyces pombe*, a shelterin-like complex harboring orthologs of the human shelterin complex was described[18–20]. TAZ1 and POT1 bind to ds and ss telomeric DNA similar to their human counterparts TRF1/TRF2 and POT1, respectively. In turn, *Saccharomyces cerevisiae* has distinct complexes binding to the ds and ss telomere[21–26]. The *S. cerevisiae* ortholog of the TRF2-interacting protein RAP1 binds ds telomeric DNA through two domains structurally related to Myb domains[27]. The ss overhang is not bound by a POT1 homolog but rather by the CST complex[22,23,25]. Overall, this indicates that different telomere-binding complexes have evolved across species to alleviate the challenges of linear chromosome ends, based on variations of recurring DNA-binding modules.

The nematode *Caenorhabditis elegans* has been employed in many seminal discoveries in molecular biology, genetics, and development[28]. Its telomeres have a repeat sequence similar to vertebrate telomeres, consisting of (TTAGGC)$_n$[29]. Moreover, *C. elegans* telomeres have a length of about 2–9 kb[29,30], and it has been proposed that its telomeric structures have both 5' and 3' ss overhangs, each recognized by dedicated ss telomere-binding proteins[31]. Telomere maintenance in this nematode is carried out by the catalytic subunit of telomerase TRT-1[32]. The RNA component of *C. elegans* telomerase has not been identified thus far. Telomeres can be maintained by additional mechanisms, since *C. elegans* can survive without a functioning telomerase pathway by employing alternative lengthening of telomere (ALT)-like mechanisms, creating more heterogeneous telomere lengths[33–37].

In *C. elegans*, four proteins with domains structurally similar to the DNA-binding domain of human POT1 were identified. Three of those proteins, namely POT-1 (also known as CeOB2), POT-2 (also known as CeOB1), and MRT-1, were confirmed to bind to the ss telomeric overhangs[31,38]. Mutants for these factors show telomere length maintenance defects. Depletion of POT-1 and POT-2 leads to telomere elongation[31,33,35,37], whereas depletion of MRT-1 results in progressive telomere shortening over several generations[38]. Concomitant to telomere shortening, *mrt-1*, *mrt-2*, and *trt-1* mutant animals share a Mortal Germline (Mrt) phenotype, characterized by a gradual decrease in fertility across generations, until animals become sterile[30,32,38]. MRT-1 was proposed to be in a pathway for facilitation of telomere elongation together with the DNA damage checkpoint protein MRT-2, and telomerase TRT-1[38]. Despite the identification of these different telomere-associated proteins, no telomere-binding complex has been described in *C. elegans* yet.

In this work, we performed a quantitative proteomics screen to identify novel telomere-binding proteins in *C. elegans*. We report the identification and characterization of R06A4.2 and T12E12.3, two previously uncharacterized paralog genes, which we named telomere-binding proteins 1 and 2 (*tebp-1* and *tebp-2*), respectively. TEBP-1 and TEBP-2 bind to the ds telomeric sequence in vitro with nanomolar affinity and co-localize with POT-1, a known telomere binder, in vivo. *tebp-1* and *tebp-2* mutants have contrasting effects on telomere length: while *tebp-1* mutants display elongated telomeres, *tebp-2* mutants have shortened telomeres. In addition, TEBP-1 and TEBP-2 have important roles in fertility, as *tebp-1; tebp-2* double mutants are synthetic sterile. Size-exclusion chromatography and interaction studies demonstrate that TEBP-1 and TEBP-2 are part of a complex with POT-1, which bridges the ds telomere binders, TEBP-1 and TEBP-2, with the ss telomere binders POT-2 and MRT-1.

## Results

**TEBP-1 (R06A4.2) and TEBP-2 (T12E12.3) are double-stranded telomere-binding proteins in *Caenorhabditis elegans*.** To identify proteins that bind to the *C. elegans* telomeric sequence, we employed a DNA pulldown assay (Supplementary Fig. 1a, b) previously used to successfully identify telomeric proteins in other species[15,16,39,40]. We incubated concatenated, biotinylated DNA oligonucleotides consisting of either the telomeric sequence of *C. elegans* (TTAGGC$_n$), or a control sequence (AGGTCA$_n$), with nuclear-enriched extracts of gravid adult worms. The experiment was performed twice using two different quantitative proteomics approaches: label-free quantitation (LFQ)[41] and reductive dimethyl labeling (DML)[42], which yielded 12 and 8 proteins enriched in telomeric sequence pulldowns, respectively, with an overlap of 8 proteins (Fig. 1a, b and Supplementary Fig. 1a, b). Among these eight proteins, we found the already known ss telomere binders POT-1, POT-2, and MRT-1[31,33,37,38], as well as the CKU-70/CKU-80 heterodimer[43], and three additional proteins: R06A4.2, T12E12.3, and DVE-1.

R06A4.2 and T12E12.3 were of particular interest, as they share 74.3% DNA coding sequence identity and 65.4% amino acid sequence identity (Supplementary Fig. 1c), suggesting that R06A4.2 and T12E12.3 are paralogs. While R06A4.2 and T12E12.3 lack any annotated protein domain, using HHpred v3.2.0[44], we could determine that the N-terminal region of both proteins shows similarity to the homeodomains of human and yeast RAP1 (Supplementary Fig. 1d, e and Supplementary Data file 1). RAP1 is a direct ds telomere binder in budding yeast[21,45], and a member of the mammalian shelterin complex through interaction with TRF2[46].

We validated binding of R06A4.2 and T12E12.3 to telomeric DNA by performing DNA pulldowns with His-tagged recombinant proteins (Fig. 1c). Using CRISPR-Cas9 genome editing, we

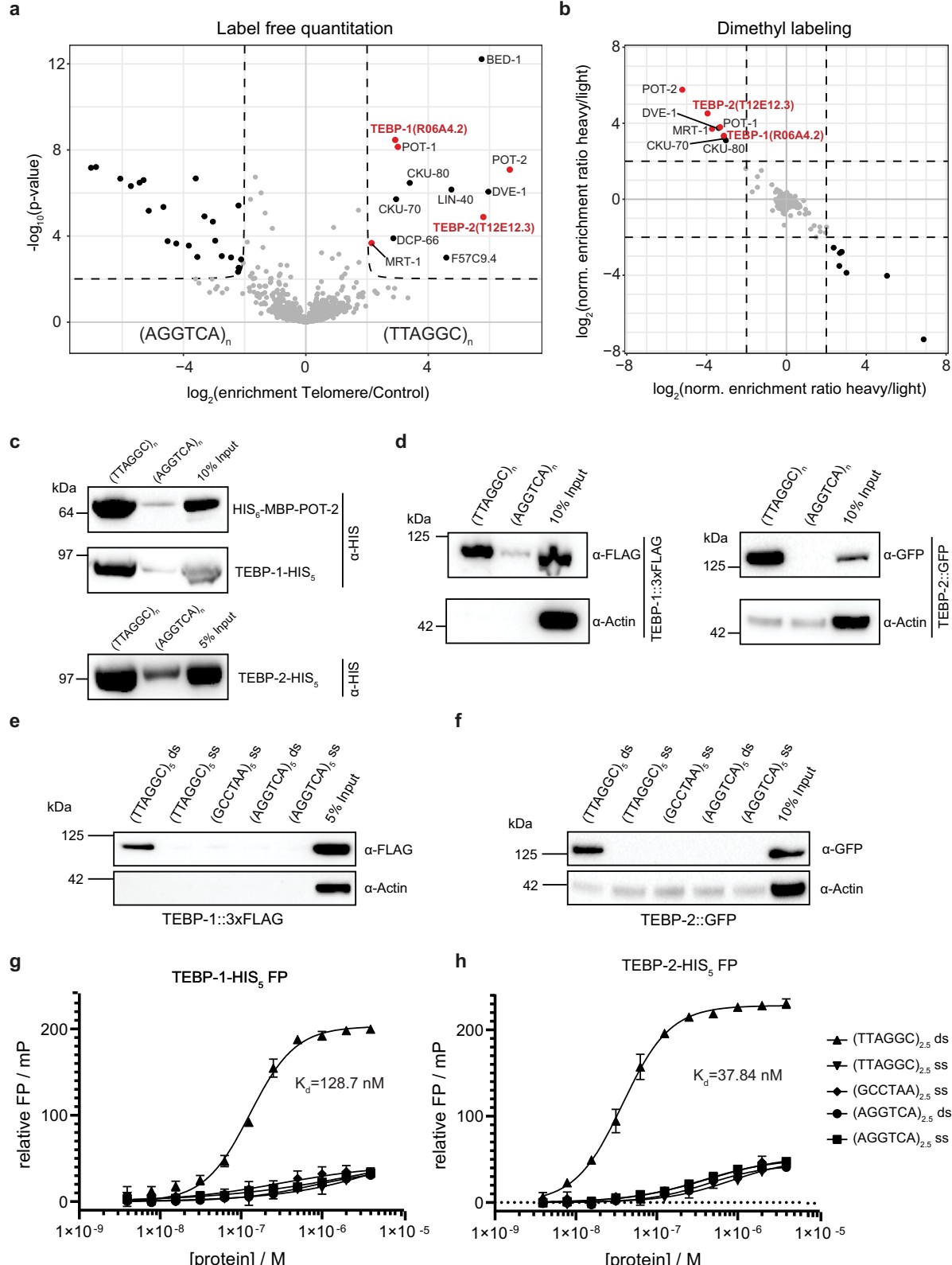

inserted a *gfp* and a *3xflag* sequence directly upstream of the endogenous stop codon of *T12E12.3* and *R06A4.2*, respectively (Supplementary Fig. 1d, e). Using these strains, we could show that the endogenously tagged versions of R06A4.2 and T12E12.3 also bind to the *C. elegans* telomere sequence (Fig. 1d).

Owing to the preparation strategy, our concatenated DNA probes contained both ds and ss DNA, which precludes any

conclusions about whether R06A4.2 and T12E12.3 bind ss or ds telomeric DNA. We thus performed additional DNA pulldowns with ss and ds probes specifically designed with five repeats (TTAGGC)$_5$. Both proteins were found to exclusively bind to the ds telomeric repeats, establishing R06A4.2 and T12E12.3 as ds telomere binders (Fig. 1e, f). To confirm and quantify the interaction of R06A4.2 and T12E12.3 with ds telomeric DNA, we

**Fig. 1 TEBP-1 (R06A4.2) and TEBP-2 (T12E12.3) are double-stranded telomere binders in *C. elegans*. a** Volcano plot representing label-free proteomic quantitation of pulldowns with biotinylated, concatenated oligonucleotide baits of telomeric DNA sequence (TTAGGC)$_n$ or control DNA sequence (AGGTCA)$_n$. Pulldowns were performed with nuclear extracts from synchronized gravid adult animals, in octuplicates per condition (two biological replicates, each with four technical replicates). Log$_2$ fold enrichment of proteins in one condition over the other is presented on the *x*-axis. The *y*-axis shows −log$_{10}$ *p*-value (Welch *t*-test) of enrichment across replicates. More than 4-fold enriched proteins with *p*-value < 0.01 are annotated as black dots, the background proteins as gray dots. Enriched proteins of interest, such as the known ss telomere binders, are annotated as red dots. **b** Scatterplot representing results of reductive dimethyl-labeling-based quantitation of pulldowns with the same extract and DNA baits as in (**a**). Per condition, pulldowns were performed in duplicates and labeled on the peptide level, including an intra-experimental label switch to achieve cross-over sets. The *x*-axis represents log$_2$ transformed ratios of the reverse experiment, whereas the *y*-axis represents log$_2$ transformed ratios of the forward experiment (see Supplementary Fig. 1b). Single proteins are depicted by dots in the scatterplot. Enriched proteins (threshold > 4) are annotated as black dots, background proteins as gray dots, and enriched proteins of interest as red dots. **c** Binding of recombinant His-tagged POT-2, TEBP-1 and TEBP-2, from crude *E. coli* lysate, to telomere or control DNA as in (**a**). Chemiluminescence western blot read-out, after probing with α-His antibody. POT-2 is used as a positive control for telomeric repeat binding. MBP: Maltose-binding protein, kDa: kilodalton. Uncropped blots in Source Data. *N* = 2 biologically independent experiments with similar results, except POT-2 *N* = 1. **d** DNA pulldowns as in **c** but on embryo extracts of transgenic *C. elegans* lines carrying either TEBP-1::3xFLAG or TEBP-2::GFP. *N* = 2 independent experiments with similar results. **e, f** DNA pulldowns with 5x telomeric (TTAGGC) double-strand (ds) repeats and both respective single-strand (ss) baits, and 5x control (AGGTCA) ds or 5x (AGGTCA) ss repeats. Pulldowns were performed with embryo extracts of TEBP-1::3xFLAG or TEBP-2::GFP animals. Uncropped blots in Source Data. *N* = 3 biologically independent experiments with similar results. **g, h** Fluorescence polarization assays of 4 μM to 4 nM purified TEBP-1-His$_5$ and TEBP-2-His$_5$, respectively. Binding affinities to 2.5x ss and ds telomeric and control repeats of FITC-labeled oligonucleotides. Error bars represent+/- the standard deviation of the mean values. Per data point *n* = 3 technical replicates. FP, fluorescence polarization; mP, millipolarization, upward triangle: 2.5x TTAGGC double-strand, downward triangle: 2.5x TTAGGC single-strand, diamond: 2.5x GCCTAA single-strand, circle: 2.5x shuffled control double-strand, square: 2.5x shuffled control single-strand.

performed fluorescence polarization with purified, recombinant proteins and FITC-labeled oligonucleotides. Both T12E12.3 and R06A4.2 displayed affinity for the ds telomeric repeat sequence in the nanomolar range ($K_d$ = 128.7 nM for R06A4.2 and $K_d$ = 37.84 nM for T12E12.3, Fig. 1g, h). Both T12E12.3 and R06A4.2 showed highest affinity for the 2.5x telomeric repeat, when incubated with a 2.5x, 2.0x, 1.5x T-rich, and 1.5x G-rich telomeric repeat sequences (Supplementary Fig. S2a–c).

In conclusion, we demonstrate that R06A4.2 and T12E12.3, two proteins with highly similar sequence, bind directly and with high affinity to the *C. elegans* ds telomeric DNA sequence in vitro. Thus, we decided to name R06A4.2 as Telomere-Binding Protein-1 (TEBP-1) and T12E12.3 as Telomere-Binding Protein-2 (TEBP-2).

**TEBP-1 and TEBP-2 localize to telomeres in proliferating cells in vivo**. To explore the expression pattern of *tebp-1* and *tebp-2* throughout animal development, we used a recently published mRNA-seq dataset[47]. Both genes show the highest expression in embryos, very low abundance during the L1–L3 larval stages, and an increase in expression in L4 larvae and young adults (YAs, Supplementary Fig. 3a–c). The observed increase in *tebp-1* and *tebp-2* mRNA expression from the L4 to YA stages coincides with the increased progression of germline development, which may hint to a higher expression level during gametogenesis. Indeed, using available gonad-specific RNA-seq datasets[48], we confirmed that *tebp-1 and tebp-2* are expressed in spermatogenic and oogenic gonads (Supplementary Fig. 3d). Similar developmental mRNA expression patterns were also found for the known ss telomere binders *pot-1*, *pot-2*, and *mrt-1* (Supplementary Fig. 3a, d). To study the expression at the protein level, we crossed our endogenously tagged strains to generate a *tebp-1::3xflag*; *tebp-2::gfp* strain to monitor protein abundance simultaneously by western blot. The protein expression patterns of TEBP-1 and TEBP-2 are highly similar to the RNA-seq data, with highest detected expression in embryos, a drop during the larval stages L1-L4, ultimately followed by an increase in YA (Fig. 2a).

To study TEBP-1 and TEBP-2 localization in vivo, we focused on embryos and on the germline of adult animals. In these two actively dividing tissues, TEBP-1 and TEBP-2 protein expression is high and condensed chromosomes facilitate visualization of telomeric co-localization. In addition to the *tebp-2::gfp* strain used above, we also generated an endogenously tagged *tebp-1::gfp*

allele, using CRISPR-Cas9 genome editing (Supplementary Fig. 1d). To check for telomeric localization in vivo, we crossed *tebp-1::gfp* and *tebp-2::gfp* each with a germline-specific *pot-1::mCherry* single-copy transgene[37], and imaged the dual-fluorescent animals. TEBP-1::GFP and TEBP-2::GFP co-localize with POT-1::mCherry inside the nuclei of oocytes and embryos (Fig. 2b–e). Confocal microscopy of TEBP-1::GFP in combination with POT-1::mCherry was challenging likely due to bleaching of TEBP-1::GFP. Co-localization of TEBP-2::GFP and POT-1::mCherry was also observed in the mitotic region of the germline and in mature sperm (Fig. 2d). These results clearly establish that TEBP-1 and TEBP-2 co-localize with a known telomeric binder in vivo in proliferating tissues, indicating that their ability to bind ds telomeric DNA in vitro may have functional relevance.

**TEBP-1 and TEBP-2 have opposing telomere length phenotypes**. As TEBP-1 and TEBP-2 localize to telomeres, we sought to address whether these proteins regulate telomere length, as is the case for the known ss telomere-binding proteins POT-1, POT-2, and MRT-1[31,33,37,38]. Using CRISPR-Cas9 genome editing, we generated *tebp-1* and *tebp-2* deletion mutants encoding truncated transcripts with premature stop codons (Supplementary Fig. 1d–g and Supplementary Fig. 4a, b). *tebp-1* and *tebp-2* mutants are viable and show no immediate, obvious morphological or behavioral defects. We analyzed telomere length in the mutants after propagation for more than 100 generations, sufficient to establish a "steady-state" telomere length phenotype, by carrying out a telomere Southern blot on mixed-stage animals. Interestingly, while *tebp-1(xf133)* shows an elongated telomere phenotype comparable to the *pot-2(tm1400)* mutant, *tebp-2(xf131)* shows a shortened telomere phenotype (Fig. 3a), similar to *mrt-1* mutants[38]. In addition, we performed quantitative fluorescence in situ hybridization (qFISH) in dissected adult germlines, which confirmed our initial observation that *tebp-1* and *tebp-2* mutants display longer or shorter telomeres than wild-type, respectively (Fig. 3b–f). Furthermore, we also measured telomere length in embryos by qFISH. Like in the germline, the telomeres of *tebp-1* mutant embryos are elongated, while the telomeres of *tebp-2* embryos are shortened (Supplementary Fig. 4c–g).

In summary, *tebp-1* and *tebp-2* mutants display opposing regulatory effects on telomere length. These experiments suggest that the TEBP-1 protein counteracts telomere elongation

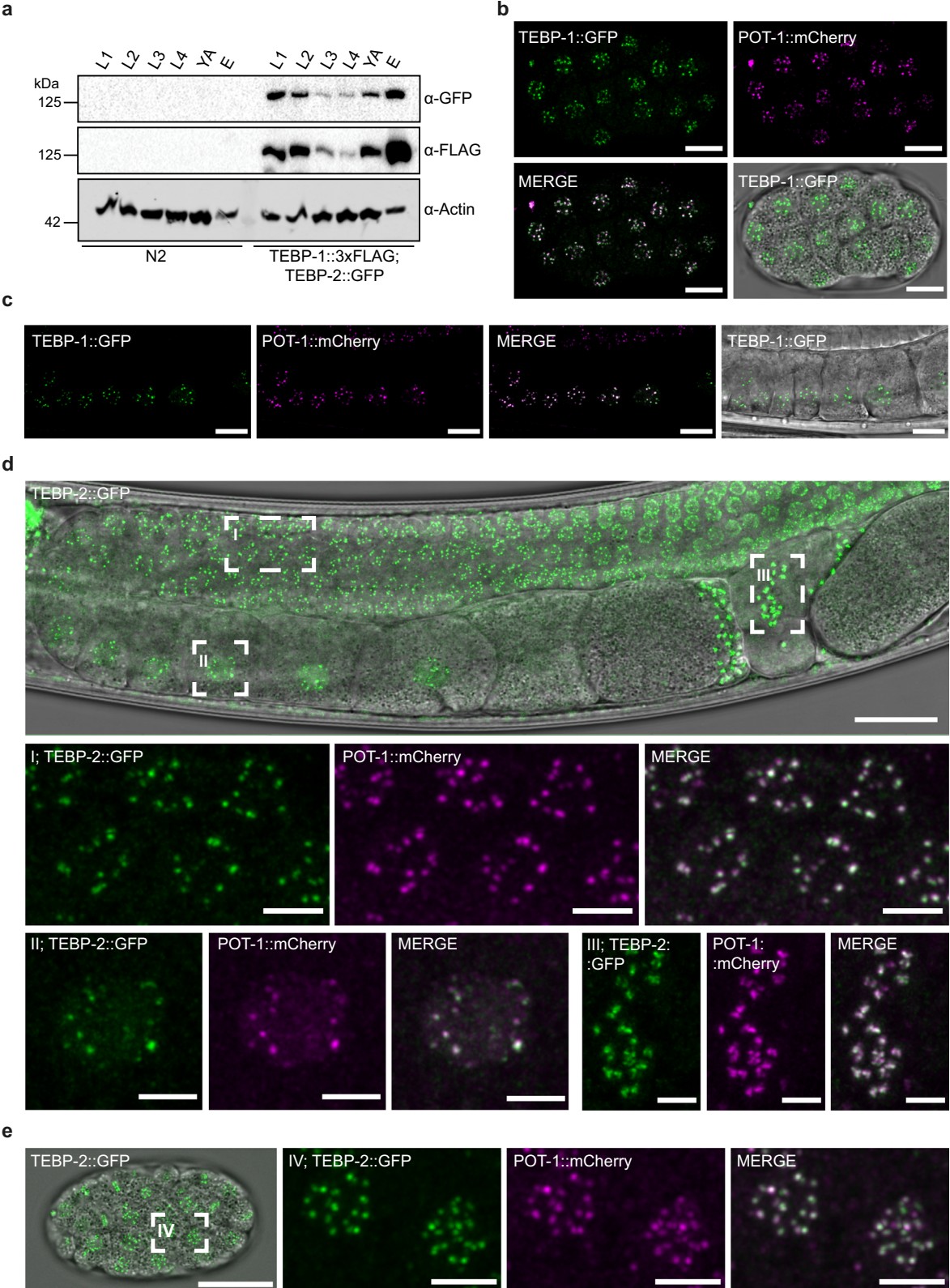

**Fig. 2 TEBP-1 and TEBP-2 are expressed throughout *C. elegans* development and localize to telomeres in vivo. a** Western blot of TEBP-1::3xFLAG and TEBP-2::GFP expression in different developmental stages of *C. elegans*. Thirty-five micrograms of extract from either N2 or a double transgenic line carrying TEBP-1::3xFLAG and TEBP-2::GFP were used. Actin was used as loading control. kDa: kilodalton. Uncropped blot in Source Data. *N* = 1 **b**, **c** Maximum intensity projections of representative confocal *z*-stacks of an embryo (**b**), or oocytes (**c**) expressing endogenously tagged TEBP-1::GFP and transgenic POT-1::mCherry. Scale bars, 10 μm. **d**, **e** Maximum intensity projections of representative confocal *z*-stacks of an adult animal (**d**), or embryo (**e**) expressing both endogenously tagged TEBP-2::GFP and transgenic POT-1::mCherry. Insets show nuclear co-localization in meiotic germ cell nuclei (I), an oocyte (II), spermatozoa (III), and embryonic cells (IV). Scale bars, 20 μm (overview) and 4 μm (insets). All microscopy images were deconvoluted using Huygens remote manager. Representative images from two individual animals per strain, *N* = 2 biologically independent experiments with similar results.

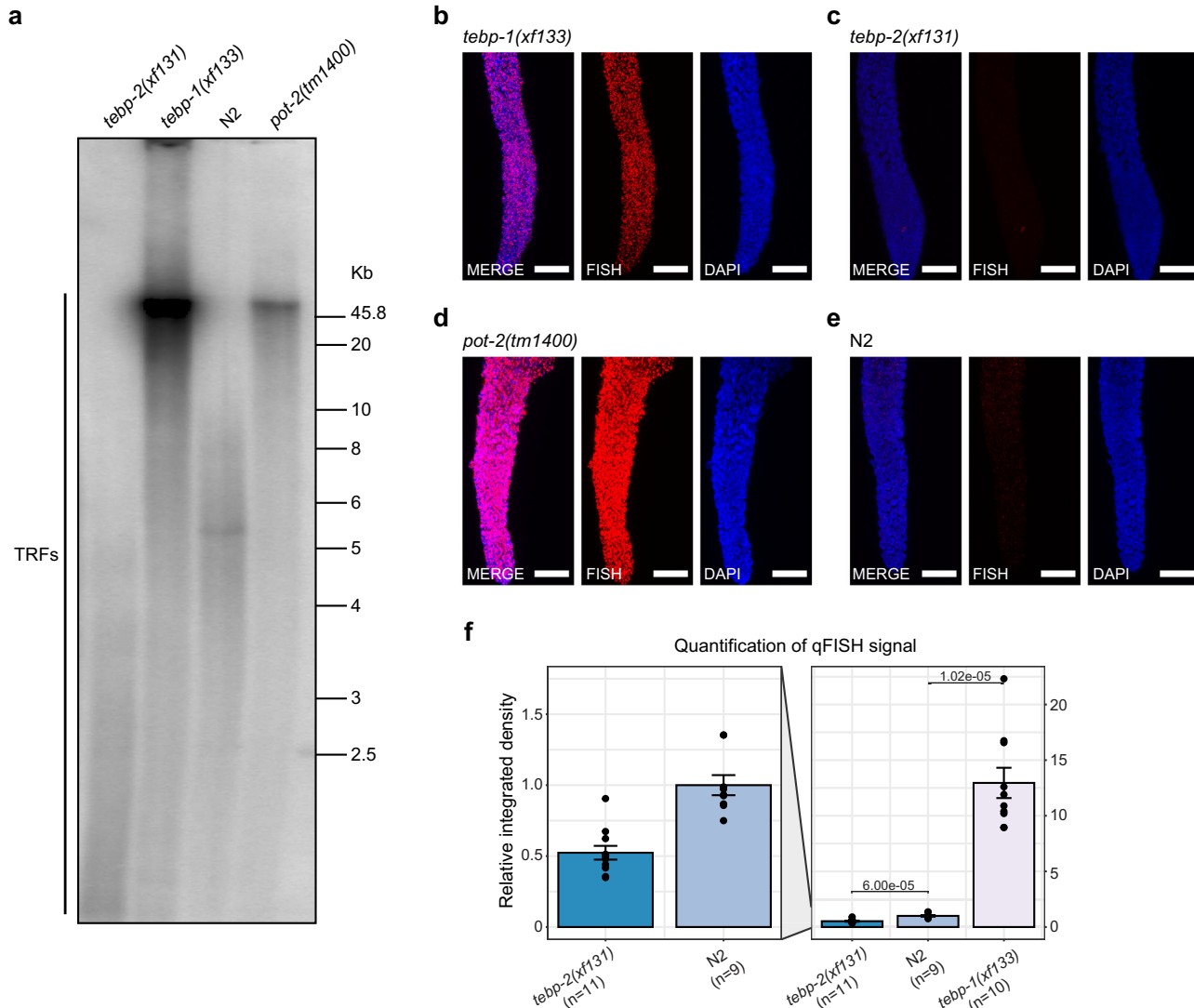

**Fig. 3 TEBP-1 and TEBP-2 regulate telomere length. a** Southern blot analysis of *C. elegans* telomeres. DNA from four different strains (*tebp-1(xf133)* grown for ~102 generations; *tebp-2(xf131)*, grown for ~124 generations; N2, and *pot-2(tm1400)*) was digested and separated by agarose gel electrophoresis. DNA was transferred to a positively charged nylon membrane and hybridized with a radiolabeled (GCCTAA)₃ oligonucleotide. Brightness and contrast of the membrane read-out were adjusted using Fiji. Telomere restriction fragments (TRFs) are indicated in the Fig.. Uncropped blot in Source Data. $N = 3$ independent experiments with similar results. **b–e** Representative maximum projection z-stacks of a qFISH assay using dissected adult germlines of the following *C. elegans* mutant strains: *tebp-1(xf133)* (grown for ~98 generations), *tebp-2(xf131)* (grown for ~120 generations), *pot-2(tm1400)*, and wild-type N2. The telomeres of dissected worms of the respective strains were visualized by hydridization with a telomeric PNA-FISH-probe. Nuclei were stained with DAPI. Scale bars, 15 μm. **f** Barplot depicting analysis of qFISH images of the strains in (**b–c**) and (**e**). Average telomere length is indicated by arbitrary units of relative integrated density, with wild-type N2 set to 1. The plot on the left shows the *tebp-2(xf131)* and N2 values zoomed-in. Analyzed n per strain derived from independent animals: *tebp-2(xf131)*: $n = 11$, N2: $n = 9$, *tebp-1(xf133)*: $n = 10$. Error bars represent the standard error of the mean (SEM) and *p*-values were calculated using Welch's *t*-test. $N = 3$ biologically independent experiments with similar results.

independently of telomerase, while TEBP-2 promotes telomere lengthening.

**Simultaneous lack of TEBP-1 and TEBP-2 leads to synthetic sterility.** To better understand how *tebp-1* and *tebp-2* mutants distinctly affect telomere length, we intended to measure telomere length in *tebp-1; tebp-2* double mutants. Surprisingly, when we crossed our single mutants, we could not establish a double homozygous *tebp-1; tebp-2* mutant strain. In fact, *tebp-1; tebp-2* double mutants displayed highly penetrant synthetic sterility (Fig. 4a). Repeating the cross with another *tebp-1* mutant allele (*xf134*), as well as the reciprocal cross, yielded the same synthetic sterility (Fig. 4b and Supplementary Fig. 5a). Only about 14–38%

of F2 or F3 *tebp-1; tebp-2* animals did not have synthetic sterility (Fig. 4a, b). These "synthetic sterility escapers" were subfertile, siring less than 60 offspring. Importantly, a *tebp-2::gfp* single-copy transgene fully rescued the appearance of sterility, demonstrating that the C-terminal tag does not disrupt TEBP-2 function (Supplementary Fig. 5a). When we combined *tebp-1* mutant animals with *mrt-1*, *trt-1*, or *pot-2* mutations, or *tebp-2* mutant animals with *trt-1* or *pot-2*, the double mutant offspring was fertile (Supplementary Fig. 5a). These results demonstrate that the synthetic sterility is specific to *tebp-1; tebp-2* double mutants, and is not a consequence of crossing shorter telomere mutants with longer telomere mutants. We further quantified the synthetic sterility on brood size by picking L2-L3 progeny of *tebp-2; tebp-1 +/−* mutants, blind to genotype and germline health, rearing those

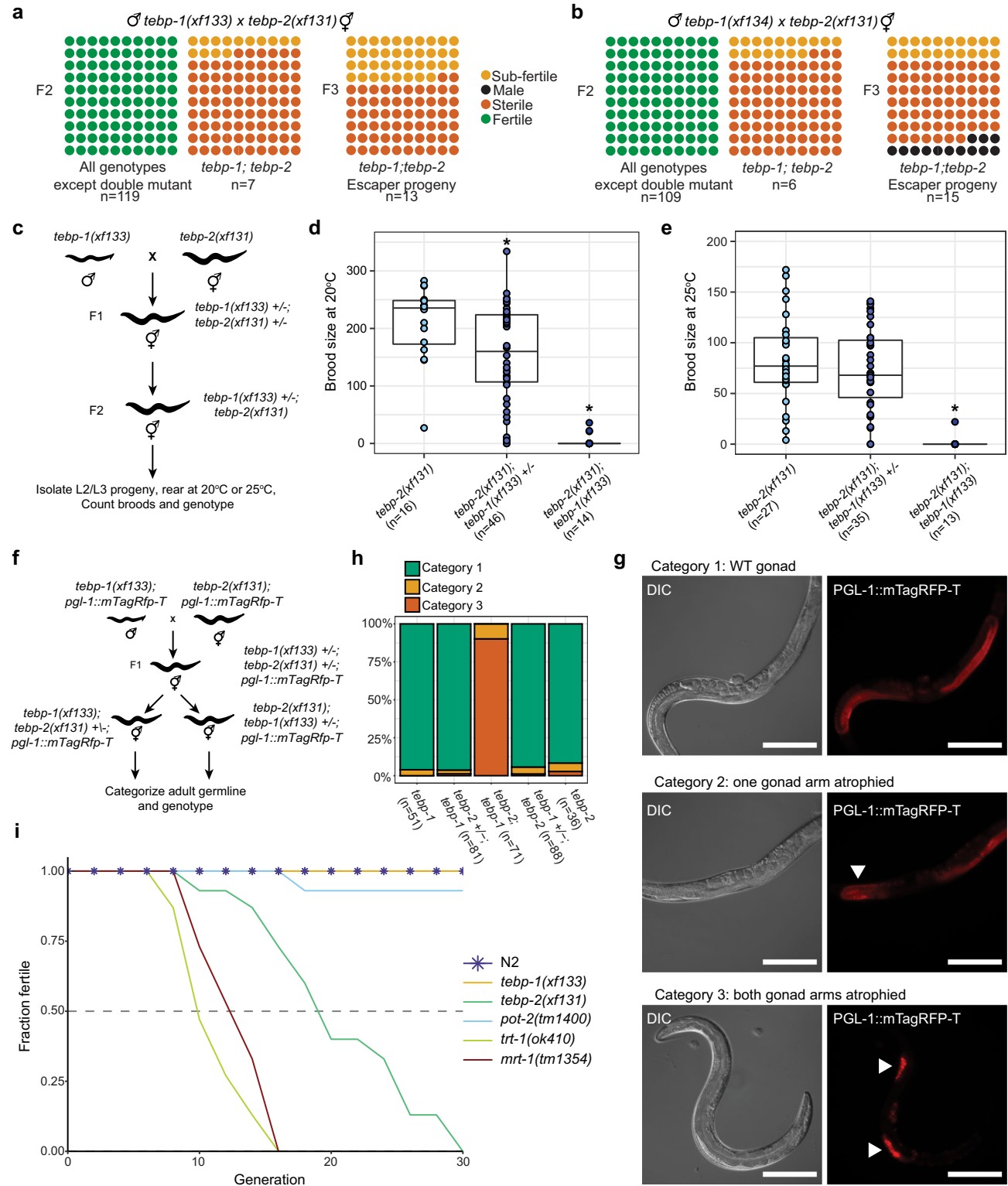

animals at 20 °C or 25 °C, later counting their brood sizes, and genotyping each animal (Fig. 4c–e). This revealed that the immediate synthetic sterility phenotype is not dependent on temperature, as the reduction of progeny numbers was apparent at both 20 and 25 °C.

Morphologically, *tebp-1; tebp-2* double mutants displayed a degenerated germline. To visualize this phenotype, we created *tebp-1* and *tebp-2* strains in combination with an endogenously tagged *pgl-1::mTagRfp-T* allele[49,50], which we used as a germ

cell reporter. PGL-1 is expressed in P-granules, perinuclear granules most important for germline development and gene regulation[51,52]. As depicted in Fig. 4f, we repeated the *tebp-1* x *tebp-2* cross with *pgl-1::mTagRfp-T* in the background, isolated cross progeny of the indicated genotypes, reared these animals to adulthood, scored them into three categories of germline morphology, and genotyped them afterwards. The categories can be described as follows: category 1 animals displayed a wild-type or near wild-type morphology (Fig. 4g, upper panels),

**Fig. 4 tebp-1; tebp-2 double mutants have synthetic sterility, and tebp-2 mutants have a Mortal Germline. a, b** Schematics depicting the quantification of fertility of the F2 (two panels on the left) and F3 (panel on the right) cross progeny of the indicated crosses. Each dot represents 1% of the indicated n per square, in a 10 × 10 matrix for 100%. Green dots indicate fertile worms, yellow dots subfertile worms (<60 progeny), orange dots sterile worms, and black dots indicate male worms. The F3 animals used for the panels on the right were the progeny of subfertile F2s, which escaped synthetic sterility. Males with two different tebp-1 mutant alleles, xf133 and xf134, were used in (**a**) and (**b**), respectively. **c** Schematic of cross performed with tebp-1(xf133) and tebp-2(xf131) to isolate progeny for determination of brood size at 20 and 25 °C. **d, e** Brood sizes of cross progeny animals, isolated as indicated in (**c**), which were grown at 20 °C (**d**), or 25 °C (**e**). Central horizontal lines represent the median, the bottom and top of the box represent the 25th and 75th percentile, respectively. Whiskers represent the 5th and 95th percentile, dots represent the data points used to calculate the box plot. n is indicated on the x-axis label. In (**d**), asterisks indicate the p-values of 9.6e-03 and 2.5e-06, as assessed by two-sided, unpaired Mann–Whitney and Wilcoxon tests comparing tebp-1 worms with the cross siblings of the other genotypes. In (**e**), asterisk indicates p-value = 4.1e-07, computed as in (**d**). **f** Schematic of a repetition of the double mutant cross as in (**c**) with pgl-1::mTagRfp-T in the background. Worms heterozygous for one of the tebp mutations were singled and their germline categorized at day 2–3 of adulthood, according to germline morphology and assessed by PGL-1::mTagRFP-T expression. Worms were genotyped after categorization and imaging. **g** Representative widefield differential interference contrast (DIC) and fluorescence pictures of the three germline morphology categories defined. Scale bars, 200 μm. Atrophied germlines in categories 2 and 3 are marked with a white arrowhead. **h** Barplot representing the quantification of each category, per genotype as indicated on the x-axis. Number of animals analyzed is shown in the x-axis labels. **i** Plot showing the fraction of fertile populations of each indicated genotype across successive generations grown at 25 °C. n = 15 populations per strain.

category 2 animals displayed one atrophied gonad arm (Fig. 4g, middle panels), and category 3 animals had both gonad arms atrophied (Fig. 4g, lower panels). Besides Fig. 4g, representative animals for categories 2 and 3 are shown in Supplementary Fig. 5b. More than 85% of tebp-1; tebp-2; pgl-1::mTagRfp-T worms had a category 3 germline, while the remainder had only one gonad arm atrophied (Fig. 4h). Atrophied gonads generally showed under-proliferation of the germ cell nuclei of the mitotic zone and rare entry into meiosis, suggesting severe defects in cell division (Fig. 4g and Supplementary Fig. 5b). In addition, almost 15% (17/114 animals) of the progeny of tebp-1; tebp-2; pgl-1::mTagRfp-T synthetic sterility escapers were males, indicative of a high incidence of males (Him) phenotype. The synthetic sterility escaper progenies of previous crosses were also Him, at least in some cases (see F3 escaper progeny in Fig. 4b). Lastly, approximately 8% (8/97) of hermaphrodite tebp-1; tebp-2; pgl-1::mTagRfp-T escaper progeny had growth defects: while some reached adulthood but remained smaller than wild-type, others arrested prior to adulthood (Supplementary Fig. 5c).

Overall, these data show that the lack of functional TEBP-1 and TEBP-2 leads to severe germline defects that impede germline development.

**TEBP-2 is required for transgenerational fertility.** Despite the synthetic sterility of the double mutants, tebp-1 and tebp-2 single mutants did not have a baseline reduction in fertility when grown at 20 and 25 °C (Supplementary Fig. 5d, e). Nevertheless, mutants of telomere regulators, like trt-1 and mrt-1, exhibit a Mrt phenotype, characterized by progressive loss of fertility across many generations[32,38]. We thus conducted a Mortal Germline assay at 25 °C using late generation mutants, and found that tebp-1 and tebp-2 mutants displayed opposing phenotypes in line with their differing effects on telomere length. While tebp-1(xf133) remained fertile across generations, like wild-type, tebp-2(xf131) showed a Mrt phenotype (Fig. 4i), the onset of which is delayed compared to mrt-1(tm1354) and trt-1(ok410), indicating a slower deterioration of germline health over generations. These results show that TEBP-2 is required to maintain germline homeostasis transgenerationally, while TEBP-1 is not.

**TEBP-1 and TEBP-2 are part of a telomeric complex in C. elegans.** Our initial mass spectrometry approach allowed us to identify proteins associated with the telomeres of C. elegans. However, it remains unknown if these factors interact and whether they are part of a telomere-binding complex. To address this,

we performed size-exclusion chromatography with embryonic extracts from a strain expressing TEBP-1::3xFLAG; TEBP-2::GFP. Western blot analysis of the eluted fractions shows that TEBP-1 and TEBP-2 have very similar elution patterns with one peak ranging from 450 kDa to 1.5 MDa, with a maximum at 1.1 MDa (Fig. 5a and Supplementary Fig. 6a). Next, we reasoned that the elution peak would shift if telomeric DNA is enzymatically degraded. To test this, embryonic extracts were treated with Serratia marcescens nuclease (Sm nuclease), a non-sequence-specific nuclease, prior to size-exclusion chromatography, but we did not observe a strong shift (Fig. 5b). While we cannot fully exclude the possibility that telomeric DNA was inaccessible to Sm nuclease digestion, the results suggest that TEBP-1 and TEBP-2 are part of a telomeric complex.

To identify proteins interacting with TEBP-1 and TEBP-2, we performed immunoprecipitation (IP) followed by quantitative mass spectrometry (qMS) in embryos (Fig. 5c, d) and YAs (Supplementary Fig. 6b, c). Notably, IP-qMS of TEBP-1 and TEBP-2 baits enriched for MRT-1, POT-1, and POT-2, the three known ss telomere-binding proteins in C. elegans. In some cases, (Fig. 5d and Supplementary Fig. 6b) it was difficult to unambiguously assign unique peptides to TEBP-1::3xFLAG and TEBP-2::GFP in our qMS analysis, given their high protein sequence identity (65.4%). However, we confirmed by co-IP experiments that TEBP-1 and TEBP-2 reciprocally interact in embryos and YA (Fig. 5e, f and Supplementary Fig. 6d). Moreover, TEBP-1 and TEBP-2 remain associated with MRT-1, POT-1, and POT-2 even after treatment with Sm nuclease (Supplementary Fig. 6e, f).

**POT-1 is required to bridge the double-stranded and the single-stranded telomere.** To reveal the architecture of the telomeric complex, we sought to identify direct interactions amongst TEBP-1, TEBP-2, POT-1, POT-2, and MRT-1, using a yeast two-hybrid (Y2H) screen. While TEBP-2 fused to the DNA-binding domain of Gal4 unfortunately self-activated the reporter (Supplementary Fig. 6g), we could identify direct interactions of POT-1 with TEBP-1 and TEBP-2 (Fig. 6a and Supplementary Fig. 6g). Furthermore, in accordance with IP-qMS and co-IP experiments (Fig. 5e, f and Supplementary Fig. 6d), we confirmed interaction between TEBP-1 and TEBP-2 in the Y2H experiment (Fig. 6a and Supplementary Fig. 6g). These results are consistent with a scenario where TEBP-1 and TEBP-2 interact directly with each other and with POT-1.

The observed direct interactions suggest that POT-1 may be a critical link between the ds and the ss telomeric region. To test this idea, we performed IP-qMS of TEBP-1 and TEBP-2, in

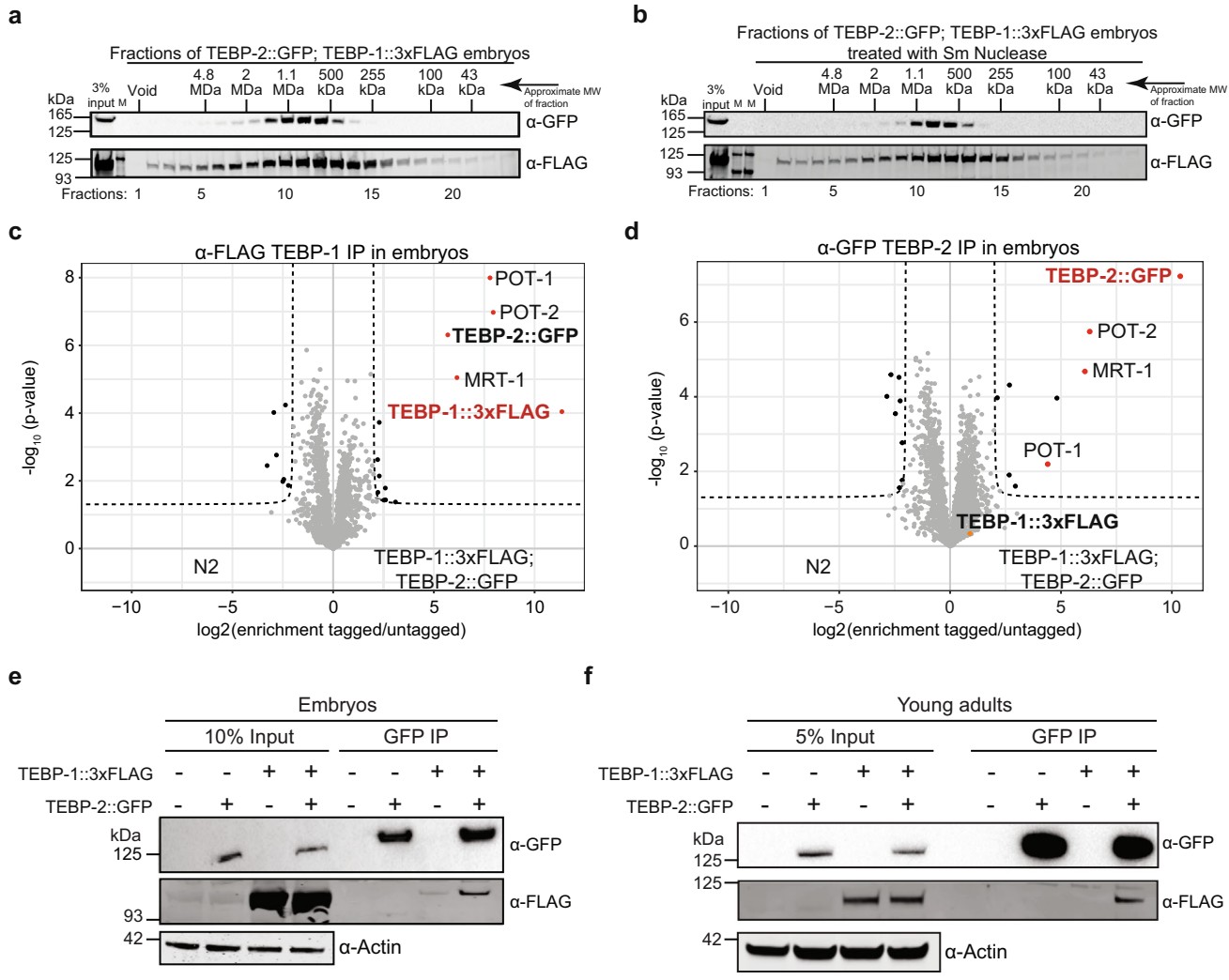

**Fig. 5 TEBP-1 and TEBP-2 are part of a telomeric protein complex. a** Size-exclusion chromatography of embryo extracts expressing TEBP-1::3xFLAG and TEBP-2::GFP, followed by western blot of the eluted fractions. The approximate molecular weight (MW) of the fractions is indicated on the Fig. panel. $N = 2$ biologically independent experiments with similar results. **b** Identical to (**a**), but with treatment of embryo extracts with Sm nuclease, prior to size-exclusion chromatography. $N = 1$. **c, d** Volcano plots showing quantitative proteomic analysis of either TEBP-1::3xFLAG (**c**) or TEBP-2::GFP (**d**) IPs in embryos. IPs were performed in quadruplicates. Enriched proteins (threshold: 4-fold, $p$-value < 0.05) are shown as black dots, enriched proteins of interest are highlighted with red or orange dots, and the baits are named in red. Background proteins are depicted as gray dots. **e** Co-IP western blot experiment of TEBP-1::3xFLAG and TEBP-2::GFP. The IP was performed with a GFP-trap, on embryo extracts from strains carrying either one or both of the endogenous tags and wild-type. Actin was used as loading control. **f** Same co-IP experiment as in (**e**) but carried out with extracts from young adult worms. For (**e**) and (**f**) $N = 3$ biologically independent experiments with similar results.

wild-type and mutant *pot-1* backgrounds. These experiments showed that interaction of the ds telomere binders TEBP-1 and TEBP-2 with the ss binders POT-2 and MRT-1, is strongly depleted in *pot-1* mutants (Fig. 6b, c). TEBP-1 and TEBP-2 protein levels are not affected by the *pot-1* mutation, indicating the loss of interaction with POT-2 and MRT-1 is not due to reduced availability of TEBP-1 or TEBP-2 (Supplementary Fig. 6h). In addition, TEBP-1 and TEBP-2 still interact with each other in the absence of POT-1 (Supplementary Fig. 6h).

Next, to map the amino acid sequences responsible for TEBP-1 and TEBP-2 DNA-binding and protein-protein interactions, with each other and with POT-1, we divided their protein sequences into seven fragments (f1–f7), and the protein sequence of POT-1 into three fragments (f1–f3, Fig. 6d). DNA pulldowns with His-MBP-tagged TEBP-1 and TEBP-2 recombinant proteins demonstrated DNA binding by their f3 fragments (Fig. 6d, e), which contain their third predicted homeo-/myb-domain. Furthermore, Y2H experiments using the fragments shown in Fig. 6d, indicate

that the C-terminal tails of TEBP-1 and TEBP-2 (f7) interact with the OB-fold of POT-1 (Fig. 6f, g). Additional Y2H assays demonstrate that TEBP-1 and TEBP-2 interact with each other via their respective f1 fragments, encompassing their first predicted homeo-/myb-domains (Fig. 6h and Supplementary Fig. 6i).

Altogether, our data strongly indicate that TEBP-1 and TEBP-2 are integral parts of a telomeric complex, or complexes, which also include the known ss telomere binders POT-1, POT-2, and MRT-1. We propose a simple working model where TEBP-1 and TEBP-2 bind to the ds telomere via their third predicted homeo-/myb-domains, have opposed effects on telomere dynamics, and are required for fertility (Fig. 6i). POT-1, with the ability of its OB-fold to directly bind the C-terminal tails of TEBP-1 and TEBP-2 (Fig. 6a, f, g), as well as ss telomeric repeats in vitro[31], may link the ds binders to the ss telomere, thereby bringing TEBP-1 and TEBP-2 in close proximity of POT-2 and MRT-1 (Fig. 6i).

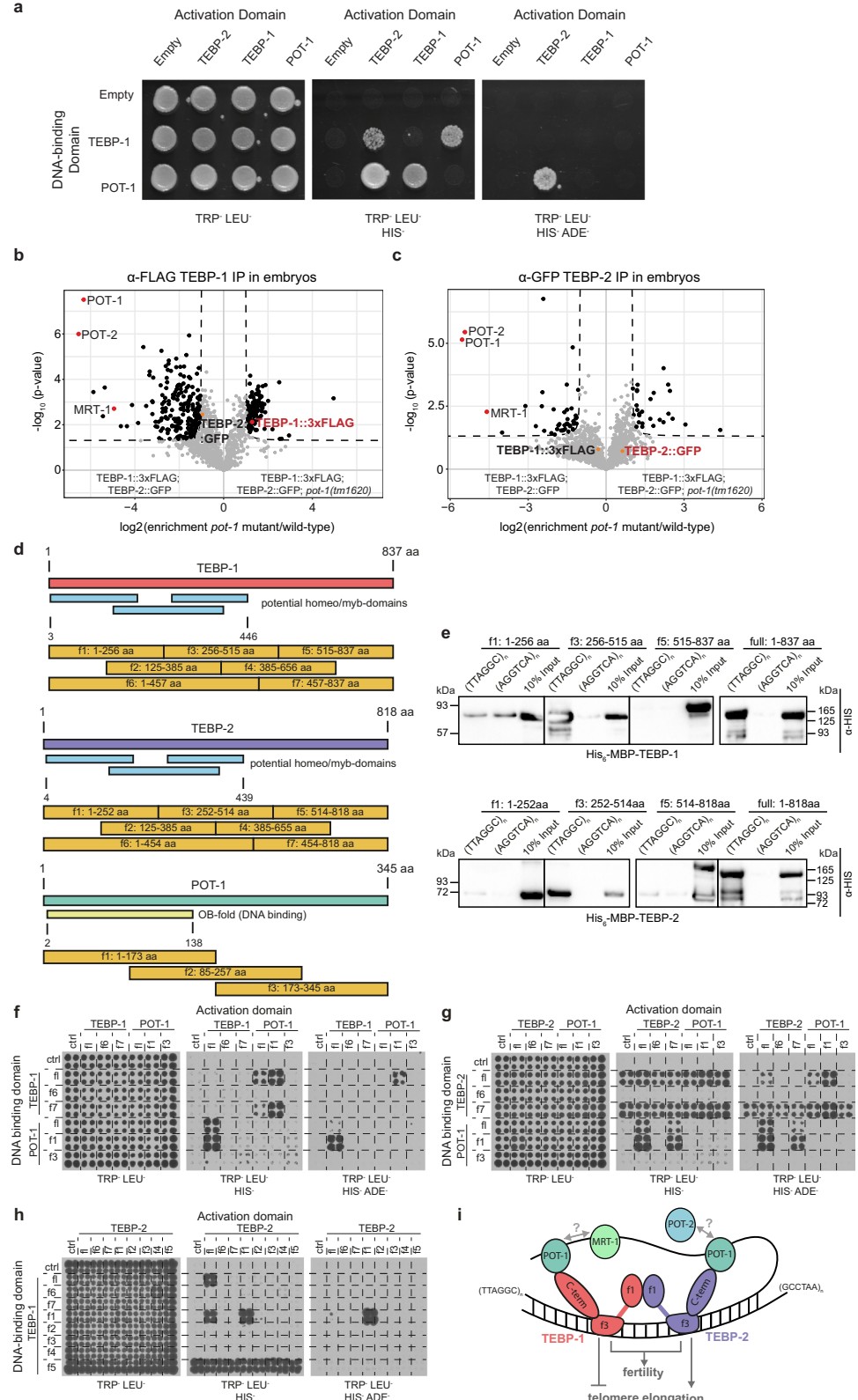

**Conservation of *tebp* genes in the *Caenorhabditis* genus**. To infer the evolutionary history of *tebp-1* and *tebp-2* genes, we identified protein-coding orthologs by reciprocal BLASTP analysis in the searchable genomes in Wormbase and Wormbase ParaSite databases. Then, we performed a multiple sequence alignment with the ortholog protein sequences, and used it to build a phylogenetic tree (Fig. 7a and Supplementary Data file 2).

Our findings suggest that *tebp* orthologs are present only in the *Caenorhabditis* genus, mostly in the *Elegans* supergroup (which includes the *Elegans* and *Japonica* groups). A distinct number of protein-coding *tebp* genes was identified per species: *C. briggsae*, *C. nigoni*, *C. sinica*, and *C. japonica* have one *tebp* ortholog; *C. elegans*, *C. inopinata*, *C. remanei*, *C. brenneri*, *C. tropicalis*, and *C. angaria* have two *tebp* orthologs; and *C. latens* has three *tebp*

**Fig. 6 POT-1 links the ds telomere binders to the ss telomere. a** Y2H assay with full length TEBP-1, TEBP-2, and POT-1 fusions to the activation or DNA-binding domains of Gal4. Growth on TRP⁻ LEU⁻ HIS⁻ plates demonstrates interaction. Growth on high stringency TRP⁻ LEU⁻ HIS⁻ ADE⁻ medium suggests strong interaction. TRP:⁻ lacking tryptophan, LEU:⁻ lacking leucine, HIS:⁻ lacking histidine, ADE:⁻ lacking adenine. **b, c** Volcano plots showing quantitative proteomic analysis of either TEBP-1::3xFLAG (**b**) or TEBP-2::GFP (**c**) IPs in embryos. IPs were performed in quadruplicates. Enriched proteins (threshold: 2-fold, $p$-value < 0.05) are shown as black dots, enriched proteins of interest are highlighted with red or orange dots, and annotated. Background proteins are depicted as gray dots and the respective bait protein annotated in red. **d** Scheme for the cloning of different fragments of TEBP-1, TEBP-2 and POT-1 for IP experiments and Y2H. TEBP-1 and TEBP-2 were divided into five fragments (f1–f5) of approx. 30 kDa, as well as two additional fragments covering the N-terminus including the predicted DNA-binding domains (f6) and the C-terminus (f7). POT-1 was divided into three fragments of around 15 kDa (f1–f3). **e** DNA pulldowns as in Fig. 1c with recombinantly expressed and N-terminally His-MBP-tagged fragments f1, f3, and f5 of TEBP-1 and TEBP-2, as well as the full length proteins with the same tags. The western blot was probed with α-His antibody and the signals detected by chemiluminescence. f1–f5: fragments of respective protein, full: full length respective protein, kDa: kilodalton, MBP: maltose-binding protein. $N = 2$ independent experiments with similar results. **f** Y2H assay like in (**a**) but with TEBP-1 and POT-1 full length proteins (fl), as well as N- and C-terminal fragments (f6 and f7 for TEBP-1, or f1 and f3 for POT-1, respectively) fused to the activation or DNA-binding domains of Gal4. Growth determined on the same medium as in **a**. **g** Y2H assay as in (**f**) but with TEBP-2 and POT-1 constructs. **h** Y2H assay as in (**f**) but with all fragments of TEBP-1 including the full length protein fused to the Gal4 DNA-binding domains, as well as all fragments of TEBP-2 including the full length protein fused to the Gal4 activation domain. f1–f7: fragments of respective protein, crtl: control/empty plasmid, fl: full length protein. **i** Proposed working model for the interactions between telomere-binding proteins and telomere repeats in *C. elegans*. TEBP-1 and TEBP-2 fragments 3 (f3), containing a predicted DNA-binding domain, bind to ds telomere repeats and have opposing effects on telomere elongation. Both proteins interact with each other via their N-terminal fragments (f1). TEBP-1, TEBP-2 and POT-1 interact directly via the C-terminal fragment (f7) of TEBP-1/TEBP-2 and the N-terminal fragment (f1) of POT-1. As a result of this interaction, the ss telomere comes in closer contact to the ds telomere. Our current data does not support direct interactions between POT-1, POT-2, and MRT-1, but these factors may interact in the presence of telomeric DNA.

---

orthologs. The multiple sequence alignment showed the N-terminal region of *tebp* genes, the region with similarity to the homeodomains of human and yeast RAP1 (Supplementary Fig. 1d, e and Supplementary Data file 1), is more similar between orthologs than the C-terminal region (Supplementary Data File 2). However, phylogenetic analysis with only the N-terminal region did not produce major differences on tree topology (Supplementary Fig. 7). In order to derive evolutionary relationships between different *tebp* genes, we evaluated local synteny information. We found a high degree of regional synteny conservation between *C. elegans tebp-1* and one of the *tebp* copies in *C. inopinata*, *C. remanei*, *C. briggsae*, *C. nigoni*, *C. sinica*, *C. tropicalis*, and *C. japonica* (Table 1 and Supplementary Data file 2). Conversely, *tebp-2* did not show any signs of regional synteny across *Caenorhabditis* species (Supplementary Data file 2), suggesting that the gene duplication event creating *tebp-2* occurred after divergence from the *C. inopinata* lineage, less than 10.5 million years ago[53]. Neither of the two *tebp* orthologs of *C. brenneri*, *C. latens*, and *C. angaria* are in synteny with *C. elegans tebp-1* (Supplementary Data file 2).

To determine whether TEBP proteins are generally telomere-binders in the *Elegans* supergroup, we performed DNA pulldowns, using nuclear extracts prepared from synchronized *C. briggsae* gravid adults. CBG11106, the only *C. briggsae* ortholog of *tebp-1* and *tebp-2*, was significantly enriched in the telomere pulldown (Fig. 7b), demonstrating that it can bind to the TTAGGC telomeric repeat. Of note, CBG22248, one of the two *C. briggsae* orthologs of MRT-1, was also enriched in the telomere pulldown, and CBG16601, the ortholog of POT-1, was just below our significance threshold, suggesting functional similarities to their *C. elegans* orthologs.

## Discussion

Telomeres and their associated proteins are important to ensure proper cell division. In the popular model nematode *C. elegans*, only ss telomere-binding proteins were known thus far[31,38]. Here, we describe a telomeric complex with the paralogs TEBP-1 and TEBP-2 as direct ds telomere-binding proteins. POT-1 seems to bridge the ds telomere-binding module of the complex, comprised of TEBP-1 and TEBP-2, with the ss telomere region. Strikingly, despite the high level of sequence similarity between TEBP-1 and TEBP-2, their mutant phenotypes are divergent.

**Robust identification of telomere-associated proteins in *C. elegans*.** Three lines of evidence demonstrate the validity and robustness of our screen. First, attesting for its technical reproducibility, the two qMS detection strategies employed shared an overlapping set of proteins enriched in telomeric sequence pulldowns (8 overlapping factors out of 12 and 8 hits). Second, within our overlapping set of enriched factors, we detected the previously identified ss telomere-binding proteins POT-1, POT-2, and MRT-1[31,33,37,38]. Lastly, the *C. elegans* KU heterodimer homologs CKU-70 and CKU-80 were enriched in the screens. In other organisms, such as *Saccharomyces cerevisiae*, *Trypanosoma brucei*, *Drosophila melanogaster*, and *Homo sapiens*, KU proteins have been shown to associate with telomeres, regulating their length and protecting them from degradation and recombination[54,55]. The *C. elegans* homologs were shown to interact with telomeres, but do not seem to have telomere regulatory functions[43]. However, CKU-70 and CKU-80 were not enriched in the TEBP-1 and TEBP-2 interactome experiments, suggesting that their binding to telomeric DNA occurs independently of the TEBP-1/TEBP-2 complex (Fig. 5 and Supplementary Fig. 6). Alternatively, these factors may be part of the telomeric complex, with no direct interaction with TEBP-1 or TEBP-2.

We identified POT-3 in the background of our LFQ screen (Supplementary Data File 3), supporting the lack of telomeric phenotypes of *pot-3* mutants[31]. Furthermore, a number of factors previously reported to have telomere DNA-binding capability or to regulate telomere length, were not detected or lacked significant enrichment in our quantitative proteomics screen. MRT-2 is a homolog of *S. cerevisiae* checkpoint gene RAD17 and human RAD1, previously reported to regulate telomere length[30]. Much like *tebp-2* and *mrt-1*, *mrt-2* mutants have shorter telomeres than wild-type and a Mrt phenotype. It is plausible that MRT-2 regulates telomere length beyond the context of direct telomeric binding. PLP-1[56], HMG-5[57], and CEH-37[58], were previously shown to bind to the *C. elegans* telomeric sequence in vitro. PLP-1 was enriched in the $(AGGTCA)_n$ scrambled control in our qMS screen (Supplementary Data file 3), suggesting that PLP-1 is a general ds DNA binder, and not a specific telomere binder. Furthermore, HMG-5 was detected in the background, and CEH-37 was not detected altogether in our screen (Supplementary Data file 3). Further studies should clarify if and how these factors interact with the telomere complex described in this work.

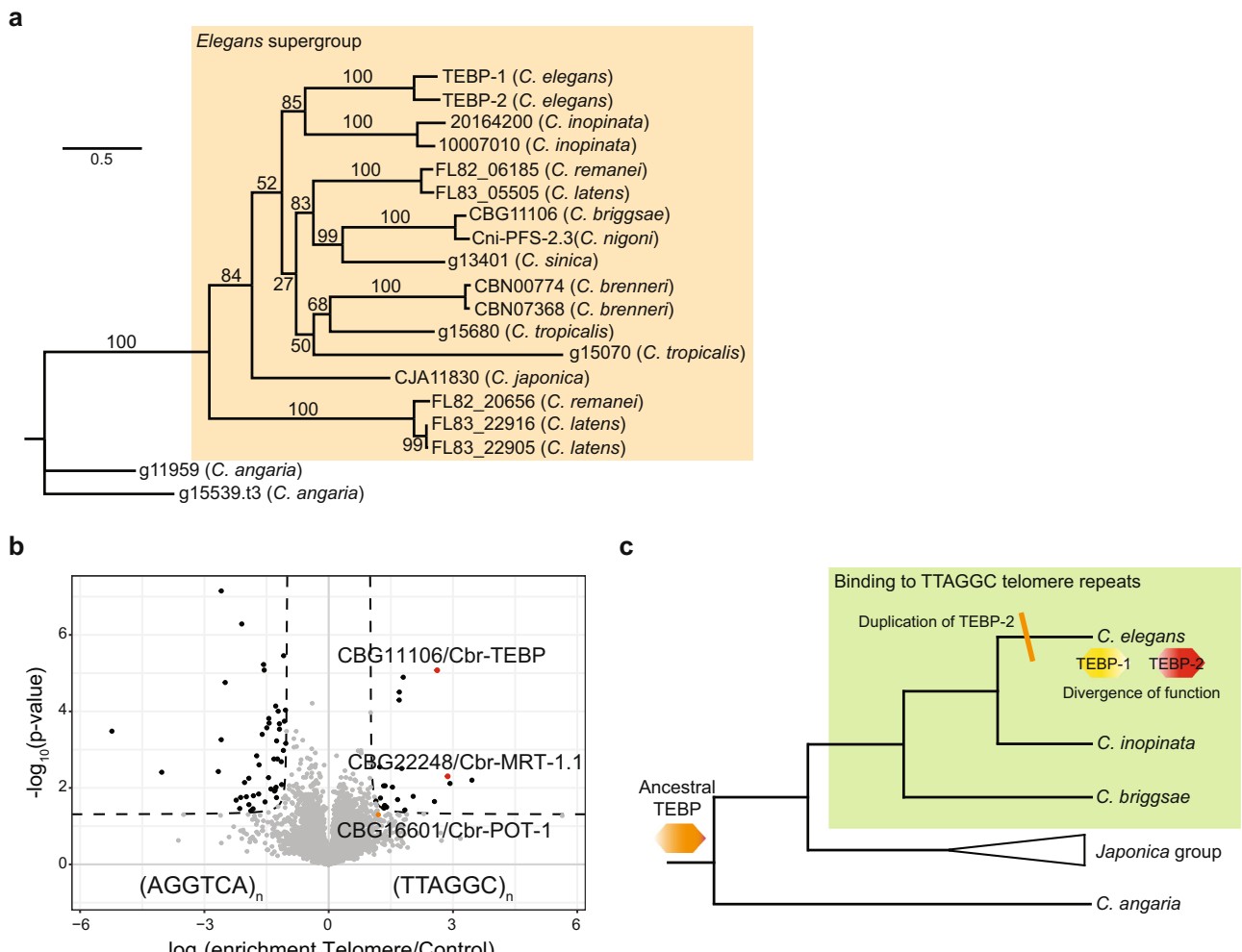

**Fig. 7 Conservation of *tebp* genes in the *Caenorhabditis* genus. a** Phylogenetic tree constructed with IQ-TREE (v1.6.12), using a MAFFT (v7.452) multiple sequence alignment of the protein sequences of TEBP orthologs (see Supplementary Data file 2, sheet 2). Values on the nodes represent bootstrapping values for 10,000 replicates, set to 100. The TEBP orthologs outside the orange background represent the outgroup of the analysis. **b** Volcano plot of telomere DNA pulldown, as in Fig. 1a, of gravid adult nuclear extracts from *C. briggsae*. Here, pulldowns were performed in quadruplicates, per condition. Enriched proteins (enrichment threshold > 2-fold, *p*-value < 0.05) are labeled as black dots, whereas enriched proteins of interest are labeled with red or orange dots. Proteins below the threshold are depicted as gray dots. Homologs of telomere binders are named. **c** Depiction of the evolution of *tebp* genes in *Caenorhabditis*. We speculate that this family originated from an ancestor TEBP (orange hexagon), presumably required for fertility and capable of binding to telomeres. As we have confirmed telomere binding in *C. elegans* and *C. briggsae* (species in bold indicate confirmed binding of TEBP proteins to telomeric DNA), it is plausible that their common ancestor was able to bind to telomeres. The gene duplication that generated *tebp-2* occurred after the divergence of *C. elegans* and *C. inopinata* (marked as orange stripe), followed by division, or diversification, of functions of these two paralogs (TEBP-1: yellow hexagon, TEBP-2: red hexagon).

**The fast-evolving paralogs TEBP-1 and TEBP-2 are required for fertility**. TEBP-1 and TEBP-2 share 65.4% of their amino acid sequence, which most likely reflects a common origin by gene duplication. Interestingly, the two paralogs TEBP-1 and TEBP-2 interact with each other, and with the same set of factors, i.e., POT-1, POT-2, and MRT-1 (Fig. 5 and Supplementary Fig. 6). This is striking, considering the divergent phenotypes of *tebp-1* and *tebp-2* mutants: *tebp-1* mutants have longer telomeres than wild-type, while *tebp-2* animals have shorter telomeres than wild-type and a Mortal Germline. Moreover, while the fertility of *tebp-1* and *tebp-2* animals is not compromised, *tebp-1*; *tebp-2* double mutants show highly penetrant synthetic sterility irrespective of the temperature the animals are grown at, indicating that TEBP-1 and TEBP-2 contribute to normal fertility (Fig. 4 and Supplementary Fig. 5). The observed synthetic sterility is likely justified by failure to enter and progress through normal mitosis and meiosis, as judged by the under-proliferation of germ cells.

The synthetic sterility of *tebp-1*; *tebp-2* animals is specific to these two paralogs, as other genetic crosses of shorter versus longer telomere mutants did not result in sterile double mutants. The synergistic role of TEBP-1 and TEBP-2 in fertility provide a puzzling contrast with their opposed telomere length mutant phenotypes. We speculate that the requirement of TEBP-1 and TEBP-2 to fertility may be independent of their functions at telomeres. Future studies on the influence of TEBP-1 and TEBP-2 on germline and embryonic gene expression may shed light on this aspect.

CBG11106, the single homolog of TEBP-1 and TEBP-2 in *C. briggsae*, interacts with telomeric DNA (Fig. 7b), suggesting that TEBP nematode homologs bind to telomeric DNA at least since the divergence of *C. elegans* and *C. briggsae*, from a common ancestor that presumably lived 80–100 million years ago[59]. To verify this, the capability of additional TEBP orthologs to bind to telomeric DNA needs to be experimentally addressed. We

**Table 1 Synteny analysis of *tebp* orthologs in other *Caenorhabditis* species.**

| *tebp* ortholog | Synteny with *tebp-1* | Synteny with *tebp-2* |
|---|---|---|
| 10007010 (*C. inopinata*) | – | – |
| 20164200 (*C. inopinata*) | + | – |
| FL82_06185 (*C. remanei*) | + | – |
| FL83_05505 (*C. latens*) | – | – |
| CBG11106 (*C. briggsae*) | + | – |
| Cni-PFS-2.3 (*C. nigoni*) | + | – |
| g13401 (*C. sinica*) | + | – |
| CBN00774 (*C. brenneri*) | – | – |
| CBN07368 (*C. brenneri*) | – | – |
| g15680 (*C. tropicalis*) | + | – |
| g15070 (*C. tropicalis*) | – | – |
| CJA11830 (*C. japonica*) | + | – |
| FL83_22916 (*C. latens*) | – | – |
| FL83_22905 (*C. latens*) | – | – |
| FL82_20656 (*C. remanei*) | – | – |
| g15539.t3 (*C. angaria*) | – | – |
| g11959 (*C. angaria*) | – | – |

Overview of synteny of the tebp orthologs of other *Caenorhabditis* species with tebp-1 or tebp-2 of *C. elegans*. A "+" indicates regional synteny, while a "−" is lack of synteny.

speculate that *tebp-1* and *tebp-2* originated from an ancestor *Caenorhabditis tebp* gene required for fertility and with the ability to bind ds telomeric repeats (Fig. 7c). The *tebp-1* ancestor was duplicated after the divergence of *C. inopinata* and *C. elegans*, 10.5 million years ago[53], likely initiating a process of functional diversification of *tebp-1* and *tebp-2*.

Given their possible recent divergence, in evolutionary terms the 65.4 % protein sequence similarity observed between the protein sequences of TEBP-1 and TEBP-2 is actually fairly low. This likely reflects fast evolution of TEBP-1 and TEBP-2, in line with the known fast evolution as suggested for other telomere-binding proteins[60]. While it is tempting to establish evolutionary relationships with vertebrate TRF1 and TRF2 proteins, TEBP-1/TEBP-2 and TRF1/TRF2 are not homologs. In addition, TRF1 and TRF2 are binding to telomeric DNA via C-terminal myb-domains[61], while DNA binding in TEBP-1 and TEBP-2 occurs N-terminally. However, on the functional level, similarity between *C. elegans* TEBP-1/TEBP-2 and vertebrate TRF1/TRF2, potentially reflecting convergent evolution between two phylogenetically independent sets of telomere-binding paralogs is possible, but needs further investigation.

**A telomere complex in actively dividing tissues in homeostasis.** Our size-exclusion chromatography, quantitative proteomics, and Y2H data support the existence of a telomere complex comprising TEBP-1, TEBP-2, POT-1, POT-2, and MRT-1 (Fig. 5 and Supplementary Fig. 6). According to our size-exclusion chromatography data, this complex elutes in a range between 600 kDa and 1.1 MDa. It should be noted that our model does not make any assumptions regarding complex stoichiometry. At the moment, we cannot exclude the existence of remaining DNA fragments in the complex, despite nuclease treatment, which could add to the total molecular weight. Thus, we propose a working model, whereby TEBP-1 and TEBP-2 bind to ds telomere repeats via their third predicted homeo-/myb-domains, and directly interact with the OB-fold of POT-1 with their C-terminal tails. Binding to POT-1 may, in turn, bring the ss telomeric repeats, and thus POT-2 and MRT-1, into closer contact (Fig. 6i). In the absence of POT-1, TEBP-1 and TEBP-2 are not able to interact with POT-2 and MRT-1 (Fig. 6b, c). We speculate that reciprocal regulation

by TEBP-2 and POT-1/TEBP-1 define normal telomere length. In this scenario, TEBP-2 might counteract telomere shortening by POT-1 and TEBP-1 (Fig. 6i). The precise interplay between these telomeric factors, namely the interactions between POT-1, POT-2, and MRT-1, and the mechanism of telomere elongation have to be further elucidated.

The mammalian shelterin complex counteracts recognition of telomeres as DNA double-strand breaks by inhibiting the DNA damage machinery. When shelterin factors are abrogated, catastrophic end-to-end chromosome fusions are observed[62,63]. Previous studies did not identify end-to-end chromosome fusions in *pot-1* and *pot-2* mutants[31,33,37]. It remains to be determined if *tebp-1* and/or *tebp-2* mutations lead to telomere fusions and whether the *C. elegans* telomeric complex is required to protect telomeres from DNA damage. It is possible that the synthetic sterility and high frequency of males observed in *tebp-1; tebp-2* double mutants, as well as the Mortal Germline phenotype of *tebp-2* and *mrt-1*, may be downstream of germline genome instability.

A germline-specific MAJIN/TERB1/TERB2 telomere-binding complex has been described in mouse testes[64–66]. Knock-outs of these factors lead to meiotic arrest and male sterility[64–66], similar to the observed phenotype in *tebp-1; tebp-2* double mutants. This mammalian protein complex tethers telomeres to the nuclear envelope, a process essential for meiotic progression. A previous study has shown that POT-1 is required in *C. elegans* to tether telomeres to the nuclear envelope during embryogenesis[67]. Given the interaction of TEBP-1 and TEBP-2 with POT-1 in vitro and in vivo, the telomeric complex may be dynamically involved in this process.

The distinct compartmentalization of post-mitotic soma versus actively dividing germline, together with a plethora of genetic tools, make *C. elegans* an enticing model organism for telomere biology in vivo, in homeostatic conditions. The identification of a telomeric complex in *C. elegans* allows further investigation of telomere regulation in this popular model organism.

## Methods

**C. elegans nuclear-enriched protein extract preparation.** Nuclear extract preparation of gravid adult worms was done as described[68]. The worms were synchronized by bleaching and harvested at the gravid adult stage by washing them off the plate with M9 buffer. After washing the worms in M9 buffer for 4 times, they were pelleted by centrifugation at 600 x *g* for 4 min, M9 buffer was removed and extraction buffer (40 mM NaCl, 20 mM MOPS pH 7.5, 90 mM KCl, 2 mM EDTA, 0.5 mM EGTA, 10% Glycerol, 2 mM DTT, and 1x complete protease inhibitors Roche) was added. Worms resuspended in extraction buffer were frozen in liquid nitrogen. The resulting pellets were ground to a fine powder in a pre-cooled mortar and transferred to a pre-cooled glass douncer. When thawed, the samples were sheared with 30 strokes, piston B. The worm suspension was pipetted to pre-cooled 1.5 ml reaction tubes (1 ml per tube) and cell debris, as well as unsheared worms were pelleted by centrifugation at 200 x *g* for 5 min at 4 °C for two times. To separate the cytoplasmatic and nuclear fractions, the supernatant was spun at 2000 x *g* for 5 min at 4 °C. The resulting pellet containing the nuclear fraction was washed twice by resuspension in extraction buffer and subsequent centrifugation at 2000 x *g* for 5 min at 4 °C. After the washing steps, the nuclear pellet was resuspended in 200 µl buffer C + (420 mM NaCl, 20 mM Hepes/KOH pH 7.9, 2 mM MgCl₂, 0.2 mM EDTA, 20% Glycerol, 0.1% Igepal CA 630, 0.5 mM DTT, 1x complete protease inhibitors). Nuclear extract of gravid adult worms of *C. briggsae* was prepared as described above.

**Oligonucleotides.** All oligonucleotides used throughout this manuscript (cloning, sequencing, DNA pulldowns, fluorescence polarization etc.) are listed in Supplementary Data file 4 with their name and sequence.

**DNA pulldowns**
*Preparation of biotinylated DNA for pulldown experiments.* Biotinylated telomeric and control DNA for the DNA pulldown for detection of telomeric interactors was prepared as previously published[16,39,40]. In short, 25 µl of 10-mer repeat oligonucleotides of either telomeric or control sequence were mixed 1:1 with 25 µl of their respective reverse complement oligonucleotide and 10 µl annealing buffer (200 mM Tris-HCl, pH 8.0, 100 mM MgCl2, 1 M KCl). The mixture was brought to

100 μl final volume with $H_2O$, heated at 80 °C for 5 min, and left to cool. Once at room temperature (RT), the samples were supplemented with 55 μl $H_2O$, 20 μl 10x T4 DNA ligase buffer (Thermo Scientific), 10 μl PEG 6000, 10 μl 100 mM ATP, 2 μl 1 M DTT and 5 μl T4 Polynucleotide Kinase (NEB, 10 U/μl, #M0201) and left at 37 °C for 2 h to concatenate. Finally, 4 μl of T4 DNA Ligase (Thermo Scientific, 5 WU/μl, #EL0011) were added and the samples incubated at RT overnight for ligation and polymerization. The ligation process was monitored by running 1 μl of the reaction on a 1% agarose gel. The samples were cleaned by phenol-chloroform extraction. For this, 1 vol. of $H_2O$ and 200 μl of Phenol/Chloroform/Isoamyl Alcohol (25:24:1; pH 8; Invitrogen, # 15593049) was added to the mixture, vortexed and centrifuged at 16,000 x g for 2 min. After centrifugation the aqueous phase was transferred to a fresh tube and the DNA precipitated by addition of 1 ml 100% Ethanol and incubation at −20 °C for 30 min. Afterwards the suspension was centrifuged at 16,000 x g for 45 min at 4 °C. The resulting DNA pellet was resuspended in 74 μl $H_2O$ and 10 μl 10x Klenow-fragment reaction buffer (Thermo Scientific), 10 μl 0.4 mM Biotin-7-dATP (Jena Bioscience, #NU-835-BIO) and 6 μl Klenow-Fragment exo- polymerase (Thermo Scientific, 5 U/μl, # EP0422) added. Biotinylation was carried out by incubation at 37 °C over night. The reaction was cleaned up by size-exclusion chromatography using MicroSpin Sephadex G-50 columns (GE Healthcare, #GE27-5330-01).

*Pulldown experiments.* Biotinylated DNA and Dynabeads™ MyOne™ Streptavidin C1 (Thermo Scientific, #65001) were mixed with PBB buffer (50 mM Tris/HCl pH 7.5, 150 mM NaCl, 0.5% NP 40, 5 mM $MgCl_2$, 1 mM DTT) and incubated at room temperature for 15 min on a rotating wheel to immobilize the DNA on the beads. After three washes with PBB buffer, the DNA coupled beads were resuspended in PBB buffer and Salmon sperm (10 mg/ml, Ambion, #AM9680) was added 1:1000 as competitor for unspecific DNA binding. The pulldowns were performed with different amounts of protein extract (see below) and incubated at 4 °C on a rotating wheel for 90 min. Following incubation the beads were washed three times with PBB buffer and resuspended in 1x Loading buffer (4x NuPAGE LDS sample buffer, Thermo Scientific, #NP0008) supplemented with 100 mM DTT. For elution, the samples were boiled at 70 °C for 10 min and afterwards loaded on a gel and processed as indicated above for MS, or below for western blot. In pulldown-MS experiments, the pulldowns were prepared in either technical quadruplicates (LFQ), or technical duplicates (DML) per condition, whereas for western blot all conditions were prepared with one replicate and an input. In all, 200–400 μg of nuclear worm extract and of *Escherichia coli* extract were used for the mass spectrometry screen and pulldowns of Fig. 1c, respectively. In all, 0.4–0.7 mg of total protein extract were used for the pulldowns shown in Fig. 1d–f. Four-hundred micrograms of *E. coli* extract was used in DNA-binding domain pulldowns in Fig. 6e.

## Mass spectrometry: sample preparation, data acquisition, and analysis

*In-gel digest.* In-gel digestion was performed as previously described[16,69] with the exception of the DML samples (see below). Samples were run on a 10% Bis-Tris gel (NuPAGE; Thermo Scientific, #NP0301) for 10 min (IP samples) or on a 4–12% Bis-Tris gel (NuPAGE, Thermo Scientific, #NP0321) for 20 min (LFQ-measured telomeric DNA pulldowns) at 180 V in 1x MOPS buffer (NuPAGE, Thermo Scientific, #NP0001). Individual lanes were excised and cut to approximately 1 mm × 1 mm pieces with a clean scalpel, and transferred to a 1.5 ml tube. For the LFQ telomeric DNA pulldowns, the lanes were split into four fractions. The gel pieces were destained in destaining buffer (50% 50 mM $NH_4HCO_3$ (ABC), 50% ethanol p.a.) at 37 °C under rigorous agitation. Next, gel pieces were dehydrated by incubation in 100% acetonitrile for 10 min at 25 °C shaking and ultimately dehydrated using a Concentrator Plus (Eppendorf, #5305000304, settings V-AQ). The gel pieces were incubated in reduction buffer (50 mM ABC, 10 mM DTT) at 56 °C for 60 min and subsequently incubated in alkylation buffer (50 mM ABC, 50 mM iodoacetamide) for 45 min at room temperature in the dark. Gel pieces were washed in digestion buffer (50 mM ABC) for 20 min at 25 °C. Next, gel pieces were dehydrated again by incubation in 100% acetonitrile and drying in the concentrator. The dried gel pieces were rehydrated in trypsin solution (50 mM ABC, 1 μg trypsin per sample, Sigma-Aldrich, #T6567) and incubated overnight at 37 °C. The supernatant was recovered and combined with additional fractions from treatment with extraction buffer (30% acetonitrile) twice and an additional step with pure acetonitrile for 15 min at 25 °C, shaking at 1400 rpm. The sample solution containing the tryptic peptides was reduced to 10% of the original volume in a Concentrator Plus, to remove the acetonitrile and purified using the stage tip protocol.

*Dimethyl labeling.* Dimethyl labeling (DML) was done as previously described[70]. For DML, in-gel digest was performed as indicated in the last section, with the exception of exchanging ABC buffer for 50 mM TEAB (Fluka, #17902) after alkylation. The volume of the extracted peptides was reduced in a Concentrator Plus. For labeling, either 4% formaldehyde solution (Sigma-Aldrich, #F8775) for light labeling or 4% formaldehyde-D2 (Sigma-Aldrich, #596388) solution for medium labeling, as well as 0.6 M NaBH3CN (Sigma-Aldrich, #156159) were added to the samples and mixed briefly. The mixture was incubated for 1 h at 20 °C, shaking at 1000 rpm and afterwards quenched by addition of a 1% ammonia solution (Sigma-Aldrich, #30501) and acidified with 10% formic acid solution (Merck, #1.00264.1000). After the labeling reaction, the respective light and

medium samples were mixed 1:1 (light telomere: medium control; medium telomere: light control) and purified by stage tip purification.

*Stage tip purification.* Stage tip purification was performed as previously described[71]. Desalting tips were prepared by using two layers of Empore C18 material (3 M, #15334911) stacked in a 200 μl pipet tip. The tips were activated with pure methanol. After two consecutive washes with Buffer B (80% acetonitrile, 0.1% formic acid) and Buffer A (0.1% formic acid) for 5 min the tryptic peptide samples were applied and washed once more with Buffer A. Upon usage, peptides were eluted with Buffer B. The samples were centrifuged in a Concentrator Plus for 10 min to evaporate the acetonitrile and adjusted to 14 μl with Buffer A.

*MS measurement and data analysis.* For MS measurement 5 μl of sample were injected. The desalted and eluted peptides were loaded on an in-house packed C18 column (New Objective, 25 cm long, 75 μm inner diameter) for reverse-phase chromatography. The EASY-nLC 1000 system (Thermo Scientific) was mounted to a Q Exactive Plus mass spectrometer (Thermo Scientific) and peptides were eluted from the column in an optimized 2 h (pulldown) gradient from 2 to 40% of 80% MS grade acetonitrile/0.1% formic acid solution at a flow rate of 225 nL/min. The mass spectrometer was used in a data-dependent acquisition mode with one MS full scan and up to ten MS/MS scans using HCD fragmentation. All raw files were processed with MaxQuant (version 1.5.2.8) and searched against the *C. elegans* Wormbase protein database (Version WS269), as well as the Ensembl Bacteria *E. coli* REL606 database (version from September 2018) for proteins from the feeding strain OP50. Carbamidomethylation (Cys) was set as fixed modification, while oxidation (Met) and protein N-acetylation were considered as variable modifications. For enzyme specificity, trypsin was selected with a maximum of two miscleavages. LFQ quantification (without fast LFQ) using at least 2 LFQ ratio counts and the match between run option were activated in the MaxQuant software. Fractions and conditions were indicated according to each experiment. Data analysis was performed in R using existing libraries (ggplot2-v 3.2.1, ggrepel-v 0.8.1, stats-v 3.5.2) and in-house scripts. Protein groups reported by MaxQuant were filtered removing known contaminants, protein groups only identified by site and those marked as reverse hits. Missing values were imputed at the lower end of LFQ values using random values from a beta distribution fitted at 0.2–2.5%. For statistical analysis, *p*-values were calculated using Welch's *t*-test. Enrichment values in the volcano plots represent the mean difference of log2 transformed and imputed LFQ intensities between the telomere and the control enriched proteins. Peptide labels created by the dimethyl-labeling reaction were selected in the MaxQuant software as "N-terminal Dimethyl 0" and "Dimethyl 0" for the light samples, as well as "N-terminal Dimethyl 4" and "Dimethyl 4" for the heavy labeled samples. The re-quant option was activated. An incorporation check was run additionally to confirm incorporation of the dimethyl labels of at least 95% in each sample. Protein groups resulting from MaxQuant analysis were filtered identically to LFQ. The normalized ratios for each protein were log2 transformed and plotted in the scatterplot. Filtering and analysis were done in R using existing libraries and an in-house script.

## In vitro single- or double-strand binding of proteins from *C. elegans* extract.

For this assay, biotinylated oligonucleotides (Metabion) were used, containing a five times repeat of telomeric G-rich, C-rich, or control sequences. To allow for proper annealing, all oligonucleotides contained unique sequences flanking both sides of the repeats. Double-stranded oligonucleotides were prepared by mixing the biotinylated forward oligonucleotide 1:1 with the respective non-biotinylated reverse complement oligonucleotide and addition of annealing buffer (200 mM Tris-HCl, pH 8.0, 100 mM $MgCl_2$, 1 M KCl). The mix was heated at 80 °C for 5 min and cooled to room temperature. The single-stranded oligonucleotides were treated similarly, only replacing the reverse compliment oligonucleotide with $H_2O$. The pulldown itself was performed as described above with 0.5 mg (TEBP-2::GFP) or 0.4 mg (TEBP-1::3xFLAG) *C. elegans* embryo total protein extract of the respective strains. After elution, the samples were run on a 4–12% Bis-Tris gel (NuPAGE, Thermo Scientific, #NP0321) at 150 V for 120 min and transferred to a membrane. Western blot detection of the tagged proteins was carried out as described below.

## Expression and purification of recombinant protein from *E. coli*. Auto-induction[72] was used for expression of His6-MBP-POT-2. An overnight culture of the expression strain BL21(DE3) was cultured at 37 °C in YG medium (2% Yeast extract, 0.5% NaCl, 3.5% Glycerol) supplemented with the respective antibiotic. A growing culture in YG medium was prepared by inoculating it with 1:50 volume of the overnight culture. At an $OD_{600}$ of 0.7, a culture of auto-induction medium (2% Peptone, 3% Yeast extract, 25 mM $Na_2HPO_4$/$KH_2PO_4$, 0.05% Glucose, 2.2% Lactose, 0.5% Glycerin, 50 mM $NH_4Cl$, 5 mM $Na_2SO_4$, 2 mM $MgSO_4$, 1x Trace Metal Solution) was inoculated with the growing culture to a density of $OD_{600}$ 0.004. 1000x Trace Metal Solution used for the auto-induction medium, has the following constitution: of 50 mM $FeCl_3$/HCl, 20 mM $CaCl_2$, 10 mM Mn(II)$Cl_2$, 10 mM $ZnCl_2$, 2 mM $CoCl_2$, 2 mM Cu(II)$Cl_2$, 2 mM $NiCl_2$, 2 mM $NaMoO_4$, 2 mM $Na_2SeO_3$. The auto-induction culture was incubated at 25 °C for 24 h and then harvested by centrifugation at 4000 x g.

TEBP-1-His$_5$ and TEBP-2-His$_5$ were expressed in Rosetta 2 (DE3) pLysS competent cells (Novagen,#71401). An overnight culture was grown in LB containing the respective antibiotic. A growing culture was inoculated and after reaching mid-log growth at 37 °C, the cultures were induced with 1 mM IPTG. Cells were grown at 18 °C and harvested after 24 h. IPTG-induced or auto-induction cultures were pelleted in 50 ml reaction tubes by centrifugation at 4000 x g after growth and lysed according to the protocol for the respective downstream use.

POT-2 expression pellets were resuspended in Tris buffer (50 mM Tris/HCL pH 7.5, 100 mM NaCl, 10 mM MgCl$_2$, 1x EDTA-free protease inhibitor (Roche, #4693132001)) and divided into 2 ml flat lid micro tubes containing 0.1 mm zirconia beads (Carl Roth, #N033.1). Lysis of the cells was achieved with a FastPrep-24™ Classic (MP Biomedicals, #116004500) using the setting 6 m/s for 30 s for two times. In between the disruption cycles the samples were centrifuged at 21,000 x g for 2 min to pellet debris, followed by an incubation on ice for 5 min before the second cycle. After lysis the suspension was centrifuged at 21,000 x g for 10 min at 4 °C.

TEBP-1 and TEBP-2 expression pellets were lysed via sonication with a Branson Sonifier 450 (duty cycle: 50%, output control: 3, 3.5 min with 5 mm tip) in lysis buffer (25 mM Tris-HCl pH 7.5, 300 mM NaCl, 20 mM imidazole) with 1 mM DTT, and protease inhibitor cocktail tablets (Roche, #4693132001). Lysates were centrifuged at 4613 x g for 10 min at 4 °C. For both preparation methods the supernatant was afterwards transferred to fresh reaction tubes.

His-MBP tagged TEBP-1 and TEBP-2 fragments were expressed in E.coli ArcticExpress DE3 cells (Agilent, #230192). Cultures were grown overnight in 5 ml LB supplemented with the respective antibiotic for the expression vector. Next day the expression culture was inoculated from the overnight culture and grown to mid-log phase at 30 °C, and then induced with 1 mM IPTG. Cultures were incubated at 12 °C and harvested after 24 h. The pellet was resuspended in binding buffer (20 mM Tris-HCl pH 7.5, 500 mM NaCl, 50 mM imidazole) with 1 mM DTT, complete protease inhibitor cocktail tablets (Roche, #4693132001), and 100 µg DNase I (NEB, M0303S). Cells were lysed using a Branson Sonifier (duty cycle: 50%, output control: 4, 6 min (3 min sonication, 3 min ice, 3 min sonication) with 9 mm tip). Lysates were cleared at 4613 x g for 10 min at 4 °C, and used for subsequent assays.

**Protein expression, purification, and fluorescence polarization assay**. E.coli ArcticExpress DE3 cells (Agilent, #230192) were grown overnight in 5 ml LB supplemented with the respective antibiotic for the expression vector. Next day the expression culture was inoculated from the overnight culture and grown to mid-log phase at 30 °C, and then induced with 1 mM IPTG. Cultures were incubated at 12 °C and harvested after 24 h. The pellet was resuspended in binding buffer (20 mM Tris-HCl pH 7.5, 500 mM NaCl, 50 mM imidazole) with 1 mM DTT, complete protease inhibitor cocktail tablets (Roche, #4693132001), and 100 µg DNase I (NEB, M0303S). Cells were lysed using a Branson Sonifier (duty cycle: 50%, output control: 4, 6 min (3 min sonication, 3 min ice, 3 min sonication) with 9 mm tip). Lysates were ultracentrifuged (Beckman Optima XE-100) at 75,000 x g for 30 min at 4 °C. After loading the lysate, the HisTrap HP column (GE Healthcare, #GE17-5247-01) was washed with binding buffer, and proteins were eluted in binding buffer containing 500 mM imidazole in 250 µl fractions. Proteins were dialyzed with the PD-10 Desalting Column (GE Healthcare, #GE17-0851-01) in a buffer consisting of 20 mM Tris-HCl pH = 7.5, 1 mM MgCl2, 150 mM NaCl, 10% (v/v) glycerol, and 1 mM DTT, and were concentrated. These fractions were then utilized for the fluorescence polarization assays.

The purified protein stocks were used from a maximum concentration of 4 µM, to a minimum concentration of 2 nM in twofold serial dilutions in ice-cold buffer containing 20 mM HEPES pH 7.0, 100 mM NaCl, and 5% (v/v) glycerol. FITC-labeled oligonucleotides (Metabion) carrying 2.5x, 2.0x, and 1.5x repeats of either telomeric (G- or C-rich), or control sequence were used for this assay. Double-stranded oligonucleotides were prepared by mixing 1:1 with the respective reverse complement oligonucleotide. For annealing, oligonucleotides were heated to 95 °C and then cooled at 0.1 °C/s until 4 °C. Diluted proteins were incubated with a final concentration of 20 nM FITC-labeled probe for 10 min at room temperature. Samples were measured with a Tecan Spark 20 M (Tecan). Experiments were conducted using three replicates for each condition. Analysis was performed with Graph Pad Prism 9.0 and specific binding was measured with Hill slope.

**C. elegans complete protein extract preparation**. Animals were washed off the plates with M9 buffer, synchronized by bleaching and grown to the desired stage, at which point worms were collected with M9 buffer. Worms were washed 3–4 times in M9, washed one last time with H$_2$O and frozen in 100–200 µl aliquots. Upon extract preparation, the aliquots were thawed, mixed 1:1 with 2x Lysis Buffer (50 mM Tris/HCl pH 7.5, 300 mM NaCl, 3 mM MgCl$_2$, 2 mM DTT, 0.2 % Triton X-100, Protease inhibitor tablets), and sonicated in a Bioruptor 300 (Diagenode) for 10 cycles with 30 s on/off, on high level. After sonication, the samples were centrifuged at 21,000 x g for 10 min to pellet cell debris. The supernatant was transferred to a fresh tube. With the exception of embryos (see below), extract of all developmental stages of C. elegans was prepared as described above. Samples of each developmental stage (for Fig. 2a) were collected in the following time points after plating of synchronized L1s: L1s were collected ~7 h after plating to recover

from starvation; L2s, ~12 h; L3s, ~28 h; L4, ~49 g; and YAs were collected ~56 h after plating.

For mixed-stage embryo extract preparations, synchronized gravid adults were harvested by washing them off the plate with M9 buffer. The worm suspension was washed with M9 until the supernatant was clear. Then, animals were bleached until all gravid adults were dissolved and only mixed-staged embryos remained. The embryos were subsequently washed in M9 buffer for three times then transferred to a new tube and washed one more time. In the last wash step the embryos were resuspended in 1x lysis buffer (25 mM Tris/HCl pH 7.5, 150 mM NaCl, 1.5 mM MgCl$_2$, 1 mM DTT, 0.1 % Triton X-100, protease inhibitors) and frozen in liquid nitrogen. After freezing, the pellets were ground to a fine powder in a pre-cooled mortar, then transferred to a cold glass douncer and sheared for 40 strokes with piston B. The suspension was pipetted to 1.5 ml tubes and spun down at 21,000 x g for 15 min at 4 °C. Finally, the supernatant was transferred to a fresh tube.

### Immunoprecipitation (IP)

*GFP IP*. IPs with GFP-tagged proteins were performed with GFP-binding magnetic agarose beads (GFPtrap MA, Chromotek, #gtma-20). Per IP sample, 10 µl of bead slurry was used and washed two times with 500 µl Wash Buffer (10 mM Tris/HCl pH 7.5, 150 mM NaCl, 0.5 mM EDTA, 1:1000 Pepstatin A/Leupeptin, 1:100 PMSF). Afterwards, the beads were resuspended in 450 µl Wash Buffer and up to 1 mg of complete extract of the respective C. elegans strain (of mixed-stage embryos or young adults) was added to a final volume between 500 and 750 µl. The IP samples were incubated at 4 °C rotating for 2 h. Following three washing steps with 500 µl Wash Buffer the beads were resuspended in 1x LDS (4x NuPAGE LDS sample buffer, Thermo Scientific, #NP0008) supplemented with 100 mM DTT and boiled at 70 °C for 10 min. When used for mass spectrometry, the samples were prepared in quadruplicates per strain/condition. In the IP-MS related to Supplementary Fig. 5e, f, the Wash Buffer was supplemented with 2 mM MgCl$_2$ and 0.05% of recombinant endonuclease from *Serratia marcescens*, or Sm nuclease[73], produced by the IMB's Protein-Production Core Facility.

*FLAG IP*. IPs with FLAG-tagged protein were performed with Protein G magnetic beads (Invitrogen™ Dynabeads™ Protein G; #10004D) and α-FLAG antibody (Monoclonal ANTI-FLAG® M2 antibody produced in mouse, Sigma Aldrich, #F3165). Per IP, 30 µl of beads were used and washed three times with 1 ml Wash Buffer (25 mM Tris/HCl pH 7.5, 300 mM NaCl, 1.5 mM MgCl$_2$, 1 mM DTT, 1 complete Mini protease inhibitor tablet per 50 ml). The beads were resuspended in 450 µl Wash Buffer and up to 1 mg of complete protein extract from the respective C. elegans strains was added. Finally, 2 µg of FLAG antibody were added and the samples were incubated for 3 h, rotating at 4 °C. After the incubation, the samples were washed three to five times with 1 ml Wash Buffer (see washing steps before), the beads were resuspended in 1x LDS/DTT, and the samples were boiled at 95 °C for 10 min. For mass spectrometry, IPs were prepared in quadruplicates per strain/ condition. When doing the IP with Sm nuclease, the wash buffer was supplemented with 0.05% Sm nuclease (as indicated above).

### Western blot

Protein samples were boiled at 70 °C for 10 min and loaded on a 4–12% Bis-Tris gel (NuPAGE, Thermo Scientific, #NP0321), running at 150-180 V for 60–120 min in 1x MOPS. After the run, the gel was shortly washed in VE H$_2$O and equilibrated in transfer buffer (25 mM Tris, 192 mM Glycine, 20% Methanol). A nitrocellulose membrane (Amersham Protran, VWR, #10600002) was equilibrated in transfer buffer as well. Membrane and gel were stacked with pre-wet Whatman paper (GE Healthcare-Whatman, #WHA10426892) and immersed in a blotting tank (Bio-Rad) filled with ice-cold transfer buffer and additionally cooled with a cooling element. The proteins were blotted at 300 mA for 60–120 min depending on the size. If blotted for 90–120 min for larger proteins, the transfer was carried out with a blotting tank on ice to keep the temperature. After blotting, the membranes were further prepared according to the respective antibody protocol.

*Anti-His antibody*. Membranes were blocked in Blocking Solution (PentaHis Kit, Qiagen, #34460) for 1 h at room temperature. After three 5 min washes in TBS-T (1x TBS, 0.1% Tween-20, 0.5% Triton X-100) the membranes were incubated with the Anti-His-HRP conjugated antibody in a dilution of 1:1000 in Blocking Solution for 1 h at room temperature. The membranes were then washed again three times in TBS-T and incubated with ECL Western Blot reagent (Thermo Scientific™ SuperSignal™ West Pico PLUS Chemiluminescent Substrate, #15626144; mixed 1:1) for detection. Western blot ECL detection was performed with the ChemiDoc XRS+system (BioRad, Software: Image Lab 5.2.1).

*Anti-GFP, Anti-FLAG, and Anti-Actin antibodies*. Western blot analysis was performed using the following primary antibodies: an anti-GFP antibody (Roche, Anti-GFPfrom mouse IgG1κ (clones 7.1 and 13.1), #11814460001; 1:1000 in Skim Milk solution), an anti-FLAG antibody (Sigma-Aldrich, mouse Monoclonal ANTI-FLAG® M2 antibody, # F3165; 1:5000 in Skim Milk solution), and an anti-Actin antibody (Sigma-Aldrich, rabbit anti-actin, #A2066; 1:500 in Skim Milk solution). After blotting, membranes were blocked in Skim Milk solution (1x PBS, 0.1% Tween-20, 5% (w/v) Skim Milk Powder) for 1 h at room temperature. The

incubation with the primary antibody was carried out at 4 °C, rotating overnight. Membranes were washed in PBS-T (1x PBS, 0.1% Tween-20) three times for 10 min, they were incubated with an HRP-linked secondary antibody (for anti-flag and anti-GFP with Cell Signaling Technology, anti-mouse IgG, #7076; 1:10,000 dilution in Skim Milk Solution; for anti-actin the secondary used was GE Healthcare, anti-Rabbit IgG, #NA934; 1:3000 in Skim Milk solution) for 1 h rotating at room temperature. Following three washes in PBS-T the membranes were incubated with ECL solution (Thermo Scientific™ SuperSignal™ West Pico PLUS Chemiluminescent Substrate, #15626144; mixed 1:1) for detection. Western blot ECL detection was performed with the ChemiDoc XRS+system (BioRad, Software: Image Lab 5.2.1). Incubation with Anti-Actin antibody was typically performed after detection of GFP/FLAG and subsequent washes.

*Antibody protocol for co-IPs (LI-COR antibodies).* For co-IP experiments, we first probed the IP bait with HRP-linked secondary antibodies, as described above. Then, we probed for the co-IP using LI-COR secondary antibodies. After incubation with primary antibody, as described above, membranes were washed and incubated with secondary antibodies compatible with the LI-COR System (FLAG/GFP: Licor IRDye® 680RD Donkey anti-Mouse IgG (H+L), #926-68072; Actin: Licor IRDye® 800CW Donkey anti-Rabbit IgG (H+L), #926-32213; both 1:15,000 in Skim Milk solution) for 1 h at room temperature. After three additional washes with PBS-T, the membranes were imaged using an Odyssey CLx scanner and processed using Image Studio software (LI-COR, Version 3.1).

**C. elegans culture and strains.** *C. elegans* was cultured under standard conditions on Nematode Growth Medium (NGM) plates seeded with *E. coli* OP50 bacteria[74]. For proteomics experiments, animals were grown on OP50 high-density plates (adapted from ref. [75]). In specific, the yolks of commercially available chicken eggs were isolated, added to LB medium (50 ml per egg yolk) and thoroughly mixed. Subsequently, the mix was incubated at 65 °C for 2–3 h. Pre-grown OP50 liquid culture is added to the mix (10 ml per egg), after the yolk-LB mixture cooled down. This preparation was poured into 9 cm plates (10 ml per plate) and plates are decanted the next day. Plates remained for 2–3 days at room temperature, for further bacterial growth and drying.

Animals were grown at 20 °C, except when noted. The standard wild-type strain used in this study was N2 Bristol. Strains used and created in this study are listed in Supplementary Table 1.

**Fertility assays.** For brood size counts of the homozygous single mutants, L3 worms were isolated, per strain and were grown either at 20 or 25 °C. After reaching adulthood, worms were transferred to a new plate every day, until no eggs were laid in 2 consecutive days. Viable progeny was counted approximately 24 h after removing the parent. For the experiment shown in Fig. 4d, e, a cross between *tebp-1(xf133)* males and *tebp-2(xf131)* hermaphrodites was performed, the genotypes of the F1 and F2 were confirmed by PCR genotyping. L2/L3 progeny of F2 *tebp-1(xf133)/ + ; tebp-2(xf131)* mothers were isolated and grown at 20 °C, or 25 °C. During adulthood, the viable brood size was counted as mentioned above. The assayed F3s were genotyped 2 days after egg laying stopped. For all brood size experiments, worms that died before egg laying terminated, e.g., by dehydration on the side of plate, were excluded from the analysis.

**Mortal germline assay.** All strains used in the Mortal Germline assay were outcrossed with wild-type N2 two times before the experiment. Six L3 larvae of the chosen strains were picked per plate ($n = 15$ plates per strain) and grown at 25 °C. Six L3 larvae were transferred to a fresh plate every 5 days (equivalent to two generations). This procedure was followed until plates were scored as sterile, when the six worms transferred failed to produce six offspring to further isolate, on 2 consecutive transfer days.

**pgl-1::mTagRfp-T; tebp-1 x pgl-1::mTagRfp-T; tebp-2 cross and definition of categories of germline defects.** We crossed *pgl-1::mTagRfp-T; tebp-1* males with *pgl-1::mTagRfp-T; tebp-2* hermaphrodites. F1 cross progeny was confirmed by genotyping. 300 F2 progeny were singled and left to self-propagate. After genotyping F2 worms, we isolated 60 F3 worms from three different *tebp-1(xf133);tebp-2(xf131)/ +* , 60 F3 worms from three different *tebp-1(xf133)/ + ;tebp-2(xf131)* mothers, as well as 10 F3 worms from two different single mutant mothers as controls. Additionally, all synthetic sterility escaper progeny from *tebp-1; tebp-2* double-homozygous worms were singled to check their fertility. Germline health, as well as growth and other phenotypes for all singled worms were determined at day 2 of adulthood. Germlines were categorized by microscopy with a Leica M80 stereomicroscope with a fluorescence lamp (Leica EL 6000), according to the morphology of the germline, as assessed by PGL-1::mTagRFP-T expression: category 1, near wild-type morphology; category 2, one gonad arm is atrophied; category 3, both gonad arms are atrophied. After germline categorization, worms were genotyped. We repeated this procedure until the F5, always using the progeny of *tebp-1(xf133);tebp-2(xf131)/ +* or *tebp-1(xf133)/ + ,tebp-2(xf131)* mothers, as well as sibling controls. The barplots depicting the final distribution of germline categories across all scored generations was created using R and publicly available packages (ggplot2-v 3.2.1, reshape–v 0.8.8, viridis–v 0.5.1, scales–v 1.0.0).

**Scoring crosses of tebp-1 x tebp-2 mutant animals.** Owing to the onset of synthetic sterility in F2 *tebp-1; tebp-2* double mutant animals, > 100 of F2 progeny was singled from the F1 heterozygous parent. F2 worms were genotyped at the adult stage after 3–4 days of egg laying and genotypes were determined and correlated with fertility. Progeny descending from *tebp-1; tebp-2* double mutant synthetic sterility escaper F2s were singled and allowed to grow and lay eggs for 3–4 days. Subsequently, these double mutant F3s were genotyped and their fertility was determined. Boxplots depicting the results were created using R and publicly available packages (ggplot2-v 3.2.1, reshape–v 0.8.8, viridis–v 0.5.1, scales–v 1.0.0).

**Creation of mutants using CRISPR-Cas9 technology.** Mutants were created as described[76], with the following specifications. To create *tebp-2(xf131)*, N2 animals were injected with a mix of three constructs: 25 ng/μl of co-injection marker pCFJ104 (*Pmyo-3:mCherry:unc-54 3'UTR*, a gift from Erik Jorgensen, Addgene plasmid #19328; http://n2t.net/addgene:19328; RRID:Addgene_19328); 100 ng/μl of a construct expressing Cas9 and a sgRNA targeting the sequence ACAT-GAGTCTGTGTTTACGG (derived from pDD162, which was a gift from Bob Goldstein, Addgene plasmid # 47549; http://n2t.net/addgene:47549; RRID: Addgene_47549); and 75 ng/μl of a construct expressing a sgRNA targeting ACGGCTCATAAGAGACTTGG (derived from p46169, which was a gift from John Calarco, Addgene plasmid # 46169; http://n2t.net/addgene:46169; RRID: Addgene_46169).

To produce *tebp-1(xf133)* and *tebp-1(xf134)*, the following mix was injected into N2 animals: 25 ng/μl of pCFJ104; 150 ng/μl of a construct expressing Cas9 and a sgRNA targeting the sequence GCATGTCGAGATTCTACTGG (derived from pDD162); and 80 ng/μl of a construct expressing a sgRNA targeting GCTTCAAAATTTCTCCAGGG (derived from p46169). After isolation, PCR genotyping and confirmation by Sanger sequencing, mutants were outcrossed four times against the wild type.

**Creation of endogenous tags and a tebp-1; pot-2 double mutant via CRISPR-Cas9-mediated genome editing.** Protospacer sequences were chosen using CRISPOR (http://crispor.tefor.net)[77], cloned in pRFK2411 (plasmid expressing Cas9 + sgRNA(F+E);[78] derived from pDD162) or pRFK2412 (plasmid expressing sgRNA(F+E)[78] with Cas9 deleted; derived from pRFK2411) via site-directed, ligase-independent mutagenesis (SLIM)[79,80]. pDD162 (Peft-3::Cas9 + Empty sgRNA) was a gift from Bob Goldstein (Addgene plasmid # 47549; http://n2t.net/addgene:47549; RRID:Addgene_47549)[81]. All plamids were purified using NucleoSpin® Plasmid from Macherey-Nagel, eluted in sterile water and confirmed by enzymatic digestion and sequencing. All Cas9 nuclease induced double-strand breaks (DSBs) were within 20 bp distance to the desired editing site. All CRISPR-Cas9 genome editing was performed using either *dpy-10(cn64)* or *unc-58(e665)* co-conversion strategies[82]. Single-stranded oligodeoxynucleotides (ssODN, 4 nmole standard desalted Ultramer™ DNA oligo from IDT) and PCR products (purified using QIAquick® PCR Purification Kit from QIAGEN) served as donor templates for small (3xFLAG epitope tag, protospacer sequences) and big (GFP tag) insertions, respectively. The *gfp* coding sequence including three introns and flanking homology regions was amplified from pDD282, which was a gift from Bob Goldstein (Addgene plasmid # 66823; http://n2t.net/addgene:66823; RRID: Addgene_66823)[83]. All donor templates contained ~35 bp homology regions[84,85]. Plasmid vectors, ssODN and PCR products were diluted in sterile water and injected at a final concentration of 30–50 ng/μl, 500–1000 nM and 300 ng/μl, respectively. For GFP insertions, the protospacer sequence used for the *dpy-10* co-conversion was transplanted to the editing site to generate d10-entry strains[86], which in turn served as reference strains for further injections. DNA mixes were injected in both gonad arms of 10–25 1-day-old adult hermaphrodites maintained at 20 °C. Co-converted F1 progeny were screened for insertions by PCR. Successful editing events were confirmed by Sanger sequencing. All generated mutant strains were outcrossed at least two times prior to any further cross or analysis. CRISPR-Cas9 genome editing reagents and DNA injection mixes are listed in Supplementary Data file 5. The *pgl-1::mTagRfp-T* is described elsewhere[49,50].

**Creation of transgenic worms using MosSCI.** A TEBP-2::GFP fusion transgene was produced as previously described[87], and as indicated in www.wormbuilder.org. Animals of the strain EG6699 were injected, in order to get insertions in locus *ttTi5605* on LGII. The injection mix contained all the injection constructs listed in www.worbuilder.org, using the recommended concentrations, including 50 ng/μl of a repair template containing the *tebp-2::gfp* sequence. Selection was performed as recommended in www.wormbuilder.org[76].

**Extraction of genomic DNA from C. elegans.** Mixed-staged animals were washed off plates with M9 and washed two to three more times in M9. Next, worms were resuspended in Worm Lysis buffer (WLB: 0.2 M NaCl, 0.1 M Tris/HCl pH 8.5, 50 mM EDTA, 0.5% SDS) and aliquoted in 250 μl samples. For genomic DNA extraction the aliquots were brought to a final volume of 500 μl with WLB and Proteinase K (30 μg/ml). To lyse the worms, the samples were incubated at 65 °C at 1400 rpm for > 2 h until all carcasses were dissolved. The samples were then centrifuged at 21,000 x *g* for 5 min to pellet debris and the supernatant was transferred to a fresh tube. Afterwards, 500 μl of Phenol:Chloroform:

Isoamylalcohol were added, the samples shaken vigorously for 30 s and spun down at 16,000 x g for 5 min. Additionally, 500 μl of chloroform were added to the samples and again shaken vigorously for 30 sec and spun at 16,000 x g for 5 min. The aqueous phase of the samples was transferred to fresh 2 ml reaction tubes and 50 μg RNase A were added to digest the RNA. The tubes were inverted once and incubated at 37 °C for > 1 h. After RNA digestion the samples were again purified by phenol:chloroform:isoamylalcohol and chloroform addition (as before). The aqueous phase was transferred to fresh tubes and the DNA was precipitated with 350 μl isopropanol for > 15 min at −80 °C. To pellet the DNA, the samples were centrifuged at 21,000 x g for 20 min at 4 °C. The supernatant was carefully removed and the DNA pellet washed once with 1 ml of ice-cold 70% ethanol and spun at 21,000 x g for 5 min at 4 °C. Washing was repeated if the samples still smelled of phenol. After washing the supernatant was completely removed, the pellet air dried for ca. 10 min, and resuspended in 20 μl H$_2$O. To fully resuspend the DNA, the samples were kept at 4 °C overnight and mixed again the next day.

**Telomere Southern blot**. For denatured telomere Southern blot 15 μg of *C. elegans* genomic DNA were digested in 80 μl total volume with 40 U HinfI (New England BioLabs, #R0155) and RsaI (New England BioLabs, #R0167), respectively. The digestion was incubated at 37 °C overnight and the next day additional 10 U of each enzyme were added and the samples incubated 1–2 h further. Afterwards the samples were evaporated in a Concentrator Plus at 45 °C to end up with a volume of 20–30 μl and supplemented with 2x DNA loading dye. A 0.6% agarose gel was prepared (with 1x TBE and 16 μl SYBR Safe DNA stain, Thermo Fischer Scientific, #S33102) and the samples loaded after boiling at 95 °C for 10 min. The GeneRuler 1 kb (Thermo Scientific, #SM0312), as well as the 1 kb extended markers (New England BioLabs, #N3239) were used. The samples were secured in the gel by running it at 100 V for 20–30 min then the voltage was set to 60 V for a run overnight (16–19 h). With a crosslinker set to 1 min crosslinking time, the DNA was broken and the gel afterwards equilibrated in transfer buffer (0.6 M NaCl, 0.4 M NaOH) for at least 20 min. After equilibration, an upward alkaline transfer was set up with whatman paper and a positively charged nylon membrane (Byodine B membrane; Pall, #60207), all equilibrated in transfer buffer. The transfer was set up overnight. Following blotting, the membrane was fixed by incubation in 0.4 M NaOH for 15 min with slight agitation and neutralized with two washes in 2x SSC for 5 min each. To keep hydrated the membrane was sealed in cling film with 2x SSC until hybridization.

The membrane was pre-hybridized in a glass hybridization tube with 20 ml hybridization buffer (3.3x SSC, 0.1% SDS, 1 mg/ml Skim Milk powder) for at least 1 h at 42 °C rotating in a hybridization oven. The oligonucleotide used for detection was a TTAGGC reverse complement triple repeat (GCCTAA)$_3$. The probe was radioactively labeled with 3 μl 32P-[γ]-ATP by a polynucleotide Kinase reaction and cleaned up using a MicroSpin Sephadex G-50 column (GE Healthcare, #GE27-5330-01). The labeled oligonucleotide was denatured at 95 °C for 10 min and mixed with 20 ml fresh hybridization buffer. This mix was added to the membrane after removing the previous buffer and incubated 3.5 days rotating at 42 °C.

After hybridization the membrane was washed by first rinsing it twice with Wash Buffer 1 (2x SSC, 0.1% SDS), then incubating it twice for 5 min in 20 ml Wash Buffer 1. For the last wash, the membrane was incubated for 2 min in Wash Buffer 2 (0.2x SSC, 0.1% SDS), then rinsed in 2x SSC to re-equilibrate the salt concentration. The membrane was dried on a Whatman paper for 3 h, sealed in cling film and exposed to a phosphoimager screen for 3 days. The screen was read out with the Typhoon Scanner with the settings 1000 V PMT and 200 μm pixel size. Contrast and brightness of the resulting tif-file were optimized using Fiji.

**Microscopy**

*Co-localization microscopy*. Strains carrying TEBP-1::GFP or TEBP-2::GFP were crossed with strain YA1197 expressing POT-1::mCherry. Adult animals were washed in M9 buffer, immobilized in M9 buffer supplemented with 40 mM sodium azide and mounted on freshly made 2% agarose pads. For imaging embryos, adult hermaphrodites were washed and dissected in M9 buffer before mounting. Animals were immediately imaged using a TCS SP5 Leica confocal microscope equipped with a HCX PL APO 63x water objective (NA 1.2), Leica hybrid detectors (HyD), and the acquisition software Leica LAS AF. Deconvolution was performed using Huygens Remote Manager and images were further processed using Fiji.

*PGL-1 fluorescence microscopy*. For imaging PGL-1::mTagRFP-T in animals of each category of germline morphology, adult worms were picked to a droplet of M9 to remove OP50 bacteria, then transferred to a drop of M9 buffer supplemented with 40 mM sodium azide in M9 for immobilization on a 2% agarose pad. Animals were immediately imaged with a Leica AF7000 widefield microscope using a 20x objective (NA 0.4) and red fluorescence filters (N3), as well as TL-DIC (acquisition software: Leica LAS X, camera: Hamamatsu, Orca Flash 4.0 V2). Images were processed using Fiji (brightness changes applied only in DIC channel for better visualization).

*Quantitative FISH (qFISH)*. For telomere length determination, fluorescence in situ hybridization (FISH) was utilized in a quantitative manner[88]. The staining protocol was optimized after the work of Seo and Lee[89]. Per strain, 100 gravid adults were

picked to an unseeded small NGM plate to remove the majority of OP50 bacteria. From there, worms were picked to a 5 μl drop of Egg buffer (25 mM HEPES/KOH pH 7.4, 118 mM NaCl, 48 mM KCl, 2 mM EDTA, 0.5 mM EGTA, 1% Tween-20) on a cover slip and dissected using 20 gauge needles (Sterican, Roth #C718.1) to release embryos and gonads. The samples were fixed by adding 5 μl of 2% Formaldehyde solution and incubating for 5 min. To remove the Formaldehyde solution, samples were washed on the cover slip by adding and removing Egg buffer carefully by pipetting. For permeabilization of the cuticle, the worms were afterwards treated by freeze cracking[90]. The cover slips were put on a Poly-lysine coated slide (Sigma Aldrich, #P0425) and the slides transferred to an aluminum block on dry ice for freezing. After 15 min freezing on the aluminum block, the cover slips were removed and the slides immersed first in ice-cold methanol, then in ice-cold acetone for 5 min, respectively. To remove the solutions the slides were washed in 1x PBS (10 mM Na2HPO4, 2 mM KH2PO4, 137 mM NaCl, 2.7 mM KCl) for 15 min. For additional permeabilization the samples were incubated in permeabilization buffer (20 mM Tris/HCl pH 7.5, 50 mM NaCl, 3 mM MgCl$_2$, 300 mM Sucrose, 0.5% Triton X-100) at 37 °C for 30 min followed by a wash in 1x PBS for 5 min at room temperature. To prevent unspecific binding of the FISH probe, the samples were treated with 20 μl RNase A solution (1x PBS, 0.1% Tween-20, 10 μg/ml RNase A) at 37 °C for 1 h in a humid chamber. Afterwards the slides were washed in 1x PBS-T (1x PBS, 0.1% Tween-20) for 10 min at room temperature and dehydrated by successive 3 min washes in 70%, 85 and 100% ethanol and air dried. For pre-hybridization 50 μl of hybridization solution (3X SSC, 50% Formamide, 10% (w/v) Dextran-Sulfate, 50 μg/ml Heparin, 100 μg/ml yeast tRNA, 100 μg/ml sheared salmon sperm DNA) were added to the sample and the slides incubated in a humid chamber for 1 h at 37 °C. The FISH probe (PNA-FISH TTAGGC telomeric probe, Panagene, resuspended to 100 μM, fluorophore: Alexa-555) was prepared as a 1:500 dilution in hybridization solution and denatured for 5 min at 70 °C. After pre-hybridization, the solution on the slides was removed as much as possible by pipetting and 20 μl of FISH probe were added, then covered by a cover slip. For hybridization of the probe the slides were denatured on a heat block prepared with wet paper towels for humidity at 80 °C for 3 min and transferred to a humid chamber for incubation overnight at 37 °C. The next day the slides were washed twice in 1x PBS-T for 5 min to remove the probe. To fixate the staining, the samples were incubated in hybridization wash solution (2X SSC, 50% Formamide) for 30 min at 37 °C. As a last step the slides were washed in 1x PBS-T twice for 15 min at room temperature and mounted by adding 10–20 μl Vectashield mounting medium containing DAPI (Vector laboratories, #H-1200-10). The pictures were taken with a Leica TCS SP5 confocal microscope (objective: CX PL APO CS 63sx oil NA: 1.4, pinhole 60.05 μm, 2x zoom, PMT detectors, acquisition software Leica LAS AF). The images stacks were composed by a sequence of pictures acquired every 0.5 μm on the z-axis. The laser and gain settings were adjusted according to the sample with the lowest FISH intensity. For analysis, images were opened in Image J/Fiji and the channels split into the DAPI and red channel. A mask of the image was created to infer the volume of the imaged object. The threshold function of the software was used with activated plugins for identification of round objects (Otsu). After setting the threshold for the image in the histogram settings, the z-stack was converted to a binary mask and using the 3D OC Options menu volume, mean gray values and integrated density of the FISH foci were calculated. Additionally, the 3D Object counter menu was used and the filters set to a minimum of 2. The values obtained by this analysis were averaged over several images of either germlines or embryos of the same strain and used for quantitative comparison of telomere length. For comparison, all values obtained for the mutant strains were scaled relative to the average of the wild type values. The barplots were created with R with standard and publicly available scripts (RColorBrewer-v 1.1-2, ggpubr-v 0.4.0, plyr-v 1.8.6, viridis-v 0.5.1, viridisLite-v 0.3.0, ggforce-v 0.3.2, ggsignif-v 0.6.0, dplyr-v 1.0.2, ggplot2-v 3.3.3, readr-v 1.4.0).

**Yeast two-hybrid assay**. Yeast two-hybrid assays were conducted in the yeast strain PJ69-4α as described before[91,92]. The respective Gal4 activation and DNA-binding domain plasmid pairs were co-transformed in PJ69-4α. The resulting transformants were resuspended in ddH$_2$O and pinned on SC Trp-Leu-, SC Trp-Leu-His-, and SC Trp-Leu-His-Ade- plates. For Fig. 6a an additional round of plasmid transformation was performed, as a biological duplicate, and the results were identical. Colonies were imaged with a ChemiDoc XRS+system (BioRad, Software: Image Lab 5.2.1) for Fig. 6a and Supplementary Fig. S6g, and scanned with an Epson Scanner (Perfection V700 Photo, Software version 3.81) for Fig. 6f–h and supplementary Fig. 6i.

**Size-exclusion chromatography**. Size-exclusion chromatography was performed as previously described[76,92]. For the first run (Supplementary Fig. 5a) two embryo samples were prepared and combined. Using a centrifugal filter with a 10 kDa cutoff (Merck, Amicon Ultra 0.5 ml 10 K, #UFC5010) the sample was concentrated to a final volume of 550 μl. Between 3.6 and 3.8 mg of total extract was separated on a Superose 6 10/300 GL column (GE Healthcare, 17517201) operated on a NGC Quest System (Bio-Rad) using lysis buffer without Triton X-100 as running buffer (25 mM Tris/HCl pH 7.5, 150 mM NaCl, 1.5 mM MgCl$_2$, 1 mM DTT, protease inhibitors). Five-hundred microliter fractions were collected according to the scheme in Supplementary table 2. Selected fractions were concentrated to 30 μl using 10 kDa cutoff centrifugal filters (Merck, Amicon Ultra 0.5 ML 10 K,

#UFC5010). The samples were supplemented with 4x LDS (NuPAGE) and 100 mM DTT to a final volume of around 40 μl and boiled at 95 °C for 10 min. After spinning down, a part of each sample was run on a 4-15% Criterion TGX Stain-Free Protein Gel (26 wells, Bio-Rad, #5678085) in 1x SDS running buffer at 200 V for 32 min. Transfer of proteins to a nitrocellulose membrane (Bio-Rad, #1620112) was performed using the Trans-Blot Turbo Transfer System (Bio-Rad). Following the transfer, western blot was performed as described above. For the second run (Fig. 5a, b), four embryo extracts were prepared, combined and concentrated, as above, to 1 ml. Then half of the sample was treated with Sm nuclease for 30 min at 4 °C, prior to size-exclusion chromatography, while the other half was not.

**Phylogenetic and synteny analysis**. The protein sequences of *C. elegans* TEBP-1 and TEBP-2 were extracted from Wormbase (WS275). These sequences were used separately as queries for Wormbase BLASTP search in the available genomes. Orthologs of TEBP-1 and TEBP-2 were defined based on two criteria: (1) BLASTP hit had an E-value lower than 1.00e-15; and (2) reciprocal BLASTP of the hit, querying the *C. elegans* proteome, resulted in TEBP-1 and TEBP-2 as top hits. Sequences of the identified orthologs were obtained from Wormbase (WS275) and Wormbase ParaSite (WBPS14/WS271). The list of identified orthologs and BLASTP results can be found in Supplementary Data file 2 (sheet 1).

The full-length protein sequences of TEBP orthologs were used for multiple sequence alignment using MAFFT, version 7.452[93]. Alignment was performed using default settings, including an automatic determination of best alignment strategy, which provided the L-INS-I result[94]. Multiple sequence alignment can be found in Supplementary Data file 2 (sheet 2). Then, the multiple sequence alignment in fasta format was used as an input for IQ-TREE version 1.6.12[95], with branch supports obtained with ultrafast bootstrap[96]. IQ-TREE was first ran to determine the best fit substitution model, which was VT+F+R3. Then, analysis was repeated with the following parameters: -redo -m VT+F+R3 -bb 10000 -o Cang_2012_03_13_00535. g11959_Cang, Cang_2012_03_13_01061.g15539.t3_Can, where -m is the best fit model, -bb is the number of ultrafast bootstrap replicates, and –o represents the defined outgroups. Output.tree file was visualized in FigTree version 1.4.4 (http://tree.bio.ed.ac. uk/software/figtree/). The *C. angaria* TEBP orthologs were used as outgroups, as this species is not part of the *Elegans* and *Japonica* groups, according to recent phylogenetic studies[97]. To create an additional tree with the N-terminal region only, the initial multiple sequence alignment was trimmed to the 600 initial alignment positions. The alignment of this region (with similarity to the homeodomain of RAP1) was substantially more reliable, as assessed by higher GUIDANCE2 scores[98]. Using this edited alignment, another tree was constructed as described above. IQ-TREE best fit model was VT+F+I+G4, parameters used: -m VT+F+I+G4 –bb 10000 -o Cang_2012_03_13_00535.g11959_Cang, Cang_2012_03_13_01061.g15539.t3_Can.

We defined local synteny across species as the maintenance of linkage in at least one of the neighboring genes upstream and downstream of the respective *tebp* gene. We used two different strategies to determine synteny. (1) Synteny was determined by navigating genome browser tracks through regions containing *tebp* orthologs, using Wormbase ParaSite (WBPS14/WS271). Currently annotated genes, adjacent to *tebp* orthologs, were selected, their predicted protein sequences were retrieved and BLASTP was performed in the *C. elegans* genome to find the corresponding ortholog. Results are summarized in Supplementary Data file 2 (sheet 3). (2) The protein sequences obtained previously by reciprocal BLASTP of TEBP-1 and TEBP-2 were used as an entry for WormBase ParaSite BioMart tool (https://parasite.wormbase.org/biomart). We recouped the neighboring 13 genes upstream and 13 genes downstream and, with the resulting gene ID list, we determined a set of orthologous genes with the following series of 'Output attributes': gene stable ID, chromosome/scaffold, start (bp) and end (bp) coordinates that were to be listed in the result from ten available complete *Caenorhabditis* genomes. Subsequently, we filtered only those genes that share the same chromosome/scaffold with the *tebp* orthologous gene, finally, we evaluate if the enlarged group meets our definition of local synteny. We repeated this process taking each of the *tebp* genes in the ten species as a reference and evaluated the filtered groups for local synteny. In the specific case of *C. remanei*, WormBase ParaSite provides three different assemblies: PRJNA248909, PRJNA248911 and PRJNA53967. The latter was the only assembly where we were able to identify synteny of *tebp-1* with BioMart, although we could verify it manually for PRJNA248911. Results are summarized in Supplementary Data file 2 (sheet 4). This strategy was not applicable to *C. angaria*, as the genome of this species is not implemented in WormBase ParaSite BioMart.

**Analysis of previously published RNA-seq datasets**. For the expression data of the telomeric proteins during development of *C. elegans* (Supplementary Fig. 2a–c), RNAseq data was taken from a previously published dataset[47]. To probe expression of the telomeric genes in spermatogenic and oogenic gonads (Supplementary Fig. 2d), previously published transcriptome data was used[48]. Gene expression and genome browser tracks were plotted using Gviz[99] and GenomicFeatures[100] on an R framework (R Core Team 2018).

**RNA extraction and library preparation**. RNA was extracted as described[47]. Synchronized young adult animals were frozen in 50–100 μl of H$_2$O after harvest.

After thawing, 500 μl TRIzol LS reagent (Invitrogen, # 10296010) was added and the worms were lysed with six freeze-thaw cycles (frozen in liquid nitrogen for ca. 30 s, then thawed for 2 min in a 37 °C waterbath and vortexed). Following lysis, the samples were spun down at full speed for 2 min to pellet debris. Supernatant was transferred to a fresh tube, mixed 1:1 with 100% ethanol and the mix was transferred to a column of the Direct-zol RNA MiniPrep Plus Kit (Zymo Research, #R2070). The following purification steps were done according to manufacturer's instructions, including the recommended in-column DNase I treatment for 25–40 min. RNA samples were eluted in 30–32 μl of RNase-free H$_2$O.

Library preparation for mRNA sequencing was performed with Illumina's TruSeq stranded mRNA LT Sample Prep Kit following Illumina's standard protocol (Part # 15031047 Rev. E). Libraries were prepared by using only ¼ of the reagents with a starting amount of 250 ng and they were amplified in ten PCR cycles. Libraries were profiled in a High Sensitivity DNA on a 2100 Bioanalyzer (Agilent technologies) and quantified using the Qubit dsDNA HS Assay Kit, in a Qubit 2.0 Fluorometer (Life technologies). Libraries were pooled in an equimolar ratio and sequenced on one NextSeq 500 Highoutput Flowcell, SR for 1 × 75 cycles plus 1 × 7 cycles for index read.

**mRNA read processing and mapping**. The library quality was assessed with FastQC (version 0.11.8) before alignment against the *C. elegans* genome assembly WBcel235 and a custom.GTF file, which included gene annotations from *C. elegans* (WormBase, c_elegans.PRJNA13758.WS269) and *E. coli* (EnsemblBacteria, Escherichia_coli_b_str_rel606.ASM1798v1). Alignment was performed with STAR aligner[101] version 2.6.1b. Reads mapping to annotated features in the custom.GTF file were counted with featureCounts[102] version 1.6.2 using featureCounts functionality. Counts aligning to *E. coli* were removed at this point from downstream analysis. Coverage tracks were generated with deepTools[103] version 2.27.1 and plotted using Gviz[99] on an R framework (R Core Team 2018).

**Reporting summary**. Further information on research design is available in the Nature Research Reporting Summary linked to this article.

## Data availability

The datasets supporting the conclusions of this article are available in the ProteomeXchange Consortium via Pride repository, PXD019241; and in the SRA, BioProject PRJNA630690.

## Code availability

Code is available upon request.

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

## Acknowledgements

We thank the members of the Butter and Ketting laboratories for helpful discussion and Brian Luke for critical reading of the manuscript. Franziska Roth of the Butter laboratory, Bruno de Albuquerque and Svenja Hellmann of the Ketting laboratory, Laura Tomini of the Ulrich laboratory, as well as Anja Freiwald and Mario Dejung of the Proteomics core facility provided critical technical assistance. The authors thank Shawn Ahmed, the *Caenorhabditis* Genetics Center (supported by NIH Office of Research Infrastructure Programs P40 OD010440), and the National Bioresource Project for the Experimental Animal C. elegans (Shohei Mitani) for kindly providing C. elegans strains used in this study. Assistance by the following IMB core facilities is gratefully acknowledged: Media Lab, Microscopy Core Facility, Genomics Core Facility, and to Martin Möckel of the Protein Production Core Facility. This project was funded by the Deutsche Forschungsgemeinschaft (DFG, German Research Foundation)—407023052/GRK2526/1 and Project-ID 393547839—SFB 1361. D.K. was supported by the National Research Foundation Singapore and the Singapore Ministry of Education under its Research Centres of Excellence initiative.

## Author contributions

Conceptualization, S.D., M.V.A., D.K., R.F.K., and F.B.; investigation, S.D., M.V.A., E.N., J.S., N.V., C.R.; formal analysis, S.D., M.V.A., E.N., N.V., A.F.-S., A.C.-N., and F.B.; visualization, S.D., M.V.A., E.N., J.S., A.F.-S., and F.B.; writing–original draft, S.D., M.V.A., and F.B.; writing–review & editing, all authors contributed; supervision: H.D.U., R.F.K., and F.B.; project administration: F.B.; funding acquisition, H.D.U., R.F.K., F.B.

## Funding

## Competing interests

The authors declare no competing interests.
