## [Peer Review File · Nature Communications]

REVIEWER COMMENTS

Reviewer #1 (Remarks to the Author):

Dietz and colleagues present a novel study where they use biochemical and genetic approaches (primarily) to define proteins that function at telomeres of *C. elegans*, a model organism that is at the forefront of many topics in contemporary experimental biology. Even though biochemical approaches have been an important unbiased method for addressing telomere biology in several experimental systems, attempts to use biochemistry to define *C. elegans* telomere proteins have yielded limited information. Telomeres from humans to yeast are known to be protected by 6 proteins that interact with telomeric DNA, two of which interact with double-stranded telomeric DNA (TRF1 and TRF2) and one of which interacts with single-stranded telomeric DNA (POT1). The only known *C. elegans* telomere proteins with homology to the Shelterin complex have homology to POT1, including the single-stranded telomere binding proteins Pot-1, Pot-2 and Mrt-1. The central power of the author's approach is to define proteins that specifically interact with a double-stranded oligonucleotide substrate of the *C. elegans* telomere sequence GGCTTA that has a single-stranded overhang, which should mimic the structure of *C. elegans* chromosome ends.

Impressively, the authors discovered most proteins known to function at *C. elegans* telomeres.

The authors also discovered Ku70 and Ku80 proteins that are known to play roles at telomeres in other experimental systems, which will be satisfying for the field.

The major advance of this manuscript is to define *C. elegans* counterparts of TRF1 and TRF2, which have been an open question for the past 20 years. Dietz and colleagues elegantly demonstrate that two highly diverged double-stranded DNA binding proteins TEBP-1 and TEBP-2 interact with double-stranded DNA at *C. elegans* telomeres, as demonstrated by compelling oligonucleotide binding assays. The authors also demonstrate that both proteins localize to *C. elegans* telomeres, and that TEBP-1 inhibits the recombination-based telomere replication pathway called ALT, loss of which causes long telomeres and enables telomere elongation without telomerase. In contrast, the authors show that TEBP-2 causes shortened telomeres and an inability to maintain the strain, which implies deficiency for telomerase. The authors then use genetics to test the possibility that loss of TEBP-1 will suppress the telomere maintenance defect of TEBP-2 mutants, as it does with telomerase mutants. However, the authors find that loss of both double-stranded telomere binding proteins results in almost immediate sterility accompanied by underproliferated germ stem cells that may be the result of cell lethality. This is a surprising interaction, because loss of one mammalian telomere binding protein TRF2 results in immediate telomere uncapping. The authors demonstrate that this telomere capping function is likely redundantly controlled by the two *C. elegans* double-stranded telomere binding proteins, TEBP-1 and TEBP-2, even though these proteins individually possess unique opposing functions in telomere maintenance.

Overall, the authors' unanticipated results speak to multifaceted functions of proteins that interact with double-stranded telomeric DNA. These results fill a major gap in the field of telomere biology by defining two *C. elegans* double-stranded telomere binding proteins. The essential nature of double-stranded telomere binding proteins in human has made it difficult to identify essential roles for these proteins in telomerase activity (which is mediated by the single-stranded telomere binding protein POT1 and its cofactor TPP1) or in telomere maintenance via ALT. Although there is more that the authors could do to expand on their findings, this study represents a great deal of work that has yielded interesting results. In addition, this study serves as a model for unbiased biochemical approaches to studying telomere biology in any experimental system, which the authors demonstrate by performing proteomics in the related nematode *C. briggsae*. This manuscript is an important study in the field of telomere biology, filling several gaps that will move the field forward. More broadly, it demonstrates a novel and highly productive use of biochemistry in *C. elegans*, which has traditionally relied on genetics, cell and developmental biology.

Comments:

1. While convincing in intensity, the morphology of the FISH signal in Fig. 3b-e appears odd, such that nuclei are not visible. Better quality images are shown for embryos in S3c-f, which show that *tebp-1* and *pot-2* mutants have much telomere FISH signal that does not appear to co-localize with nuclei. Do the authors have an explanation for this result? Also, the telomeres of wildtype can barely be made out.

2. The lifespan experiments of the authors, while interesting, may not be powered to detect small changes in mean longevity. However, it appears that maximal longevity could longer for several mutants than for wildtype. Perhaps soften the results to state that there is little correlation with deficiency for telomere length proteins and longevity.

3. While the authors suggest that they demonstrate no correlation between telomere length and longevity, they are using mutant backgrounds that have altered telomere length and altered proteins that interact with telomeres. One should look at changes in telomere length in a single genetic background to conclude that length per se does not affect longevity. Instead, the authors may wish to conclude that mutations in proteins that interact with telomeres do not have a significant effect on longevity.

4. The authors show that TEBP-1 and TEBP-2 interact in yeast two hybrid. Does this mean that they might form a heterodimer? If these proteins also homodimerize, then perhaps there are three distinct TEBP combinations that could modulate the distinct properties of the single and double mutants?

5. The authors argue that TEBP1 duplicated recently, after the divergence of *C. elegans* from the very close relative *C. inopinata*. Partly, this conclusion may be based on the apparent presence of a single TEBP homolog in *C. japonica*, which is ancestral to both *C. elegans* and *C. inopinata*. However, three other species had a single TEBP homolog: *C. briggsae*, *C. nigoni* and *C. sinica*. These three species are very closely related, and it is possible that their common ancestor lost one TEBP gene.

6. The authors discuss use of synteny to assess evolutionary relationships. The logic of this is not exactly clear. Perhaps the authors could offer a published example of why synteny should be a useful criteria for gene relationships. Given that so many species have two TEBP paralogs, as is the case in humans, whereas some have a single TEBP gene, as is the case in *S. pombe*, perhaps the authors could simply point to the possibility is that genetic redundancy is often maintained between these genes. If one of the TEBP genes is not well conserved, perhaps this means that it is under strong evolution selection and evolves so rapidly that it is difficult to see evolutionary relationships by comparing sequence conservation. This might also explain why no TEBP orthologs are apparent outside of *elegans* and *japonica* groups. Perhaps move synteny results to the supplemental data if their significance is not clear?

7. If *tebp-2* is not conserved, what might this mean based on *tebp-2* function?

8. It would be interesting to know if the sterility of *tebp* double mutants was due to telomere fusions, but likely out of the scope of this manuscript.

9. The authors may have identified the first double-stranded telomere binding protein where an essential role in telomerase activity is apparent.

10. In the discussion, the authors state that 'they did not identify telomere fusions in *tebp-1* and *tebp-2* mutants'. The authors should specify that this data is based on analysis of mutants after they were recently created. The authors conclude that TEBP genes may not protect telomeres from DNA damage, but this is in fact what the TEBP counterparts TRF1 and TRF2 do. It is probably better not to speculate on lack of DNA damage unless the authors show this directly, as it is possible for some damage response to occur in the absence of chromosome fusions or lethality.

11. Do the proteins identified as enriched in TEBP-1 and TEBP-2 IPs correspond to some of the proteins identified in figures 1a and 1b?

12. It was interesting that significant proteins like POT-1 might not be significantly enriched. If the authors detected telomerase could they highlight this dot in their plots?

Minor:

1. The authors state that they combined *tebp-1* and *tebp-2* with *pot-1*, *pot-2* or *mrt-1* and did not see sterility. However, the supplemental table does not show a strain mutant for *tebp-1* and *pot-2*.
2. Do the authors see somatic expression of either *tebp* gene? If not please say so.
3. Line 380. mortal germline should be Mortal Germline – phenotypes have their first letters in capitals.

Reviewer #2 (Remarks to the Author):

The authors identified two new telomere binding proteins in *C. elegans* using proteomics techniques. They confirmed telomere localization of the proteins through in vitro affinity assay and in vivo fluorescence expression. They also showed that *TEBP-1* and *TEBP-2* had different effects on telomere length, and that the double mutant animals had a synthetic phenotype because only double mutants became sterile. They presented biochemical data that proved that the *TEBP1* and *TEBP2* proteins form a complex with other telomere proteins, *MRT-1*, *POT-2* and *POT-1*. *TEBP-1* and *TEBP-2*. While their biochemical and cellular analysis results on the proteins are solid, the roles of the proteins in telomere maintenance are weakly supported. It would be better if the authors provide insights into how the proteins they identified act at telomeres.

Specific comments

1. Figure 3a and 3b: The quality of the southern blot is hardly acceptable. If they wanted to show that the proteins are involved in telomere length regulation, they should have shown southern blots of worms for many generations so that they could show progressive shortening or lengthening of telomeres. This is a crucial piece of data about the functions of the proteins identified in this study. In the figure 3b, it would be better to show the qFISH along the generations as well. Telomere lengthening does not occur in a couple of generations, but rather gradually through generations.
2. Figure 3g. The data shows that *tebp-1*; *trt-1* double mutants maintained or even lengthened the telomeres after long generations. However, it is difficult to imagine how the double mutant animals were maintained through such long generations. While *trt-1* mutants are known to become sterile after 10-15 generations, with accumulation of fusion chromosomes, the double mutant strain shown here is well maintained even after 45 generations. If this is true, then it must be that *tebp-1* mutation completely suppressed the *trt-1* mutant phenotypes. More controls that should be shown here are qFISH signals of *tebp-1* and *trt-1* single mutant animals. (It is ridiculous to ask, but.. how did you get and confirm the double mutants of *tebp-1*; *trt-1*?) By the way, the genotype of the double mutants should be written as *trt-1*; *tebp-1* because *trt-1* is on chromosome I and *tebp-1* on chromosome II.
3. It is quite striking, as the authors already said, that *tebp-1* and *tebp-2* proteins have opposite roles in telomere maintenance. The authors also showed that the proteins make complex with other telomere proteins together. Now, it is difficult to imagine how the two proteins can have opposite roles while residing in the same complex. Another point is that the double mutant of *tebp-1* and *tebp-2* are synthetically sterile, suggesting that these two proteins have redundant (similar and replaceable with each other) roles in terms of fertility, which must be different from the opposite roles of the proteins for telomeres. Are they involved in any cellular processes other than telomere binding and maintenance? Discussion or speculation on these matters would be helpful for the reader to understand the results.
4. While telomere shortening mutants (such as *trt-1* and *mrt-2*) show chromosomal fusions, *tebp-2* mutants described in this study did not show any chromosome fusion. The authors interpret this such that the proteins are not involved in DNA damage protection. However, please note that *trt-1* mutants, which is defective in telomerase, do show chromosome fusion, because they have very short telomeres after some generations. It is the short telomeres that eventually caused chromosome fusion in *trt-1* mutants, and it is conceivable that *tebp-2* mutants may show chromosome fusions in later generations. Have the authors observed the chromosomes of the mutants in their late generations?

5. The authors showed that the lifespan of the *tebp-1* and *tebp-2* mutants are not different from the wild type, *mrt-1*, or *pot-2* mutants. However, the number of worms examined was rather small. Furthermore, it does not seem to contain at least three independent trials. It is dangerous to draw any conclusion about lifespan using this incomplete data. In addition, the genetic background of *mrt-1* and *pot-2* could be different from the other strains unless they outcrossed the strains freshly with their N2 strain. Different genetic backgrounds will influence the lifespan of the worms.

6. Lines 366-: The authors claim that PLP-1, HMG-5, and CEH-37 are not telomere binding proteins in *C. elegans*. It is not fair to conclude that a protein is invalid because it was not in the list of qMS screen results; qMS screen is not the perfect method to identify ALL the proteins involved in telomere binding. qMS screen was quite good enough to identify new telomere binding proteins, but not good enough to exclude other proteins. If the method was perfect for identifying proteins involved in telomeres, how come they did not see TRT-1, the telomerase? Or other proteins that may bind to TEBP-1 and TEBP-2 proteins for specific functions? It is advised that the authors discuss other possibilities of the proteins that were not identified in their screen. For example, it is difficult to deny the biochemical and cellular evidence that CEH-37 binds telomeres in *C. elegans* if you read papers in JBC (Kim et al, 2003) and BBRC (Moon et al, 2014). There is also a paper describing a CEH-37-like telomere binding protein in mammals identified using PICH method (Déjardin and Kingston, 2009).

Reviewer #3 (Remarks to the Author):

Review NCOMMS-20-36677

"The double-stranded DNA-binding proteins TEBP-1 and TEBP-2 form a telomeric complex with POT-1"

Within this manuscript the Butter group identifies TEBP-1 and 2 as *C. elegans* telomeric binding proteins. They characterize them as double stranded TTAGGC repeat binding factors in vitro. In vivo the proteins colocalized with POT-1 and showed strongest expression in larvae and adults. Truncation mutants were viable and had opposing effects on telomere length, with TEBP-1 having longer ones and TEBP-2 having shorter ones. Telomere length did not influence life span. The double mutants displayed a degenerate germ line and were mostly sterile. According to telomere length, TEBP-2 displayed a mortal germ line phenotype. Size exclusion chromatography suggested a telomeric complex of approximately 1.1MDa, and IP-Mass spec revealed a complex of TEBP-1/2, MRT-1, POT-1, and POT-2. TEBP-1/2 interacted with each other and POT-1 was found to directly interact with TEBP-1/2, suggesting a larger telomeric complex.

In summary this is outstanding work and the isolation of the long-sought telomeric double stranding binding proteins without doubt warrants a high profile publication, such as Nature Communications. However, some questions remain and some additional work is required:

- 1) In vitro binding assays: It would be important to use oligos of different length to determine the minimal length required for binding, as well as whether longer sequences display better affinity. Similarly, it would be valuable to know whether the repeats have to be terminal, or can be embedded in non-telomeric sequences. Are single stranded to double stranded transitions required?
- 2) It would be important to map the DNA binding domain of TEBP-1/2 to understand whether the effects on telomere length are dependent on nucleic acid interactions.
- 3) Telomere length effects: Considering that strains of different telomere length can be isolated from a single parental strain and that the mutant strains were only outcrossed twice, more independent strains need to be shown to make the effects on telomere length convincing.
- 4) Are the truncations expressed? If so, do they still interact with any of the other proteins? Why not generate complete null alleles?
- 5) Is there a speculation about the short telomeres in the TEBP-2 strain? Is there a lack of interaction with telomerase? Does the TEBP-2:trt1 double have even shorter telomeres?
- 6) The 'sterility escapers' of the double mutants still produce some offspring. Is it possible to determine telomere length?
- 7) The telomeric complex is described as approximately 1.1MDa. However, the protein sizes of the

described members do not add up to that at all. Are there dimers/multimers present? Are there other factors in the complex?

Reviewer #4 (Remarks to the Author):

The manuscript "The double-stranded DNA-binding proteins TEBP-1 and TEBP-2 form a telomeric complex with POT-1" by Dietz and colleagues describes previously uncharacterized telomere binding proteins in the worm *C. Elegans*. Using mass spectrometry approaches, the authors identified two double stranded telomere binding proteins termed TEBP-1 and TEBP-2. They could confirm experimentally that both proteins bind specifically telomeric dsDNA in vitro and that the endogenously tagged protein localized to chromosome ends in vivo. Experimental CRISPR/Cas9-mediated knock-outs of *tebp-1* and *tebp-2* led to opposing effects on telomere maintenance. In TEBP-1 KO worms, telomeres were elongated in a telomerase independent manner, while in *tebp-2* KO induced telomere shortening. Finally, animals with double mutation for both genes were synthetic sterile supporting a role for TEBP-1 and TEBP-2 in germline maintenance. This is an important work because genuine telomere binding factors have so far remained elusive in *C. elegans*. The manuscript is overall well written and the experimental data and rationale behind the experiments is logic. The data presented by the authors are clear and would benefit to the field of telomere biology. Thus, I would support publication of this work in Nature Communications. However, certain details about the function of TEBP-1 and TEBP-2 are missing and would strengthen the conclusions of the authors.

1. The authors show that *tebp-1/2* double KO is synthetic sterile, with few escapers which are males. However, the fate of the germline in these mutants is not described. Are there any meiotic defects? Or DNA damage?
2. TEBP-1 and TEBP-2 have "opposing" effects on telomere maintenance. Do they alter telomerase function (localization or activity)?
3. *tebp-1* KO induces telomerase independent elongation of telomeres. The authors thus suggest maintenance of telomeres by an ALT pathway. Do they notice specific features of the ALT pathway like telomeric sister chromatid exchange (T-SCE) or presence of C-circle in this mutant?
4. As mentioned by the authors, *pot-2* KO induces telomere elongation. Although POT-2 does not seem to interact directly with TEBP-1 or TEBP-2. Could the absence of one or both proteins alter POT-2 function/localization to the telomere?
5. Does the endogenous tagging of *tebp-1* and *tebp-2* affects the animal survival?
6. *tebp-1* and *tebp-2* are dynamically expressed during the life cycle of the worm. Does it mean that the function of these factors is restricted to certain stages? Is this regulation necessary for the animal development?

We are grateful to the reviewers for their critical reading of the manuscript and excellent comments. We believe our revised version, with all the reviewer's comments addressed, is stronger and our conclusions are now better supported. Our replies to the points raised by the four reviewers can be found below. References to specific lines in the manuscript refer to the manuscript version where all changes have been accepted.

Reviewer #1:

Dietz and colleagues present a novel study where they use biochemical and genetic approaches (primarily) to define proteins that function at telomeres of *C. elegans*, a model organism that is at the forefront of many topics in contemporary experimental biology. Even though biochemical approaches have been an important unbiased method for addressing telomere biology in several experimental systems, attempts to use biochemistry to define *C. elegans* telomere proteins have yielded limited information. Telomeres from humans to yeast are known to be protected by 6 proteins that interact with telomeric DNA, two of which interact with double-stranded telomeric DNA (TRF1 and TRF2) and one of which interacts with single-stranded telomeric DNA (POT1). The only known *C. elegans* telomere proteins with homology to the Shelterin complex have homology to POT1, including the single-stranded telomere binding proteins Pot-1, Pot-2 and Mrt-1. The central power of the author's approach is to define proteins that specifically interact with a double-stranded oligonucleotide substrate of the *C. elegans* telomere sequence GGCTTA that has a single-stranded overhang, which should mimic the structure of *C. elegans* chromosome ends. Impressively, the authors discovered most proteins known to function at *C. elegans* telomeres. The authors also discovered Ku70 and Ku80 proteins that are known to play roles at telomeres in other experimental systems, which will be satisfying for the field. The major advance of this manuscript is to define *C. elegans* counterparts of TRF1 and TRF2, which have been an open question for the past 20 years. Dietz and colleagues elegantly demonstrate that two highly diverged double-stranded DNA binding proteins TEBP-1 and TEBP-2 interact with double-stranded DNA at *C. elegans* telomeres, as demonstrated by compelling oligonucleotide binding assays. The authors also demonstrate that both proteins localize to *C. elegans* telomeres, and that TEBP-1 inhibits the recombination-based telomere replication pathway called ALT, loss of which causes long telomeres and enables telomere elongation without telomerase. In contrast, the authors show that TEBP-2 causes shortened telomeres and an inability to maintain the strain, which implies deficiency for telomerase. The authors then use genetics to test the possibility that loss of TEBP-1 will suppress the telomere maintenance defect of TEBP-2 mutants, as it does with telomerase mutants. However, the authors find that loss of both double-stranded telomere binding proteins results in almost immediate sterility accompanied by underproliferated germ stem cells that

may be the result of cell lethality. This is a surprising interaction, because loss of one mammalian telomere binding protein TRF2 results in immediate telomere uncapping. The authors demonstrate that this telomere capping function is likely redundantly controlled by the two *C. elegans* double-stranded telomere binding proteins, TEBP-1 and TEBP-2, even though these proteins individually possess unique opposing functions in telomere maintenance. Overall, the authors' unanticipated results speak to multifaceted functions of proteins that interact with double-stranded telomeric DNA. These results fill a major gap in the field of telomere biology by defining two *C. elegans* double-stranded telomere binding proteins. The essential nature of double-stranded telomere binding proteins in human has made it difficult to identify essential roles for these proteins in telomerase activity (which is mediated by the single-stranded telomere binding protein POT1 and its cofactor TPP1) or in telomere maintenance via ALT. Although there is more that the authors could do to expand on their findings, this study represents a great deal of work that has yielded interesting results. In addition, this study serves as a model for unbiased biochemical approaches to studying telomere biology in any experimental system, which the authors demonstrate by performing proteomics in the related nematode *C. briggsae*. This manuscript is an important study in the field of telomere biology, filling several gaps that will move the field forward. More broadly, it demonstrates a novel and highly productive use of biochemistry in *C. elegans*, which has traditionally relied on genetics, cell and developmental biology.

1. While convincing in intensity, the morphology of the FISH signal in Fig. 3b-e appears odd, such that nuclei are not visible. Better quality images are shown for embryos in S3c-f, which show that *tebp-1* and *pot-2* mutants have much telomere FISH signal that does not appear to co-localize with nuclei. Do the authors have an explanation for this result? Also, the telomeres of wildtype can barely be made out.

Authors: We thank reviewer #1 for this comment. In figures 3b-e and S4c-f, we are showing maximum intensity projections of z-stacks. This is one reason why the morphology of the FISH and DAPI signals may appear odd and the FISH signal sometimes outside the nuclei. We enclose a file with figures (RevisionFigure1) corresponding to two single planes, for each mutant, both in extruded germlines and in embryos. As can be observed in these figures, the majority of qFISH signal overlaps with the nuclei. However, it is possible that, at very few occasions, some qFISH signal observed outside nuclei is unspecific, and that the harshness of the qFISH affected the nuclear morphology.

In the mutants showing high-intensity signals for the FISH probe (*tebp-1(xf133)* and *pot-2(tm1400)*) it is likely that signal from nuclei of lower planes is still visible in higher planes, while the corresponding DAPI staining is not, leading to the impression of missing co-localization. The lower signal in the

wildtype can be explained by the choice of settings for the qFISH. The settings for image acquisition are set for the samples with the lowest intensity to be just visible to not over-saturate the signal for the high intensity samples. In our case, these low intensity samples are *tebp-2(xf131)*. While the low signal intensity is enough for the calculations of the relative integrated density by Fiji/ImageJ, it might not be enough to sufficiently show the FISH signal in a figure image.

2. The lifespan experiments of the authors, while interesting, may not be powered to detect small changes in mean longevity. However, it appears that maximal longevity could longer for several mutants than for wildtype. Perhaps soften the results to state that there is little correlation with deficiency for telomere length proteins and longevity.

Authors: We thank reviewer #1 for this comment. As the lifespan assay does not add significantly to the overall message, and reviewer #2 raised a similar point, we have removed this data and corresponding text from the manuscript.

3. While the authors suggest that they demonstrate no correlation between telomere length and longevity, they are using mutant backgrounds that have altered telomere length and altered proteins that interact with telomeres. One should look at changes in telomere length in a single genetic background to conclude that length per se does not affect longevity. Instead, the authors may wish to conclude that mutations in proteins that interact with telomeres do not have a significant effect on longevity.

Authors: Please see our response to the comment 2 directly above.

4. The authors show that TEBP-1 and TEBP-2 interact in yeast two hybrid. Does this mean that they might form a heterodimer? If these proteins also homodimerize, then perhaps there are three distinct TEBP combinations that could modulate the distinct properties of the single and double mutants?

Authors: That is a good point; we have now provided new data to address this. Y2H data presented in the revised figures 6a,d,h and S6g,i indicates that the most likely possible combination is the TEBP-1/TEBP-2 heterodimer. In these experiments, TEBP-1 did not show interaction with itself, and no conclusion could be drawn for TEBP-2 due to the self-activation of the TEBP-2-Gal4 DNA-binding domain fusions. In the revised version of the manuscript we now include Figures 6d,f-h and S6i with Y2H data further corroborating this results and demonstrating that TEBP-1 and TEBP-2 interact via their N-terminal regions.

5. The authors argue that TEBP1 duplicated recently, after the divergence of *C. elegans* from the very close relative *C. inopinata*. Partly, this conclusion may be based on the apparent presence of a single TEBP homolog in *C. japonica*, which is ancestral to both *C. elegans* and *C. inopinata*. However, three other species had a single TEBP homolog: *C. briggsae*, *C. nigoni* and *C. sinica*. These three species are very closely related, and it is possible that their common ancestor lost one TEBP gene.

Authors: The conclusion that *tebp-2* was originated by gene duplication recently was not deduced by the apparent presence of only one *tebp* homolog in *C. japonica*. Instead, this deduction was reached due to the lack of synteny of *C. elegans* *tebp-2* with *tebp* gene orthologs in other *Caenorhabditis* species, especially in more closely related species like *C. inopinata*. The lack of synteny of *tebp-2* is contrasted by *tebp-1*, which can be found in the same syntenic block across the *Elegans* group and in *C. japonica*. Please see our reply to the next comment for further argumentation.

We were cautious and did not make any claims regarding the early origins of *tebp* gene family in the common ancestor of the *Elegans* supergroup (which includes the *Elegans* and *Japonica* groups), or in the common ancestor of the *Elegans* group. Improved *Caenorhabditis* genome assemblies and gene annotations are required to confidently make conclusions regarding the origins of this fast evolving gene family in the *Caenorhabditis* genus.

6. The authors discuss use of synteny to assess evolutionary relationships. The logic of this is not exactly clear. Perhaps the authors could offer a published example of why synteny should be a useful criteria for gene relationships. Given that so many species have two TEBP paralogs, as is the case in humans, whereas some have a single TEBP gene, as is the case in *S. pombe*, perhaps the authors could simply point to the possibility is that genetic redundancy is often maintained between these genes. If one of the TEBP genes is not well conserved, perhaps this means that it is under strong evolution selection and evolves so rapidly that it is difficult to see evolutionary relationships by comparing sequence conservation. This might also explain why no TEBP orthologs are apparent outside of *elegans* and *japonica* groups. Perhaps move synteny results to the supplemental data if their significance is not clear?

Authors: We performed a synteny analysis due to the possible strong selection and rapid evolution of *tebp* genes, which complicates the determination of orthology based only on sequence conservation. This is why we focused on the genomic collinearity of the genes flanking *tebp-1* and *tebp-2*. Closely related species, like *C. inopinata* and *C. elegans*, show low genomic rearrangement (Kanzaki et al., 2018, Nature Communications). Therefore, the absence of synteny between the second *tebp* gene

copy (that not in syntenic block of *C. elegans tebp-1*) in these two species, supports the recent evolution of these copies.

In genomics studies, analysis of synteny is a staple analysis to address the evolution of genomes (e.g. Ghiurcuta & Moret, 2014, Bioinformatics; Meyer et al., 2021, Nature; Simakov et al., 2020, Nature Ecology and Evolution). Likewise, the same evolutionary logic can be applied to analyze the evolutionary relationship of particular genes (Grimholt et al., 2015, BMC Evolutionary Biology; Lara-Ramírez et al., 2017, Development Genes and Evolution; Zhao et al., 2017, The Plant Cell). This is especially relevant when looking at fast evolving genes, for which the establishment of homology relationships by sequence alone is impossible (as is the case for *tebp-1* and *tebp-2*). Regarding groups of closely related organisms, as an example consider gene A flanked by genes B and C in the genome of animal clade 1. If the closely related animal clade 2 has two duplicates of A, termed A' and A*, but only A' is flanked by orthologs of B and C, it can be concluded that A' is in the genomic position of the ancestral A gene, and A* is a result of a duplication of A.

Furthermore, we would like to note that there are no *tebp* orthologs in nematodes beyond *Caenorhabditis*, or in humans and yeast, and we made no such claims. What we agree on is that these genes share domains similar to human and yeast RAP1. Given the fast evolution of telomere proteins (see e.g. Saint-Leandre & Levine, 2020, Trends in Genetics), it is hard to establish homology beyond particular phylogenetic groups. Therefore, we are cautious to establish parallels between distantly related organisms, despite domain similarity between *TEBP-1*, *TEBP-2* and *RAP1*. Currently, we feel we do not understand well enough how these proteins function in nematodes to make these claims. It is very hard to establish homology relationships between telomere genes in evolutionarily distant species, but this is not the case for closely related phylogenetic groups, such as the *Caenorhabditis* genus. In conclusion, we think our logic is sound and would prefer to leave part of the evolutionary analysis in the main figure.

7. If *tebp-2* is not conserved, what might this mean based on *tebp-2* function?

Authors: Upon gene duplication, the novel duplicate must first be fixed in the population by drift. After fixation, selective pressures relax on both paralogs, allowing for the accumulation of mutations in these genes. (Innan & Kondrashov, 2010, Nature Reviews in Genetics) This results in one of three main classic outcomes: 1) one of the copies becomes a pseudogene and the other retains the ancestral function; 2) there is subfunctionalization, with each of the two paralogs assuming mutually exclusive parts of the original function; or 3) one of the copies evolves a new function, termed neofunctionalization. As mutation is a stochastic process, it is possible that the ancestral paralog (in synteny with orthologs of other species) is the gene that accumulates mutations and subsequently diverges in function.

Therefore, it may be misleading to assume that TEBP-2 would be the protein with the different function in lieu of TEBP-1.

For example, as described in Hockemeyer et al., 2006, mouse has two POT1 paralogs, POT1a and POT1b, of which POT1a is syntenic to human POT1. Interestingly, POT1a and POT1b seem to share functions in prevention of DNA damage signaling, senescence, and cell proliferation but also have independent functions. In fact, POT1b, the novel paralog has retained the function of regulation of telomere structure, while POT1a is required for early embryonic development. POT1a and POT1b, like TEBP-1 and TEBP-2, nicely illustrate the fast evolution of paralog telomere factors.

While it is highly likely that *tebp-2* has originated by gene duplication in the *C. elegans* lineage after the divergence from the *C. inopinata* lineage, that does not necessarily mean enough time has elapsed for TEBP-2 neofunctionalization. However, the strikingly different mutant phenotypes in terms of telomere length indicate that TEBP-1 and TEBP-2 may have already diverged. This, associated with the fact that both TEBP-1 and TEBP-2 are simultaneously required for normal fertility (indicating potential redundancy), suggests that TEBP-1 and TEBP-2 have undergone, or are undergoing, subfunctionalization. As we feel we lack evidence for this, we preferred not to comment extensively on these aspects in the manuscript main text.

8. It would be interesting to know if the sterility of *tebp* double mutants was due to telomere fusions, but likely out of the scope of this manuscript.

Authors: Yes, we agree this would be interesting to know, yet outside the scope of this manuscript. Nevertheless, we are looking to characterize the onset of the observed germline defects and synthetic sterility in detail in the future. Such analysis in embryos and germline will be experimentally challenging and specific functional genetic tools (i.e. transgenics and conditional protein depletion systems) need to be developed to properly tackle these issues, due to the severe germline atrophy observed in the great majority of the animals (see figures 4g-h and S5b).

9. The authors may have identified the first double-stranded telomere binding protein where an essential role in telomerase activity is apparent.

Authors: We thank reviewer #1 for this comment and we do intend to more thoroughly dissect the interplay between the double-stranded telomere binders and telomerase in the future.

10. In the discussion, the authors state that ‘they did not identify telomere fusions in *tebp-1* and *tebp-2* mutants’. The authors should specify that this data is based on analysis of mutants after they were recently created.

The authors conclude that TEBP genes may not protect telomeres from DNA damage, but this is in fact what the TEBP counterparts TRF1 and TRF2 do. It is probably better not to speculate on lack of DNA damage unless the authors show this directly, as it is possible for some damage response to occur in the absence of chromosome fusions or lethality.

Authors: Reviewer #1 raises a pertinent point. While we checked this in oocytes and found no significant number of fusions in *tebp-1* and *tebp-2* mutants vs wild-type N2s, we feel this warrants further investigation in higher numbers and other cell types. This was checked in worms growing in the lab for hundreds of generations (~410 for *tebp-1* and ~432 for *tebp-2*). Therefore, we have now altered the text in the discussion, in lines 408-410, to “It remains to be determined if *tebp-1* and/or *tebp-2* mutation lead to telomere fusions, and whether the *C. elegans* telomeric complex is required to protect telomeres from DNA damage.” This textual change more clearly and cautiously illustrates our intentions.

While TEBP-1 and TEBP-2 are not homologs of TRF1 and TRF2 (in spite of similarity of their DNA-binding domains), it is tempting to speculate they may share some functional similarities. We want to note that as telomere-binding proteins evolve very quickly, the claim that TEBP-1 and TEBP-2 are counterparts to TRF1 and TRF2 is very speculative. We did not make these claims precisely due to our lack of understanding of the influence of the *C. elegans* telomeric complex in DNA damage protection at telomeres. Follow-up studies should harness the powerful genetic toolkit of *C. elegans* to understand the connection between telomere factors and DNA damage.

11. Do the proteins identified as enriched in TEBP-1 and TEBP-2 IPs correspond to some of the proteins identified in figures 1a and 1b?

Authors: Besides the core factors which are indicated in the figure (i.e. TEBP-1, TEBP-2, POT-1, POT-2, and MRT-1), no other factors were significantly enriched in both the telomeric DNA pulldowns and our IPs in a consistent manner. Reasons for this absence of interaction are proposed in the discussion (Section “Robust identification of telomere-associated proteins in *C. elegans*”, starting in line 337). This can be appreciated in a new enclosed excel table with the enriched proteins per experiment and per IP (RevisionTable1).

12. It was interesting that significant proteins like POT-1 might not be significantly enriched. If the authors detected telomerase could they highlight this dot in their plots?

Authors: We assume reviewer #1 might mean TRT-1 and not POT-1, as POT-1 is indeed enriched in all the IP-qMS shown except for figure S6c (in which case its level of enrichment is very close to the threshold). We have double-checked all IP-qMS data and we have not found any TRT-1 peptides detected. In any case, TRT-1 would not necessarily be expected to be found in our IP-qMS data. Enzymes function by means of fairly transient interactions with their substrate, which may impede detection in IP-qMS. Also, TRT-1 may elude detection due to low expression levels.

Minor:

1. The authors state that they combined tebp-1 and tebp-2 with pot-1, pot-2 or mrt-1 and did not see sterility. However, the supplemental table does not show a strain mutant for tebp-1 and pot-2.

Authors: Thank you for spotting the imprecisions in our description. We have now corrected it textually, in lines 204-205, and in figure S5a, and included information on extra crosses we have established in the meantime.

2. Do the authors see somatic expression of either tebp gene? If not please say so.

Authors: We did observe TEBP-1::GFP and TEBP-2::GFP in nuclear foci in somatic cells (RevisionFigure2). Because the main focus of our work was on proliferating germline and embryonic cells, we decided not to include these data in the manuscript.

3. Line 380. mortal germline should be Mortal Germline – phenotypes have their first letters in capitols.

Authors: We have now corrected this in the text, thank you for pointing it out.

Reviewer #2

The authors identified two new telomere binding proteins in *C. elegans* using proteomics techniques. They confirmed telomere localization of the proteins through in vitro affinity assay and in vivo fluorescence expression. They also showed that TEBP-1 and TEBP-2 had different effects on telomere length, and that the double mutant animals had a synthetic phenotype because only double mutants became sterile. They presented biochemical data that proved that the TEBP1 and TEBP2 proteins form a complex with other telomere proteins, MRT-1, POT-2 and POT-1. TEBP-1 and TEBP-2. While their

biochemical and cellular analysis results on the proteins are solid, the roles of the proteins in telomere maintenance are weakly supported. It would be better if the authors provide insights into how the proteins they identified act at telomeres.

Specific comments

1. Figure 3a ad 3b: The quality of the southern blot is hardly acceptable. If they wanted to show that the proteins are involved in telomere length regulation, they should have shown southern blots of worms for many generations so that they could show progressive shortening or lengthening of telomeres. This is a crucial piece of data about the functions of the proteins identified in this study. In the figure 3b, it would be better to show the qFISH along the generations as well.

Telomere lengthening does not occur in a couple of generations, but rather gradually through generations.

Authors: We respectfully disagree with the reviewer's assessment of the telomere Southern blot in figure 3a. We used adequate controls: a negative control with unaltered telomeres (wild-type N2) and a positive control with elongated telomeres (*pot-2[tm1440]*). For the sake of clarity, we have now added to the figure legend the number of generations of the *tebp-1* (~102 generations) and *tebp-2* (~124 generations) mutant worms (lines 1298-1307).

Mutants that regulate telomere length typically change their telomere length unidirectionally, i.e. from a wild-type telomere length, the length either becomes progressively shorter and shorter (*mrt-1* and *trt-1*, see Meier et al., 2006 and Meier et al., 2009), or progressively longer and longer (*pot-1* and *pot-2*, see Cheng et al., 2012; Raices et al., 2008; and Shtessel et al., 2013). Logically, it follows that measuring telomere length at a time-point where enough generations have elapsed after disruption of a given protein, will be sufficient to know the net effect of the mutation on telomere length. Our question was not to address the dynamics of the changes, but just to determine the net effect specific mutations have on telomere length (either shorter or longer). Thus, we do believe that the data in figure 3a-f answers the questions in an adequate manner.

2. Figure 3g. The data shows that *tebp-1;trt-1* double mutants maintained or even lengthened the telomeres after long generations. However, it is difficult to image how the double mutant animals were maintained through such long generations. While *trt-1* mutants are known to become sterile after 10-15 generations, with accumulation of fusion chromosomes, the double mutant strain shown here is well maintained even after 45 generations. If this is true, then it must be that *tebp-1* mutation

completely suppressed the *trt-1* mutant phenotypes. More controls that should be shown here are qFISH signals of *tebp-1* and *trt-1* single mutant animals. (It is ridiculous to ask, but.. how did you get and confirm the double mutants of *tebp-1; trt-1*?) By the way, the genotype of the double mutants should be written as *trt-1; tebp-1* because *trt-1* is on chromosome I and *tebp-1* on chromosome II.

Authors: Reviewer #2 is correct. To conclusively demonstrate ALT, we would require a great deal of further experimental validation, for which we miss tools at the moment, and their development would likely go beyond a timely revision. As such, we decided to remove previous figure 3g. Thank you for the insightful comment pointing out the suppression of the poor health of *trt-1* mutant animals, when *TEBP-1* is depleted. This suppression and a potential ALT mechanism should indeed be more thoroughly addressed in the future. We enclose RevisionFigure3 with the detailed information on how the *trt-1; tebp-1* strain was created and genotyped.

3. It is quite striking, as the authors already said, that *tebp-1* and *tebp-2* proteins have opposite roles in telomere maintenance. The authors also showed that the proteins make complex with other telomere proteins together. Now, it is difficult to image how the two proteins can have opposite roles while residing in the same complex. Another point is that the double mutant of *tebp-1* and *tebp-2* are synthetic sterile, suggesting that these two proteins have redundant (similar and replaceable with each other) roles in terms of fertility, which must be different from the opposite roles of the proteins for telomeres. Are they involved in any cellular processes other than telomere binding and maintenance? Discussion or speculation on these matters would be helpful for the reader to understand the results.

Authors: *MRT-1*, *POT-1*, and *POT-2*, the other known telomere factors showed the highest enrichment in *TEBP-1* and *TEBP-2* IP-qMS experiments. No other factors were consistently enriched in either *TEBP-1* or *TEBP-2* IP-qMS (RevisionTable1), but we cannot fully exclude that the opposed phenotypes of their mutants are explained by a protein interactor (or a set thereof) interacting at or near the telomeres exclusively with *TEBP-1* or *TEBP-2*. As enzymes interact with substrates transiently, it is possible that we missed interactions with telomerase protein subunit *TRT-1* in our IP-qMS setup. In this scenario, we can envision, for example, a mechanism whereby *TEBP-2* transiently facilitates *TRT-1* telomerase activity. The interplay of *TEBP-1* and *TEBP-2* with *TRT-1* will be an interesting topic of future investigations.

We do not know at the moment if *TEBP-1* and *TEBP-2* are required for distinct cellular functions, but this is something we are pursuing. We have preliminary data suggesting a role in gene expression beyond telomeres. We will have to repeat and further validate these observations and release them in a future study. We have now added to lines 376-380: "The synergistic role of *TEBP-1* and *TEBP-2* in

fertility provide a puzzling contrast with their opposed telomere length mutant phenotypes. We speculate that the requirement of TEBP-1 and TEBP-2 to fertility may be independent of their functions at telomeres. Future studies on the influence of TEBP-1 and TEBP-2 on germline/embryonic gene expression may shed light on this aspect.”

4. While telomere shortening mutants (such as *trt-1* and *mrt-2*) show chromosomal fusions, *tebp-2* mutants described in this study did not show any chromosome fusion. The authors interpret this such that the proteins are not involved in DNA damage protection. However, please note that *trt-1* mutants, which is defective in telomerase, do show chromosome fusion, because they have very short telomeres after some generations. It is the short telomeres that eventually caused chromosome fusion in *trt-1* mutant,s and it is conceivable that *tebp-2* mutants may show chromosome fusions in later generations. Have the authors observed the chromosomes of the mutants in their late generations?

Authors: Reviewer #2 raises a pertinent point. We have counted chromosome numbers in oocytes of late generation *tebp-1* and *tebp-2* animals (grown for ~410 and ~432 generations in the lab) and found no significant number of fusions in *tebp-1* and *tebp-2* mutants vs wild-type N2s. However, we feel this warrants further investigation in higher numbers and other cell types. Therefore, we have now altered the text in the discussion, in lines 408-410, to “It remains to be determined if *tebp-1* and/or *tebp-2* mutation lead to telomere fusions, and whether the *C. elegans* telomeric complex is required to protect telomeres from DNA damage.” This textual change more clearly and cautiously illustrates our intentions.

5. The authors showed that the lifespan of the *tebp-1* and *tebp-2* mutants are not different from the wild type, *mrt-1*, or *pot-2* mutants. However, the number of worms examined was rather small. Furthermore, it does not seem to contain at least three independent trials. It is dangerous to draw any conclusion about lifespan using this incomplete data. In addition, the genetic background of *mrt-1* and *pot-2* could be different from the other strains unless they outcrossed the strains freshly with their N2 strain. Different genetic backgrounds will influence the lifespan of the worms.

Authors: We thank reviewer #2 for this comment. As the lifespan assay does not add significantly to the overall message, and reviewer #1 raised a similar point, we have removed this data and corresponding text from the manuscript.

6. Lines 366-: The authors claim that PLP-1, HMG-5, and CEH-37 are not telomere bind proteins in *C. elegans*. It is not fair to conclude that a protein is invalid because it was not in the list of qMS screen results; qMS screen is not the perfect method to identify ALL the proteins involved in telomere binding. qMS screen was quite good enough to identify new telomere binding proteins, but not good enough to exclude other proteins. If the method was perfect for identifying proteins involved in telomeres, how come they did not see TRT-1, the telomerase? Or other proteins that may bind to TEBP-1 and TEBP-2 proteins for specific functions? It is advised that the authors discuss other possibilities of the proteins that were not identified in their screen. For example, it is difficult to deny the biochemical and cellular evidence that CEH-37 binds telomeres in *C. elegans* if you read papers in JBC (Kim et al, 2003) and BBRC (Moon et al, 2014). There is also a paper describing a CEH-37-like telomere binding protein in mammals identified using PICCH method (Déjardin and Kingston, 2009).

Authors: We apologize if our words seemed harsh and unfair. Our intention was not to discredit previous studies, but to thoroughly discuss possible reasons why we failed to observe certain proteins in our qMS screen. Here we used two independent qMS detection strategies that retrieved a small pool of largely overlapping factors. We also note that we wrote “our qMS screen does not support previous claims that PLP-1, HMG-5, and CEH-37 are binding to telomeric DNA in *C. elegans*.” This sentence, in our view, did not intend to cast doubt on the conclusions of previous work but instead to demonstrate that our experimental setup and results did not recapitulate these studies. However, for clarity, the revised version will not include the sentence above, and instead will read in lines 362-363: “Further studies should clarify if and how these factors interact with the telomere complex described in this work.”

TRT-1 telomerase would not necessarily be expected to be found in our qMS data. Enzymes function by means of fairly transient interactions with their substrate, which may impede detection in protein IP/DNA pulldown-qMS. Also, TRT-1 may elude detection due to low expression levels.

Reviewer #3

Within this manuscript the Butter group identifies TEBP-1 and 2 as *C. elegans* telomeric binding proteins. They characterize them as double stranded TTAGGC repeat binding factors in vitro. In vivo the proteins colocalized with POT-1 and showed strongest expression in larvae and adults. Truncation mutants were viable and had opposing effects on telomere length, with TEBP-1 having longer ones and TEBP-2 having shorter ones. Telomere length did not influence life span. The double mutants displayed a degenerate germ line and were mostly sterile. According to telomere length, TEBP-2 displayed a mortal germ line phenotype. Size exclusion chromatography suggested a telomeric complex of approximately 1.1MDa, and IP-Mass spec revealed a complex of TEBP-1/2, MRT-1, POT-1, and POT-2.

TEBP-1/2 interacted with each other and POT-1 was found to directly interact with TEBP-1/2, suggesting a larger telomeric complex.

In summary this is outstanding work and the isolation of the long-sought telomeric double stranding binding proteins without doubt warrants a high profile publication, such as Nature Communications. However, some questions remain and some additional work is required:

1) *In vitro* binding assays: It would be important to use oligos of different length to determine the minimal length required for binding, as well as whether longer sequences display better affinity. Similarly, it would be valuable to know whether the repeats have to be terminal, or can be embedded in non-telomeric sequences. Are single stranded to double stranded transitions required?

Authors: We thank Reviewer #3 for this comment, and we have now provided new data for this in our revised manuscript (New Figure S2). Now in the amended manuscript, we have performed additional fluorescent polarization assays. This revealed that TEBP-1 and TEBP-2 sustain binding to a 1.5X telomeric repeat, but they still possess the highest affinity to 2.5X telomeric repeats. Additionally, we confirmed that binding is possible at interstitial TTAGGC sequences, as both the 2.0X and 1.5X oligos are flanked by non-telomeric sequence. We have now more thoroughly addressed the *in vitro* binding dynamics of TEBP-1 and TEBP-2.

2) It would be important to map the DNA binding domain of TEBP-1/2 to understand whether the effects on telomere length are dependent on nucleic acid interactions.

Authors: In our revised manuscript we have now included Figure 6d-e, indicating that TEBP-1 and TEBP-2 bind to telomeric DNA via their third predicted homeo/myb domain (in TEBP-1 from 256-515aa and in TEBP-2 from 252-514aa). Furthermore, we have determined that TEBP-1 and TEBP-2 interact with each other via their N-terminal region containing their predicted homeo/myb domains, and with POT-1 via their unstructured C-terminal tails. These results are shown in Figure 6d, f-g. Overall, these new insights provide a more detailed understanding of the protein-protein and DNA-protein interactions of the TEBP-1/TEBP-2/POT-1 complex.

3) Telomere length effects: Considering that strains of different telomere length can be isolated from a single parental strain and that the mutant strains were only outcrossed twice, more independent strains need to be shown to make the effects on telomere length convincing.

Authors: Evidence from Cook et al., 2016 and Raices et al., 2005 has shown that there is natural variation in telomere length across different *C. elegans* strains, wild isolates, and within wild-type N2 (the wild-type we also use in this study). This is certainly a relevant point, but we would like to note that variation between clonal N2 lines seems to be lower than the variation between wild isolates (Raices et al., 2005). *C. elegans* N2 has been maintained under laboratory conditions since the 70s and given its hermaphroditic reproduction by self-fertilization, it has become strongly inbred. So, it is not surprising that N2s have lower variability in telomere length than other *C. elegans* wild isolates, which have been grown in the laboratory for considerably shorter periods and thus may still possess more genetic variation. It is also worth mentioning that variation in N2 was observed, in Raices et al., 2005, when clonal lines were founded by isolating one single hermaphrodite. This is a striking bottleneck, which does not recreate the conditions in which we normally grow our worms. This type of severe bottleneck is applied also in mortal germline experiments.

The *tebp-1* and *tebp-2* mutant alleles we used in this work were created in an N2 background and outcrossed 4 times (to N2) directly after their initial isolation. Further outcrosses to N2 were performed before specific experiments. Altogether, it is unlikely that striking variability is observed after consistently outcrossing and not subjecting the animal populations to bottlenecks. We agree that the telomere Southern blot is a measure of telomere length in a single population of worms, but this is not the case with the qFISH. In the qFISH experiments we analyzed the embryos and germlines of many animals and observed consistent telomere lengths.

4) Are the truncations expressed? If so, do they still interact with any of the other proteins? Why not generate complete null alleles?

Authors: In figure S4a-b the transcript levels of *tebp-1* and *tebp-2* can be observed in the wild-type and respective mutants. Transcript levels are slightly lower in the mutants, but the deletions create frameshifts that lead to premature stop codons in the new frame. For clarity, we have now altered the text in lines 180-181 to "...we generated *tebp-1* and *tebp-2* deletion mutants encoding truncated transcripts with premature stop codons." Furthermore, if translated, our *tebp-1* and *tebp-2* alleles only express truncated versions of the first 98-99aa, or 52aa of their wild-type proteins, respectively. In fact, in our revised manuscript we have provided Y2H evidence that the N-terminal region of TEBP-1 (1-256aa) and TEBP-2 (1-252aa), which is predicted to contain a homeo/ myb-domain, is required for their reciprocal interaction (Figure 6d,h and supplementary Fig. 6i). It is very unlikely that TEBP-1 and TEBP-2 can still interact in those mutants with only 20-38% of their predicted homeo/myb-domains. Even if the truncated proteins produced by these mutant alleles would maintain binding to the other TEBP protein, they still would not bind DNA, which we mapped to the homeo/myb-domain contained in the

protein fragments f3 (Figure 6d-e). In addition, binding to POT-1 mapped to protein fragments f7, the unstructured C-terminal tails of TEBP-1 and TEBP-2, would also not occur (see Figure 6d,f-g). Further, we note that Yamamoto et al., created *tebp-1* and *tebp-2* deletion alleles larger than ours, and observed similar synthetic sterility and telomere length mutant phenotypes. Altogether, we strongly believe that our alleles are genetic null.

5) Is there a speculation about the short telomeres in the TEBP-2 strain? Is there a lack of interaction with telomerase? Does the TEBP-2:trt1 double have even shorter telomeres?

Authors: It is possible that we missed the interaction of TEBP-2 with the telomerase protein subunit TRT-1 in our IP-qMS setup. In this scenario, we can envision a mechanism whereby TEBP-2 transiently facilitates TRT-1 telomerase activity. We did attempt TRAP assays in our laboratory but were not successful in *C. elegans* and a viable alternative is so far not published. Such an assay would have the potential to provide great insight into the interplay between TRT-1 and the entire telomeric complex described here. The time required to accomplish this would surely surpass that of this revision. Likewise, there are no available strains expressing tagged TRT-1 or an antibody recognizing *C. elegans* TRT-1. The lack of these critical reagents precludes rigorous testing of the interplay between TEBP-2 and TRT-1. This would surely be a relevant line of investigation, but we feel the creation of the necessary tools, experimental setup, and execution of these experiments would likely be a long endeavor.

6) The 'sterility escapers' of the double mutants still produce some offspring. Is it possible to determine telomere length?

Authors: That is a very good point, we considered ways of doing this viably, but we could not currently implement a proper experimental approach. For the time being, our work on telomere factors and the telomeric complex was focused on the proliferating tissues of *C. elegans*: the germline and embryos. The great majority of *tebp-1; tebp-2* double mutant animals are sterile and have severely atrophied germlines (see Figures 4a-h and S5b). We could look at somatic cell lineages, but those results would be hard to compare with the rest of our results in proliferating tissues. The double mutant escapers also do not have enough progeny to maintain an adequate number of animals for analysis (because most of them lack a germline and thus will not produce progeny). We are very interested in understanding the biology behind the germline degeneration, but we are not able to experimentally address it in the timeframe of this revision.

7) The telomeric complex is described as approximately 1.1MDa. However, the protein sizes of the described members do not add up to that at all. Are there dimers/multimers present? Are there other factors in the complex?

Authors: We acknowledge the difference between the predicted molecular weight of the complex and the individual molecular weights of the telomeric proteins we identified. What we do know, is that TEBP-1 and TEBP-2 form a heterodimer and each is able to bind POT-1. This accounts for approximately 280 kDa. MRT-1 and POT-2 account for another 100 kDa. Further multimerization of some of those proteins or of the entire complex is unknown thus far. In addition, there may be DNA associated with the telomeric complex, which is protected from nuclease treatment and that adds to the total molecular weight. We improved our discussion on this matter and now we state in the discussion in lines 394-396: "It should be noted that our model does not make any assumptions regarding complex stoichiometry. At the moment, we cannot exclude the existence of remaining DNA, despite nuclease treatment, which could add to the total molecular weight." It is also unknown whether other factors may bind to this complex, although the consistent lack of detection in our quantitative proteomics setup would argue otherwise. Further dissection of this complex at the biochemical and structural levels is required in future to illuminate these aspects, and we consider these to be beyond the current manuscript.

Reviewer #4

The manuscript "The double-stranded DNA-binding proteins TEBP-1 and TEBP-2 form a telomeric complex with POT-1" by Dietz and colleagues describes previously uncharacterized telomere binding proteins in the worm *C. Elegans*. Using mass spectrometry approaches, the authors identified two double stranded telomere binding proteins termed TEBP-1 and TEBP-2. They could confirm experimentally that both proteins bind specifically telomeric dsDNA in vitro and that the endogenously tagged protein localized to chromosomes ends in vivo. Experimental CRISPR/Cas9-mediated knock-outs of *tebp-1* and *tebp-2* led to opposing effects on telomere maintenance. In TEBP-1 KO worms, telomeres were elongated in a telomerase independent manner, while in *tebp-2* KO induced telomere shortening. Finally, animals with double mutation for both genes were synthetic sterile supporting a role for TEBP-1 and TEBP-2 in germline maintenance. This is an important work because genuine telomere binding factors have so far remained elusive in *c.elegans*. The manuscript is overall well written and the experimental data and rationale behind the experiments is logic. The data presented by the authors are clear and would benefit to the field of telomere biology. Thus, I would support

publication of this work in Nature Communications. However, certain details about the function of TEBP-1 and TEBP-2 are missing and would strengthen the conclusions of the authors.

1. The authors show that *tebp-1/2* double KO is synthetic sterile, with few escapers which are males. However, the fate of the germline in these mutants is not described. Are there any meiotic defects? Or DNA damage?

Authors: That is a good point, but specific functional genetics tools (i.e. transgenics and conditional protein depletion systems) need to be developed to properly tackle these issues. We would like to note that some of the escapers are males, but not all as commented by the reviewer (see Figure 4A-B). The great majority of *tebp-1; tebp-2* double mutants, including the escapers, are sterile with a spectrum of germline defects and atrophy (see Figures 4a-h and S5b), precluding a thorough analysis of the double mutant germline and embryos. Escaper animals with these defects have very few or no progeny. The atrophy observed in the germline may suggest defects in cell division by mitosis and/or meiosis of germ cells, as we point out in lines 373-375.

In conclusion, we agree that the synthetic sterility of *tebp-1; tebp-2* double mutants deserves a thorough characterization of its germline defects and of possible DNA damage, and this is something of great interest to us that we will pursue in the future. Indeed, we are now developing tools to interrogate DNA damage in live worms. However, we think a thorough characterization of these aspects would require more experimental tools and go beyond the scope of this work, in which we report our qMS screen, the phenotypic defects of *tebp-1* and *tebp-2*, and the telomeric complex of *C. elegans*.

2. TEBP-1 and TEBP-2 have “opposing” effects on telomere maintenance. Do they alter telomerase function (localization or activity)?

Authors: As enzymes interact with substrates transiently, it is possible that we missed interactions with telomerase protein subunit TRT-1 in our IP-qMS setup. In this scenario, we can envision, for example, a mechanism whereby TEBP-2 transiently facilitates TRT-1 telomerase activity. We did attempt TRAP assays in our laboratory but were not successful in *C. elegans*, and a viable alternative is so far not published. Such assays in *C. elegans* have the potential to provide great insight into the interplay between the telomeric complex described here (not just TEBP-1 and TEBP-2) with TRT-1. Likewise, there are no available strains expressing tagged TRT-1 and no antibodies recognizing *C. elegans* TRT-1. At the moment, contrary to other organisms, telomerase activity in *C. elegans* is not adequately understood. Its activity may differ markedly from other organisms as many short amino acid stretches

are missing (Malik et al., 2000). Notably, its RNA-component is unknown. The lack of these critical reagents and current understanding precludes rigorous testing of these hypothesis.

3. *tebp-1* KO induces telomerase independent elongation of telomeres. The authors thus suggest maintenance of telomeres by an ALT pathway. Do they notice specific features of the ALT pathway like telomeric sister chromatid exchange (T-SCE) or presence of **C-circle** in this mutant?

Authors: We thank the reviewer for raising this point. After careful consideration, we decided to remove the experiment previously shown in Figure 3g, as a greater amount of experimental validation is required to support our initial observation. It would be definitely interesting for future studies to address these aspects. However, we think that extensively testing for T-SCE and for the presence of C-circles would be beyond the scope and main conclusions of our paper. Furthermore, we have no current working setup in the laboratory to tackle these aspects experimentally in *C. elegans*.

4. As mentioned by the authors, *pot-2* KO induces telomere elongation. Although POT-2 does not seem to interact directly with TEBP-1 or TEBP-2. Could the absence of one or both proteins alter POT-2 function/localization to the telomere?

Authors: We do not have neither a POT-2 fluorescent reporter in the lab, nor an antibody recognizing POT-2 compatible with immunofluorescence in *C. elegans*. Thus, at the moment we cannot address this directly. However, we have performed TEBP-1 IP-qMS in a *tebp-2* mutant background and TEBP-2 IP-qMS in a *tebp-1* mutant background. We did not include these experiments in the manuscript as we felt that they would be beyond the main message. The results are summarized in the barplot in RevisionFigure4 for the telomeric complex factors. POT-1, POT-2, and MRT-1 still associate to TEBP-1 when TEBP-2 is depleted, and conversely, still associate to TEBP-2 when TEBP-1 is depleted (RevisionFigure4). This suggests that POT-2 is still associated with the complex in the absence of TEBP-1 or TEBP-2, presumably at the telomere.

5. Does the endogenous tagging of *tebp-1* and *tebp-2* affects the animal survival?

Authors: We constructed a strain with TEBP-1 and TEBP-2 simultaneously tagged on their endogenous locus using CRISPR-Cas9 technology, which we grew for experiments in figures 2a, 5, 6a-b, and S5a-f. This strain could be nicely grown for hundreds of generations (more than one year growing in the lab), arguing against synthetic sterility, and showed no signs of an effect on survival, like embryonic or larval lethality. Furthermore, we have created a *tebp-2::gfp* single-copy transgene, which rescues the synthetic sterility [Figure S4a, cross of *tebp-1(xf133)* x *tebp-2(xf131); xfls148(tebp-2::gfp)*]. In figure S4d-e, it is further shown that *tebp-2::gfp* in a *tebp-2* mutant background does not affect the brood

size. Altogether, these observations demonstrate that a C-terminal tag does not influence the function of TEBP-1 and TEBP-2, and thus has no effect on animal survival.

6. *tebp-1* and *tebp-2* are dynamically expressed during the life cycle of the worm. Does it mean that the function of these factors is restricted to certain stages? Is this regulation necessary for the animal development?

Authors: Reviewer #4 raises a relevant point. Whether the function of telomeric factors is restricted to certain stages is a matter of great interest to us. We did focus on the adult germline and embryos due to the highest abundance of these proteins in these stages. Our GFP reporters for TEBP-1 and TEBP-2 expression deserve close examination in the larval stages L1-L4, both in the germline and in the soma. Relevant experiments to address function of TEBP-1 and TEBP-2 across development could include: stage-specific IP-qMS to address whether these factors are still associated to their interactors like in adulthood and in embryos; and conditional depletion of TEBP-1 and/or TEBP-2 in specific tissues (initially germline vs soma) to address the requirements for the synthetic sterility.

RevisionFigure1

a

b

a) Single plane images of Fig. 3b *tebp-1(xf133)* germline (planes 3 and 14)
b) Single plane images of Fig. S4c *tebp-1(xf133)* embryo (planes 19 and 26)
Scale bar: 15 μ m

c

d

c) Single plane images of Fig. 3c *tebp-2(xf131)* germline (planes 8 and 13)
 d) Single plane images of Fig. S4d *tebp-2(xf131)* embryo (planes 7 and 22)
 Scale bar: 15 μ m

e) Single plane images of Fig. 3d *pot-2(tm1400)* germline (planes 8 and 17)
 f) Single plane images of Fig. S4e *pot-2(tm1400)* embryo (planes 16 and 24)
 Scale bar: 15 μ m

g) Single plane images of Fig. 3e N2 germline (planes 15 and 25)
 h) Single plane images of Fig. S4f N2 embryo (planes 6 and 14)
 Scale bar: 15 μ m

RevisionFigure2

- (a) Maximum projection of an L4 showing TEBP-2::GFP-foci in somatic nuclei in the tail region.
green channel: GFP, blue channel: DNA DAPI staining (scale bars: 50 μm , 15 μm)
- (b) Image of an L4 showing TEBP-2::GFP-foci in intestinal somatic nuclei.
green channel: GFP, blue channel: DNA DAPI staining (scale bars: 20 μm , 5 μm)

(c) L4 showing faint GFP-foci in intestinal somatic nuclei.

green channel: GFP, blue channel: DNA DAPI staining (scale bars: 20 μm , 5 μm)

(d) Intestinal somatic nuclei showing faint TEBP-1::GFP-foci in intestinal somatic nuclei.

green channel: GFP, blue channel: DNA DAPI staining (scale bars: 5 μm , 10 μm)

RevisionFigure3: Genotyping of *trt-1(ok410)* x *tebp-1(xf133)* double mutants

Oligonucleotides used:

tebp-1_fw: GAGCAGCTTACGAGTAGTAAGT

tebp-1_rev: CGTCTCGAGATCCTTGACCG

tebp-1_wt_fw: TAGGCTCTTGGATGCGGTTG

trt-1_fw: TGTGCAAATATACTGGAGTGGA

trt-1_rev: GGATGTCCCTGTGGAAGTCC

trt-1_wt_fw: GCGCTTCGTCGAAAAATGGA

Conditions:

Taq Buffer w 5 M Betain (10x)	1 μ l
dNTPs	0.1 μ l
Primer for	0.2 μ l
Primer rev	0.2 μ l
Taq Polymerase	0.05 μ l
Worm lysis	2 μ l
BSA (10 mg/ml)	0.05 μ l
H ₂ O	To 10 μ l

94°C	30 sec	35 x
94°C	30 sec	
60°C	1 min	
68°C	50 sec	
68°C	5 min	
10°C	Pause	

tebp-1 PCR (1): *tebp-1_fw* + *tebp-1_rev* → WT: 2028 bp, mut: 607 bp

tebp-1 PCR (2): *tebp-1_wt_fw* + *tebp-1_rev* → WT: 600 bp, mut: no signal due to deletion

trt-1 PCR (1): *trt-1_fw* + *trt-1_rev* → WT: 1650 bp, mut: 207 bp

trt-1 PCR (2): *trt-1_wt_fw* + *trt-1_rev* → WT: 507 bp, mut: no signal due to deletion

RevisionFigure3. Strain *trt-1(ok410); tebp-1(xf133)* used for the experiment in previous Fig. 3g was derived from a sibling of worms 1.1-1.4 (numbered in red). Worms 1.1 to 1.4 were picked for re-confirmation of homozygosity of both mutations. Please disregard the genotyping of worms 2.1-2.4 as these refer to worms used in another experiment. Upper row: *tebp-1* PCR (1) in odd wells and *tebp-1* PCR (2) in even wells. Lower row: *trt-1* PCR (1) in odd wells and *trt-1* PCR (2) in even wells. These PCRs are numbered accordingly in purple. R06 refers to *tebp-1(xf133)* as single mutant control. TRT refers to *trt-1(ok410)* as single mutant control. N2 is the WT control. EL is the empty lysis where no DNA template is added to the PCR mix to detect potential contaminations of the reagents.

RevisionFigure4

RevisionFigure4: Peptide detection intensities of the label-free quantification (shown as a log₂) of the indicated IPs show that MRT-1, POT-1, and POT-2 are still detected in TEBP-1 IPs, irrespective of the presence (light green bar) or absence (dark green bar) of TEBP-2. The converse is also true: MRT-1, POT-1, and POT-2 are still detected in TEBP-2 IPs in the presence (light blue bar) and absence (dark blue bar) of TEBP-1.

REVIEWERS' COMMENTS

Reviewer #1 (Remarks to the Author):

The manuscript by Butter and colleagues has been nicely improved and the authors have responded well in their rebuttal letter to all comments from the reviewers. I appreciate their reasonable and productive responses. Overall, this manuscript describes a very important contribution to the study of *C. elegans* telomeres by defining two double-stranded DNA telomere binding proteins TEBP-1 and TEBP-2 and elegantly showing that the POT-1 single-stranded DNA binding protein bridges these dsDNA telomere binding proteins to other single-stranded DNA telomere proteins POT-2 and MRT-1.

I can recommend this manuscript for publication as it currently reads, given several exceptional and elegant contributions. That said, some of the nice explanations in the rebuttal letter were not added to the manuscript in a way that might help to clarify what is happening for the reader. Below I provide a few suggestions that might further strengthen this intriguing manuscript, which should be very well received by the field of telomere biology.

Comments:

- "We analyzed telomere length in these mutants by carrying out a telomere Southern blot on mixed stage animals". In their rebuttal letter, the authors nicely explain the logic of looking at single DNA preps from each of the mutant strains by stating that *tebp-1* and *tebp-2* had been grown for ~100 generations prior to DNA analysis. It would be helpful to mention this in the text of the Results, so the reader can understand why you present data from single DNA samples per mutant.

- These "synthetic sterility escapers" were subfertile, siring less than 60 offspring. Importantly, a *tebp-2::gfp* single-copy transgene fully rescued the appearance of sterility". Conclude that this epitope-tagged protein not only localizes to telomeres but is functional?

- "We further quantified the synthetic sterility on brood size by picking L2-L3 progeny of *tebp-2*; *tebp-1 +/-* mutants, blind to genotype and germline health, rearing those animals at 20°C or 25°C, later counting their brood sizes, and genotyping each animal." conclude that this revealed strong immediate reductions in fertility for the double mutants occur independent of temperature?

"*tebp-1(xf133)* remained fertile across generations, like wild-type, while *tebp-2(xf131)* showed a *Mrt* phenotype". This will be confusing to read if the authors include a statement that DNA from 100 generation *tebp-1* and *tebp-2* mutants was used for Southern blotting (above). Yet the maximum generations of *tebp-2* is 30 at 25°C. The authors might wish to state here that mutant strains of *tebp-2* can be grown for >100 generations at low temperature of 20°C but that a possible telomerase defect is observed if at 25°C? If this is the case, then the authors may be able to conclude that *tebp-2* causes short but stable telomeres at 20°C but is likely to be essential for telomere maintenance at 25°C?

- "the onset of which is delayed compared to *mrt-1(tm1354)* and *trt-1(ok410)*, indicating a slower deterioration of germline health over generations". This might also be explain if *mrt-1* and *trt-1* mutants contained already shortened telomeres that resulted in faster sterility than observed for a newly created *tebp-2* mutation where all telomeres were initially wildtype in length.

- The authors are cautious about interpreting the evolutionary origin of these proteins and prefer not to call them paralogs of TRF1/TRF2 proteins that are present in mammals and yeast. I would suggest that it might be worth adding a discussion paragraph about the possible relationship between TRF1/TRF2 and TEBP-1/TEBP-2, as some readers from the telomere field may wish to conclude that these proteins could be evolutionarily related. My impression is that the authors found homology of TEBP-1/TEBP-2 to RAP1. Therefore, they could suggest that TRF1/TRF2 could have been lost in nematodes and replaced by RAP1, which may have independently evolved to become the dsDNA telomere binding protein in *S. cerevisiae*. Another possibility is that the TRF1/TRF2 telomere binding proteins were ancestral to TEBP-1/TEBP-2, but that these proteins are

very rapidly evolving in nematodes, resulting in proteins that now look more like RAP1 than TRF1/TRF2. This discussion would help readers from the telomere field to better understand why these proteins were so difficult to identify in nematodes and might help to round out the Discussion.

- The model in Fig. 6i concerns the possible structure of the *C. elegans* telomeric complex based on IP and two hybrid analysis. The authors do a nice job of showing that POT-1 is necessary to bridge interactions between TEBP-1 and TEBP-2 with MRT-1 and POT-2. But the authors show evidence that nuclease treatment does not disrupt interactions between the wildtype complex, which may be at odds with the model in Fig. 6i that shows MRT-1 and POT-2 binding to single-stranded DNA independently of POT-1. As this entire complex is nuclease-resistant, I suggest creating an additional alternative model, where POT-1 forms direct contacts with MRT-1 and POT-2, thereby creating a physical bridge between the dsDNA telomere binding proteins TEBP-1/TEBP-2 and MRT-1 or POT-2. The authors may not have formal proof for this bridge, but it could be shown by dotted lines as a proposed interaction that remains to be tested experimentally, although it may be supported by their nuclease experiments.

- 'The synergistic role of TEBP-1 and TEBP-2 in fertility provide a puzzling contrast with their opposed telomere length mutant phenotypes. We speculate that the requirement of TEBP-1 and TEBP-2 to fertility may be independent of their functions at telomeres.'
Another possibility is that the synergistic role of TEBP-1 and TEBP-2 in fertility reflects a common function of double-stranded telomere binding proteins in telomere protection. Reflecting this, loss of *tebp-1* and *tebp-2* results in increased levels of males, which suggests a chromosome instability phenotype. I would suggest that this is an exciting interpretation of the results, which might indicate how to find other genes that play essential roles in telomere end-protection. The authors bring up the possibility of genome instability in the second to last paragraph, but it might be easier to place this contrasting statement in the Discussion of effects on fertility. Note that the authors' suggestion that the sterility could reflect a distinct non-telomeric function of these proteins is also clearly possible.

- I now understand better the concept of synteny in the context of gene duplication. The authors are suggesting that because TEBP-2 is in a different region of the genome in *C. elegans* and *C. inopinata*, that there may have been a single TEBP gene in ancestors of both these very closely related species, which duplicated independently after the divergence of a common ancestor. Given that two TEBP genes are present in other more distantly related nematode species, one of which does not display synteny, this could mean that the TEBP gene duplicates frequently and lands in different regions of the genome. A second possibility is that the common ancestor to all of these closely related species had two TEBP genes, one of which was in a chromosome location that possesses strong synteny and that for some reason the second gene is in a chromosome region that either does not display synteny or else tends to move to different parts of the genome on a regular basis. In other words, the authors could propose a second possibility, that *tebp-2* somehow breaks the rules of synteny independent of *tebp-1* being frequently duplicated. In my mind, this second possibility is a more interesting option that might be useful for those who study gene and genome evolution to consider.

Minor comments:

- abstract "Notably, *tebp-1*; *tebp-2* double mutant animals have synthetic sterility, with germlines showing signs of severe mitotic and meiotic arrest". Conclude that they function redundantly to promote cell proliferation or differentiation?

- summary sentence: 'which are together required for fertility and have distinct individual effects on telomere dynamics'

- 'When a subset of telomeres shorten beyond a critical point, cellular senescence is triggered'. This should read cellular senescence or apoptosis.

Reviewer #2 (Remarks to the Author):

The authors successfully addressed most of the comments. Considering the pandemic situation, the reviewer understands that the authors may not be able to try all the experimental improvement, and now thinks that the manuscript is acceptable for publication.

Reviewer #3 (Remarks to the Author):

I am satisfied with the revision and the responses by the authors. I maintain that this is outstanding work and look forward to it in print.
Jan Karlseder

Reviewer #4 (Remarks to the Author):

The authors have satisfactorily addressed my comments. Congratulations for a very nice work !

J Dejardin

Reviewer #1 (Remarks to the Author):

The manuscript by Butter and colleagues has been nicely improved and the authors have responded well in their rebuttal letter to all comments from the reviewers. I appreciate their reasonable and productive responses. Overall, this manuscript describes a very important contribution to the study of *C. elegans* telomeres by defining two double-stranded DNA telomere binding proteins TEBP-1 and TEBP-2 and elegantly showing that the POT-1 single-stranded DNA binding protein bridges these dsDNA telomere binding proteins to other single-stranded DNA telomere proteins POT-2 and MRT-1. I can recommend this manuscript for publication as it currently reads, given several exceptional and elegant contributions. That said, some of the nice explanations in the rebuttal letter were not added to the manuscript in a way that might help to clarify what is happening for the reader. Below I provide a few suggestions that might further strengthen this intriguing manuscript, which should be very well received by the field of telomere biology.

We thank Reviewer 1 for further suggestions to improve the manuscript and where changes were added the line numbers refer to the manuscript with tracked changes sent to the editor.

Comments:

- “We analyzed telomere length in these mutants by carrying out a telomere Southern blot on mixed stage animals”. In their rebuttal letter, the authors nicely explain the logic of looking at single DNA preps from each of the mutant strains by stating that *tebp-1* and *tebp-2* had been grown for ~100 generations prior to DNA analysis. It would be helpful to mention this in the text of the Results, so the reader can understand why you present data from single DNA samples per mutant.

The authors appreciate the suggestion and the respective part in the results section now reads: “We analyzed telomere length in the mutants after propagation for more than 100 generations, sufficient to establish a “steady-state” telomere length phenotype, by carrying out a telomere Southern blot on mixed-stage animals. Interestingly, while *tebp-1(xf133)* shows an elongated telomere phenotype comparable to the *pot-2(tm1400)* mutant, *tebp-2(xf131)* shows a shortened telomere phenotype (Fig. 3a), similar to *mrt-1* mutants.” (lines 190-194)

- These “synthetic sterility escapers” were subfertile, siring less than 60 offspring. Importantly, a *tebp-2::gfp* single-copy transgene fully rescued the appearance of sterility”. Conclude that this epitope-tagged protein not only localizes to telomeres but is functional?

The authors thank Reviewer 1 for pointing out this important detail and changed the respective part to read: “Importantly, a *tebp-2::gfp* single-copy transgene fully rescued the appearance of sterility ,

demonstrating that the C-terminal tag does not disrupt TEBP-2 function (Supplementary Fig. 5a). (lines 212/213)

- “We further quantified the synthetic sterility on brood size by picking L2-L3 progeny of *tebp-2*; *tebp-1* +/- mutants, blind to genotype and germline health, rearing those animals at 20°C or 25°C, later counting their brood sizes, and genotyping each animal.” conclude that this revealed strong immediate reductions in fertility for the double mutants occur independent of temperature?

Based on this suggestion we have added the following sentence after: “This revealed that the immediate synthetic sterility phenotype is not dependent on temperature, as the reduction of progeny numbers was apparent at both 20°C and 25°C.” (lines 220/221)

In addition we also added this information in the discussion: “Moreover, while the fertility of *tebp-1* and *tebp-2* animals is not compromised, *tebp-1*; *tebp-2* double mutants show highly penetrant synthetic sterility irrespective of the temperature the animals are grown at, indicating that TEBP-1 and TEBP-2 contribute to normal fertility (Fig. 4, Supplementary Fig. 5).” (lines 388-390)

“*tebp-1*(xf133) remained fertile across generations, like wild-type, while *tebp-2*(xf131) showed a Mrt phenotype”. This will be confusing to read if the authors include a statement that DNA from 100 generation *tebp-1* and *tebp-2* mutants was used for Southern blotting (above). Yet the maximum generations of *tebp-2* is 30 at 25°C. The authors might wish to state here that mutant strains of *tebp-2* can be grown for >100 generations at low temperature of 20°C but that a possible telomerase defect is observed if at 25°C? If this is the case, then the authors may be able to conclude that *tebp-2* causes short but stable telomeres at 20°C but is likely to be essential for telomere maintenance at 25°C?

The authors agree that the scenario suggested by the reviewer is possible, but want to remark that the Mortal Germline experiment is not only conducted at 25°C but also introduces a rigorous bottleneck in the population, as only six animals are transferred each time. This strict bottleneck is not applied to the worms growing at 20°C under normal conditions with standard maintenance.

- “the onset of which is delayed compared to *mrt-1*(tm1354) and *trt-1*(ok410), indicating a slower deterioration of germline health over generations”. This might also be explain if *mrt-1* and *trt-1* mutants contained already shortened telomeres that resulted in faster sterility than observed for a newly created *tebp-2* mutation where all telomeres were initially wildtype in length.

The authors thank the Reviewer for pointing out this potential conclusion but we want to clarify that the *tebp-2* mutant animals used for the Mortal Germline experiment were not the newly created

mutants. As the Mortal Germline experiment was performed even after the Southern Blot and qFISH experiments we expect the animals to possess the same established telomere length phenotype. Before the experiment all strains were outcrossed against WT twice so the telomere lengths should have been mended equally for *tebp-2* as well as *mrt-1* and *trt-1* mutants. To make this point more apparent, we added the information about the mutants in lines 249-254: “We thus conducted a Mortal Germline assay at 25°C using late generation mutants, and found that *tebp-1* and *tebp-2* mutants displayed opposing phenotypes in line with their differing effects on telomere length: While *tebp-1(xf133)* remained fertile across generations, like wild-type, *tebp-2(xf131)* showed a Mrt phenotype (Fig. 4i), the onset of which is delayed compared to *mrt-1(tm1354)* and *trt-1(ok410)*, indicating a slower deterioration of germline health over generations.”

- The authors are cautious about interpreting the evolutionary origin of these proteins and prefer not to call them paralogs of TRF1/TRF2 proteins that are present in mammals and yeast. I would suggest that it might be worth adding a discussion paragraph about the possible relationship between TRF1/TRF2 and TEBP-1/TEBP-2, as some readers from the telomere field may wish to conclude that these proteins could be evolutionarily related. My impression is that the authors found homology of TEBP-1/TEBP-1 to RAP1. Therefore, they could suggest that TRF1/TRF2 could have been lost in nematodes and replaced by RAP1, which may have independently evolved to become the dsDNA telomere binding protein in *S. cerevisiae*. Another possibility is that the TRF1/TRF2 telomere binding proteins were ancestral to TEBP-1/TEBP-2, but that these proteins are very rapidly evolving in nematodes, resulting in proteins that now look more like RAP1 than TRF1/TRF2. This discussion would help readers from the telomere field to better understand why these proteins were so difficult to identify in nematodes and might help to round out the Discussion.

We want to reiterate that TEBP-1 and TEBP-2 are not homologues to TRF1/TRF2. However, these two pairs of evolutionarily unrelated paralog telomere-binding proteins may be a case of convergent evolution on the functional level. TRF1/TRF2 containing their distinctive TRFH domain are to our knowledge only found in vertebrates. We thus do not find it reasonable to assume that TRF1 and TRF2 have originated in more basal, non-vertebrate, animal lineages, and then lost in all the non-vertebrate lineages. We find it more likely that the TRFH domain containing genes TRF1 and TRF2 trace back to a common vertebrate ancestor. Furthermore, the myb-like DNA binding domains of TRF1 and TRF2 are C-terminal, while they are N-terminal in TEBP-1 and TEBP-2. This structural feature is further support for our assumption that these protein paralogs are not phylogenetically related. Also, TRF1 and TRF2 each have one single DNA-binding domain, while TEBP-1 and TEBP-2 may have tandem DNA-binding domains, much like RAP1. Although, the DNA-binding domains of

TEBP-1 and TEBP-2 are somewhat similar to those of RAP1, we cannot faithfully determine a phylogenetic relationship due to the fast evolution of these proteins. To emphasize these points, which are indeed important to the reader as the reviewers correctly points out, an altered version of this paragraph is now included in the discussion in lines (407-415).

- The model in Fig. 6i concerns the possible structure of the *C. elegans* telomeric complex based on IP and two hybrid analysis. The authors do a nice job of showing that POT-1 is necessary to bridge interactions between TEBP-1 and TEBP-2 with MRT-1 and POT-2. But the authors show evidence that nuclease treatment does not disrupt interactions between the wildtype complex, which may be at odds with the model in Fig. 6i that shows MRT-1 and POT-2 binding to single-stranded DNA independently of POT-1. As this entire complex is nuclease-resistant, I suggest creating an additional alternative model, where POT-1 forms direct contacts with MRT-1 and POT-2, thereby creating a physical bridge between the dsDNA telomere binding proteins TEBP-1/TEBP-2 and MRT-1 or POT-2. The authors may not have formal proof for this bridge, but it could be shown by dotted lines as a proposed interaction that remains to be tested experimentally, although it may be supported by their nuclease experiments.

We appreciate the suggestion, and amended the model in Fig. 6i respectively, introducing potential interactions between POT-1 and POT-2/MRT-1. As we still lack experimental proof for these interactions the reciprocal interactions are depicted with arrows and a question mark.

- 'The synergistic role of TEBP-1 and TEBP-2 in fertility provide a puzzling contrast with their opposed telomere length mutant phenotypes. We speculate that the requirement of TEBP-1 and TEBP-2 to fertility may be independent of their functions at telomeres.' Another possibility is that the synergistic role of TEBP-1 and TEBP-2 in fertility reflects a common function of double-stranded telomere binding proteins in telomere protection. Reflecting this, loss of *tebp-1* and *tebp-2* results in increased levels of males, which suggests a chromosome instability phenotype. I would suggest that this is an exciting interpretation of the results, which might indicate how to find other genes that play essential roles in telomere end-protection. The authors bring up the possibility of genome instability in the second to last paragraph, but it might be easier to place this contrasting statement in the Discussion of effects on fertility. Note that the authors' suggestion that the sterility could reflect a distinct non-telomeric function of these proteins is also clearly possible.

We added the changes in lines 437-439 to incorporate the reviewer's suggestions but wished to keep the original placement of our statement.

- I now understand better the concept of synteny in the context of gene duplication. The authors are suggesting that because TEBP-2 is in a different region of the genome in *C. elegans* and *C. inopinata*, that there may have been a single TEBP gene in ancestors of both these very closely related species, which duplicated independently after the divergence of a common ancestor. Given that two TEBP genes are present in other more distantly related nematode species, one of which does not display synteny, this could mean that the TEBP gene duplicates frequently and lands in different regions of the genome. A second possibility is that the common ancestor to all of these closely related species had two TEBP genes, one of which was in a chromosome location that possesses strong synteny and that for some reason the second gene is in a chromosome region that either does not display synteny or else tends to move to different parts of the genome on a regular basis. In other words, the authors could propose a second possibility, that *tebp-2* somehow breaks the rules of synteny independent of *tebp-1* being frequently duplicated. In my mind, this second possibility is a more interesting option that might be useful for those who study gene and genome evolution to consider.

We thank the reviewer for this suggestion and while we agree with there being a potential second possibility of *tebp-2* breaking synteny rules, we do not wish to extend the discussion of synteny in the manuscript further at this point. As more nematode genomes and data sets are not yet readily available we do not want to draw stronger conclusions without being able to properly support these claims.

Minor comments:

- abstract "Notably, *tebp-1*; *tebp-2* double mutant animals have synthetic sterility, with germlines showing signs of severe mitotic and meiotic arrest". Conclude that they function redundantly to promote cell proliferation or differentiation?

We appreciate the suggestion but think that this sentence is for now sufficient to underline our work. In future follow-up studies we will further investigate the true synergy/redundancy of the two telomeric factors.

- summary sentence: 'which are together required for fertility and have distinct individual effects on telomere dynamics'

We have added the reviewer's suggestion to the summary sentence in the editorial comments.

- 'When a subset of telomeres shorten beyond a critical point, cellular senescence is triggered'. This should read cellular senescence or apoptosis.

We thank the reviewer for pointing this out and amend the respective sentence in the introduction accordingly (see line 59).